# Circuit Complexity from Cosmological Islands

**Sayantan Choudhury** [1,2,*,†] , **Satyaki Chowdhury** [1,2], **Nitin Gupta** [3], **Anurag Mishara** [4], **Sachin Panneer Selvam** [5], **Sudhakar Panda** [1,2], **Gabriel D. Pasquino** [6], **Chiranjeeb Singha** [7,8] **and Abinash Swain** [9]

1 National Institute of Science Education and Research, Bhubaneswar 752050, Odisha, India; satyaki.chowdhury@niser.ac.in (S.C.); panda@niser.ac.in (S.P.)
2 Homi Bhabha National Institute, Training School Complex, Anushakti Nagar, Mumbai 400085, India
3 Department of Physical Sciences, Indian Institute of Science Education & Research Mohali, Mohali 140306, Punjab, India; nickeltingupta@gmail.com
4 Department of Physics and Astronomy, National Institute of Technology, Rourkela 769001, Odisha, India; anuragmishra199@gmail.com
5 Department of Physics, Birla Institute of Technology and Science, Pilani, Hyderabad Campus, Hyderabad 500078, India; f20160863@hyderabad.bits-pilani.ac.in
6 Department of Physics and Astronomy, University of Waterloo, 200 University Ave W, Waterloo, ON N2L 3G1, Canada; gabrielpasquino@gmail.com
7 Department of Physical Sciences, Indian Institute of Science Education and Research Kolkata, Mohanpur 741246, India; chiranjeeb.singha@gmail.com
8 Chennai Mathematical Institute, H1, SIPCOT IT Park, Siruseri, Kelambakkam 603103, India
9 Department of Physics, Indian Institute of Technology Gandhinagar, Palaj, Gandhinagar 382355, India; abinashswain2010@gmail.com
* Correspondence: sayantan.choudhury@niser.ac.in or sayanphysicsisi@gmail.com
† This project is the part of the non-prot virtual international research consortium "Quantum Aspects of Space-Time & Matter" (QASTM).

**Abstract:** Recently, in various theoretical works, path-breaking progress has been made in recovering the well-known page curve of an evaporating black hole with quantum extremal islands, proposed to solve the long-standing black hole information loss problem related to the unitarity issue. Motivated by this concept, in this paper, we study cosmological circuit complexity in the presence (or absence) of quantum extremal islands in negative (or positive) cosmological constant with radiation in the background of Friedmann-Lemaître-Robertson-Walker (FLRW) space-time, i.e., the presence and absence of islands in anti de Sitter and the de Sitter space-time having SO(2, 3) and SO(1, 4) isometries, respectively. Without using any explicit details of any gravity model, we study the behavior of the circuit complexity function with respect to the dynamical cosmological solution for the scale factors for the above mentioned two situations in FLRW space-time using squeezed state formalism. By studying the cosmological circuit complexity, out-of-time ordered correlators, and entanglement entropy of the modes of the squeezed state, in different parameter space, we conclude the non-universality of these measures. Their remarkably different features in the different parameter space suggests their dependence on the parameters of the model under consideration.

**Keywords:** quantum extremal surface; cosmological complexity; cosmological islands

# Contents

$\mathcal{D}$r. $\mathcal{S}$ayantan $\mathcal{C}$houdhury
**would like to dedicate this work**
**to**
**his lovable father**
**and**
**prime inspiration**
$\mathcal{P}$rofessor $\mathcal{M}$anoranjan $\mathcal{C}$houdhury
**who**
**recently have passed away due to**
**COVID-19.**

## 1. Prologue

In recent times, various outstanding quantum information theory concepts have been extensively used to decode the enormous number of hidden secrets of a quantum theory of gravity. Among all of these well known successful tools and techniques, in this paper, we mainly concentrate on the underlying physics of circuit complexity [1]. The notion of circuit complexity was first introduced in physics by Professor Leonard Susskind to understand the mysteries of quantum aspects of black holes. It has been an important and very successful probe in diagnosing certain interesting features underlying the system. Before going into the details of this paper's subject material, let us familiarize the readers with the bigger picture to develop the strong background motivation of this paper.

The concept of quantum extremal islands [2,3] has been a very successful theoretical concept in reproducing the page curve for an evaporating black hole [4] from semi-classical considerations, which has been recently studied in various remarkable works [2,5–8]. This program came into the picture to propose a possible solution to the long-standing famous Hawking information loss paradox [9–11]; related to the preserving unitarity in evaporating black holes. A better understanding of von-Neumann entropy was needed to understand Hawking radiation. As a result, the idea of page curve remained incomplete and led to many contradicting views. It was found that the consistent way of computing the entropy

involved an area of a surface that is not the horizon. It is the surface that extremizes the generalized entropy, and, hence, the name arises *quantum extremal surface* (QES) in the associated literature. In References [2,5,6,12], the authors explicitly showed that using QES, one can systematically start from a pure quantum state black hole. During the entire process of evaporation, a consistent definition of the fine-grained entropy can be given. The curve displayed by this entropy shows that the expected *page curve* is devoid of any contradictions. Hence, the entropy computed, including QES was consistent with the unitary evolution as expected from quantum mechanics. To successfully describe the process, we need to add a bulk region in the QES after the *page* transition time, which aids in reproducing the page curve. These bulk regions are known as *islands*.

Over the years, many formulations for entanglement entropy came into the picture and were applied in various cases [13–18]. However, the computation of this entanglement entropy is not always very trivial. One can expect to find various underlying unexplored features of entanglement entropy by studying complexity without going into the technical details of computing the entanglement entropy from a given gravitational paradigm taking motivation from Leonard Susskind's path-breaking idea [19]. He explicitly showed the rate of change of complexity [20–22] is equal to the product of entropy and the equilibrium saturation temperature [19], i.e., $dC/dt = ST$. Thus, he established a connecting relation between the circuit complexity and the entanglement entropy for black hole systems. However, this is a conjectured relation and it has not been explicitly proved. Hence one cannot take this relation to be universal. An explicit check of the validity of this conjecture might be a very useful prospect and a generalization of this conjecture for a general gravitational system might be reveal extremely interesting features of the considered system.

The notion of cosmological circuit complexity is intimately related with the *out-of-time-ordered correlation* (OTOC) functions [23–27] which is generally used as a probe of quantum chaos [28]. (This relationship between OTOC and complexity in presence of quantum extremal islands within the framework of FLRW cosmology, particularly for the two solutions for the scale factors may not be universal).

$$\textbf{Cosmological Island bound}: \quad \textbf{OTOC} = \exp(-\bar{c}\exp(\lambda a)) = \exp(-\mathcal{C}) \qquad (1)$$

where the equality holds for the maximal chaos inside the Island-inspired bulk region in the FLRW cosmological background. This relation is also extremely useful in the context of condensed matter systems, where computing the OTOCs is not always a trivial task. Here the quantity, $\bar{c} \sim N^{-1/2}$, where $N$ represents the number of degrees of freedom. Additionally, it is important to note that, the *quantum Lyapunov exponent*, to describe the quantum description of the chaotic phenomena has to satisfy the following constraint, commonly cited as the *Maldacena Shenker Stanford* (MSS) bound in the associated literature, as given by [28] (In the context of FLRW cosmology by utilizing the fact that, the *quantum Lyapunov exponent* can be computed from the scale factor dependence of the Cosmological Complexity, i.e.,).

$$\lambda = \frac{d\ln\mathcal{C}(a)}{da}, \qquad (2)$$

one can give a Cosmological extended version of the MSS bound, which is given by:

$$\textbf{cosmological MSS bound}: \quad \beta^{-1} = T \geq \frac{1}{2\pi}\frac{d\ln\mathcal{C}(a)}{da}. \qquad (3)$$

We will talk about the technical details and derivations of this extended bound in the latter half of this paper:

$$\textbf{MSS bound}: \quad \lambda \leq \frac{2\pi}{\beta} = 2\pi T, \quad \text{where } \beta = \frac{1}{T} \text{ in } \hbar = 1, c = 1, k_B = 1. \qquad (4)$$

Here, $\beta$ represents the inverse equilibrium temperature of the chaotic system during saturation of the OTOC at a large evolutionary scale. In this connection, here, it is essential to note that recently in Reference [29], Tom Hartman and co-authors investigated the cosmological islands and the conditions for them to appear in gravitational curved space-time without singularities. They showed that islands appear in the four-dimensional FLRW cosmology with radiation and negative cosmological constant (cc). In contrast, in the positive cosmological constant case, islands are absent in the bulk region. We take a leaf from this paper, and, using the connection between complexity and OTOC as our primary guiding principle, we reinvestigate these two particularly well-known cases from the perspective of quantum chaos. We also study the behavior of these measures in two different parameter spaces to show the non-universality of these measures. In this paper, we show that the complexity for four-dimensional FLRW with radiation and negative cc (AdS case, with SO(2, 3) isometry) resembles page curve behavior which is absent for the positive cc (dS case, with SO(1, 4) isometry). We also compute the entanglement entropy in the language of squeezed parameters and study their evolution with the scale factor. The purpose of using this squeezed state formalism is to translate the given problem entirely in the language of a general quantum mechanical system. The squeezed state formalism also provides an elegant and efficient way of computing the entanglement entropy particularly, von-Neumann and Renyi entropy in terms of the squeezed state parameters. It also provides a way to connect the entanglement entropy with the circuit complexity for the model under consideration. Thus, though not exactly, one will be able to develop an idea about how the entanglement entropy of a particular system is related to the circuit complexity and whether the conjectured relation proposed by Prof. Leonard Susskind holds true in that particular case or not. We also observed that the complexity measure and the OTOCs are parameter dependent quantities and have widely different behavior in different regions of parameter space.

The present computation of circuit complexity, entanglement entropy, the four-point OTOCs, and many physical observables and quantities can be computed by following the standard techniques of quantum field theory of curved space-time for a general gravitational metric [30–33], other than the situation where one tries to understand the black hole geometry with and without quantum extremal islands in presence of time-evolving FLRW metric. Even in the global and planar coordinates (inflationary patch) of De Sitter space, one can compute a quantum effective action or a partition function by following a semiclassical approach, where gravity is taken to be classical and fields which are embedded in the gravity are taken to be quantum, if due to some additional physical criteria or speciality in the physical set up, anisotropy and inhomogeneity are introduced from the starting point [30–38]. However, once we talk about pure FLRW space-time from the starting point, things are not as simple as mentioned above. Once a field is embedded in the FLRW geometrical background (in presence or absence of islands), then due to the homogeneity and isotropy property of the FLRW metric, fields are considered to be only time-dependent, provided no influence of additional physics is considered here. Most importantly this is not an assumption. Now, as we are interested in the quantum fluctuations of the fields rather than the background embedded field, one usually introduces the inhomogeneity and anisotropy in the metric, as well as in the field in such a way that the underlying setup and physical outcomes do not get effected due to this. In this construction, the metric and the field after introducing the anisotropy and inhomogeneity can be written as:

$$g_{\mu\nu}(\mathbf{x}, \tau) = \bar{g}_{\mu\nu}(\tau) + \delta g_{\mu\nu}(\mathbf{x}, \tau), \tag{5}$$

$$\phi(\mathbf{x}, \tau) = \bar{\phi}(\tau) + \delta\phi_{\mu\nu}(\mathbf{x}, \tau), \tag{6}$$

In the above equations in both cases, the first terms represents only the conformal time $\tau$ (Conformal time and physical times are related by the following expression:

$$\tau = \int \frac{dt}{a(t)}. \tag{7}$$

Once we substitute the explicit physical time-dependent scale factor $a(t)$ in FLRW space-time then one can explicitly compute the connecting relationship between the conformal time and physical time coordinates. In the present context of the discussion, we are specifically interested in two special types of solutions of the scale factor in FLRW geometric background, which can able to capture the information of quantum extremal islands, along with having tradition with two different signatures of cosmological constants (positive as well as negative) dependent background metric and the fields, respectively, which preserve the mentioned homogeneity and isotropy in the FLRW space-time. On the other hand, the second terms in the above equations are the outcome of perturbations in the FLRW background which capture the effects of inhomogeneity and anisotropy. Now, as we do not know how to treat a full quantum theory of gravity and how to deal with the gravitational fluctuations at a quantum level, even for FLRW background geometry the usual approach is to treat them classically. However, since we know how to quantize the inhomogeneous perturbed field, we treat it at the quantum level. So here once again we are using the semi-classical treatment to write down the quantum field theory in FLRW space-time from the perturbed contributions but it is implemented in a little bit different way to break the homogeneity and isotropy, which a general gravitational background might not in principle always have. Now, having this setup, one can either follow the semi-classical path integral approach by writing down the partition function or the associated quantum effective action and can compute the quantities that we evaluated in this paper within the context of time-evolving black hole geometry describing quantum extremal islands in presence of radiation in FLRW background. The other approach is to treat the problem by quantizing the Hamiltonian in terms of creation and annihilation operators using the well-known canonical quantization technique in an appropriate gravitational gauge. As we proceed further with the material of this paper, one can clearly visualize that we also have chosen a preferred gravitational gauge which helps us to directly connect the scalar part of the gravitational isotropy and homogeneity breaking fluctuation with the field fluctuation. Consequently one can translate the Hamiltonian and its quantized version in terms of gauge-invariant perturbations, which is commonly used in the context of perturbation theory in FLRW space-time, or commonly known as the cosmological perturbation theory [39–44]. Apart from using the usual canonical quantization technique, in the present context of the discussion, we have used the single field squeezed state formalism which helps us to think of the quantized Hamiltonian in terms of a mode having momentum **k** and other having momentum $-\mathbf{k}$ in the Fourier space. Additionally, using this formalism the quantized version of the Hamiltonian of the problem written in the present set up can be parametrized in terms two parameters, which quantifies the amplitude and phase of the two mode squeezed states. In the corresponding literature it is identified as the squeezing amplitude $r_{\mathbf{k}}(\tau)$ and the squeezing angle $\phi_{\mathbf{k}}(\tau)$. For the two given expressions for a scale factors $a(\tau)$ describing two different physical solutions of FLRW space-time describing the time evolving black hole geometry it is possible to compute the conformal time evolutions of these two parameters. Once this is done the rest of the problem can be automatically solved as most of the physical observables can be expressed in terms of these two time evolving parameters. This approach that we have followed in this paper actually expresses a very complicated underlying semi-classical computation from the quantum field theoretic set up in a very simplified language which helps us to extract all physical information from the computation. Not only this approach helps to simplify the complicated structure of the set up, but also helps to compute and physically interpret many quantum-mechanical observables, circuit complexity function, entanglement entropy of the black hole with or without having an island, and last but not the least also helps us to predict the features of four-point OTOC in a very simplest language.

The main motivations behind the current work are as follows:

- **Motivation-I**
  Instead of using the semi-classical approach to write down the quantum effective

action of the theory, express the entire set-up in terms of single field two mode squeezed state formalism [45,46];

- **Motivation-II**
  To understand signatures of quantum chaos in FRW space-time in the presence and absence of cosmological islands by using the quantum information theoretic measure known as circuit complexity. Circuit complexity, which is much more computationally easier compared to other probes, gives much more information about the underlying system;

- **Motivation-III**
  To comment about another probe of quantum chaos, i.e., OTOCs without directly computing it but by establishing a closed relation with the circuit complexity. Computing the OTOCs in this set-up is not a trivial task and is much more challenging, but it can be very easily predicted by computing the circuit complexity;

- **Motivation-IV**
  To study the dependence of circuit complexity on the parameters of the theory, thereby establishing the fact that the behavior of circuit complexity is not universal throughout the entire regime of the parameter space. The behavior of circuit complexity may not unique in the entire parameter space of the model under consideration. Thus, circuit complexity provides a way of probing the behavior of the model in the entire regime of the parameter space;

- **Motivation-V**
  To try to provide an alternative way of calculating entanglement entropy without going into the gravitational details of the model but from the perspective of circuit complexity which is much more easier to calculate. Circuit complexity also provides much more information than entanglement entropy, which in itself is a great motivation for computing it for any gravitational or field theory model.

The organization of the rest of the paper is as follows:

- In Section 2, we provide a brief review of the long-standing black hole information loss problem and various proposals given till date to solve it, focusing mainly on the Island proposal as our prime motivation to study the cosmological extension of this concept;
- In Section 3, we introduce the concept of circuit complexity to the readers to motivate them about this computational tool's role in probing various unexplored theoretical framework related to quantum chaos and information theory related issues;
- We then provide the useful cosmological FLRW models with radiation and AdS and dS space-time that we have considered in this paper to study chaos and complexity in Section 4 of this paper;
- Following that in Section 5, we provide the analytical expressions of circuit complexity calculated from two different cost functionals commonly used in the perspective of cosmological perturbation theory written in the language of squeezed quantum states;
- In Section 6, We provide the expressions of the von-Neumann entanglement entropy, Renyi entropy, and equilibrium temperature of the modes in terms of the squeezed state parameters;
- Section 7 contains our numerical calculation of the cosmological version of the circuit complexity and estimation of quantities like the measure of quantum chaos, i.e., quantum Lyapunov exponent;
- Finally, in Section 8, we conclude with our all findings in this paper with some interesting future prospects of the present work.

## 2. A Brief Review on Islands Paradigm

Black holes are thought to hold the key for quantum aspects of the gravitational paradigm. It has kept everyone puzzled ever since its appearance in Einstein's theory of general relativity. One of the biggest conundrums in quantum mechanics of black hole physics, for half a century or so, has been the problem of unitarity or, in other words,

the black hole information loss problem. Over the years, many proposals have tried to solve this problem *viz.* black hole complementarity principle [47,48], fuzzball paradigm [49–56], and firewall paradigm [57,58].

### 2.1. Pedagogical Details

In 2019, many renowned physicists came up with the path-breaking concept of tackling the black hole information loss problem by introducing *Islands* [2,3,5,6,59–64]. They proposed that the contribution to the black hole entropy came not only from inside the horizon (bulk) region but also from the outside horizon region of the evaporating black hole. Considering the generalized quantum entanglement entropy that includes the contribution from outside the black hole, the unitarity issue can be solved perfectly and consequently; the black hole information loss paradox can be resolved.

It is important to mention here that, without the inclusion of the island, the associated entropy of the Hawking radiation increases monotonically with respect to the evolutionary time scale. On the other hand, quantum mechanics demands that the final state of a black hole should be a pure state, and the entanglement entropy should go to zero after a long time. The island proposal provides a quantum correction in the result for the Bekenstein Hawking entropy, for an evaporating black hole. By considering the existence of the island region, it has been shown that after the page time, the island corrected entropy starts to decrease to reach zero after a long time, which means there is no entanglement at the very late time. Hence, the fundamental notion of unitarity of the black hole information is restored, as shown in Figure 1.

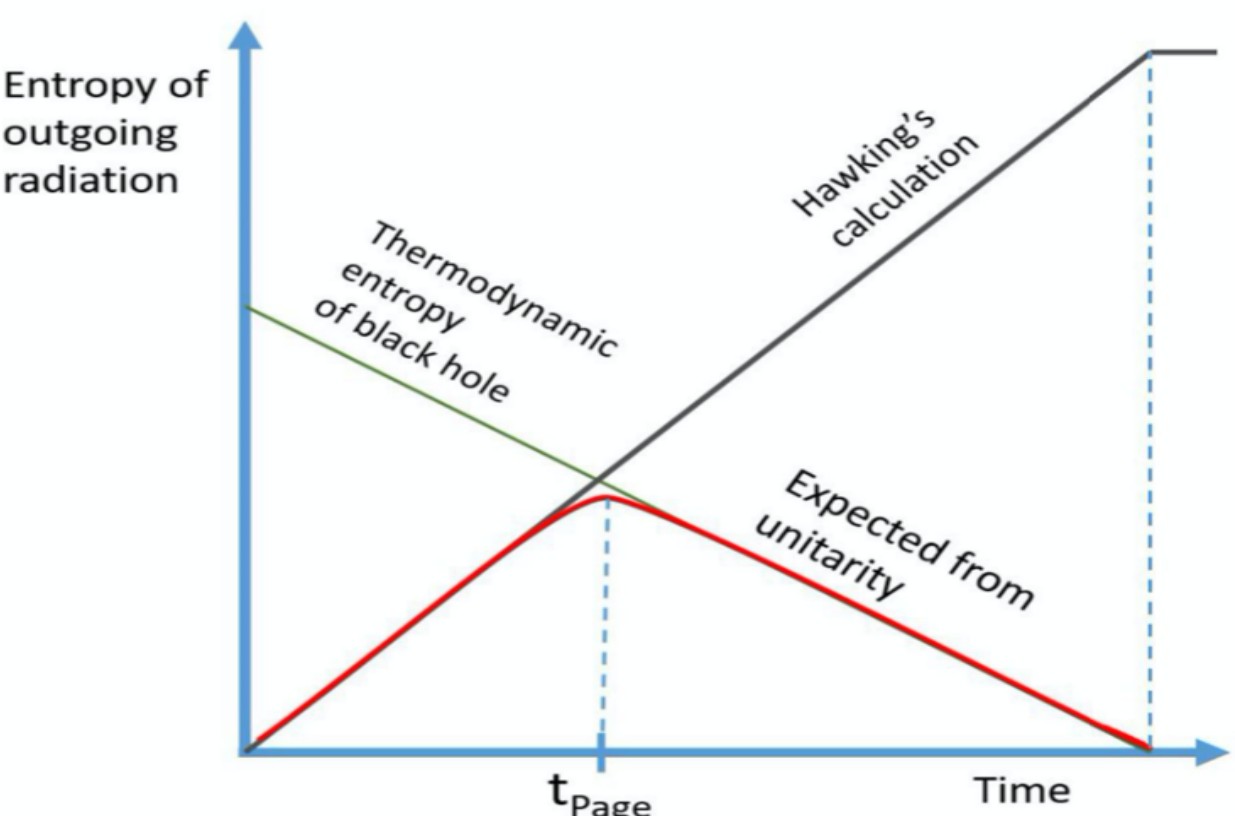

**Figure 1.** Schematic diagram showing the entropy of the outgoing radiation of the evaporating black hole as a function of physical evolutionary time scale. This schematic diagram was taken from [65].

### 2.2. Technical Details

Island formula generalizes the well known Ryu-Takayanagi formula used to quantify the holographic entanglement entropy [13–18,66–68]. Let us consider, $R$ to be a non-gravitational system which is entangled with a gravitational system. Then von Neumann entropy of the region $R$ is given by the following island formula:

$$\textbf{Island formula}: \quad S(R) = \min \text{ext}_\text{I} \, S_{gen}(I \cup R) \,, \tag{8}$$

where the generalized entropy is given by the following expression (Here in the first term, we have fixed $G_N = 1$ and $\hbar = 1$ in the natural unit system):

$$\textbf{Generalized entropy}: \quad S_{gen}(I \cup R) = \frac{\text{Area}(\partial I)}{4} + S_m(I \cup R) - S_{div}(\partial I) \,. \tag{9}$$

Here $I$ is the region in the gravitational system known as island, $S_m(I \cup R)$ is the von Neumann entropy of the region $(I \cup R)$ and $S_{div}(\partial I)$ is the UV divergent entropy of boundary of I, i.e., $\partial I$. Region $I$ is entangled with region $R$, i.e., degrees of freedom of $I$ are encoded in $R$. It means that the operators in $I$ can be rewritten as operators in $R$ no matter how complicated the form is.

Initial work on the Island prescription was done in the context of two dimensional dilaton gravity [5,59,69–76]. The representative action for this system is given as follows:

$$\textbf{Action in 1+1D}: \quad S = \frac{1}{2} \int d^2x \sqrt{-g} \, \phi \left\{ R + K(\phi)(\partial \phi)^2 - 2V(\phi) \right\}, \tag{10}$$

where the dilaton dependent coupling $K(\phi)$ and the representative potential $V(\phi)$ are defined by the following expressions:

$$\textbf{Jackiw} - \textbf{Teitelboim (JT) gravity}:$$
$$K(\phi) = 0, \qquad V(\phi) = -\lambda^2, \tag{11}$$

$$\textbf{Callan} - \textbf{Giddings} - \textbf{Harvey} - \textbf{Strominger (CGHS) gravity}:$$
$$K(\phi) = \frac{1}{\phi^2}, \qquad V(\phi) = -2\lambda^2. \tag{12}$$

Action given in Equation (10) is an example of modified measure theory as discussed in refs. [77,78]. Recently in refs. [79,80], the authors has shown the existence of islands in higher dimensional space-time as well. Furthermore, in References [81–83], Professor Robert C. Myers and co. also have studied quantum extremal Islands in arbitrary space-time dimensions. Such islands can exist even without the presence of black holes. In this $D + 1$ dimensional space-time the representative island action can be written as:

$$\textbf{Action in D+1}: \quad S = \frac{1}{2} \int d^{D+1}x \sqrt{-g} \, \phi \left\{ R + K(\phi)(\partial \phi)^2 - 2V(\phi) \right\}. \tag{13}$$

In this paper, we consider $3 + 1$ dimensional space-time model to study the circuit complexity in FLRW islands to have realistic cosmological implications since our observed universe is $3 + 1$ dimensional. Recently, Hartman and collaborators in Reference [29] has studied the presence of such islands in $3 + 1$ dimensional FLRW space-time.

In $3 + 1$ dimensional FLRW cosmological space-time the representative island action can be expressed as:

$$\textbf{Action in 3+1 D (Jordan)}: \quad S_J = \frac{1}{2} \int d\tau \, d^3x \, a^4(\tau) \, \phi \left\{ R + K(\phi)(\partial \phi)^2 - 2V(\phi) \right\}. \tag{14}$$

Now, to extract the underlying the physical insight from the above mentioned model, we need to perform conformal transformation on the above mentioned action. This allows

us to express the Island action written in Jordan frame to the Einstein frame, which is given by:

$$\textbf{Action in 3+1 D (Einstein)}: \quad S_E = \frac{1}{2} \int d\tau \, d^3x \, a^4(\tau) \left\{ \tilde{R} + (\partial\tilde{\phi})^2 - 2V(\tilde{\phi}) \right\}. \quad (15)$$

where in the newly defined Einstein frame the redefined field can be expressed in terms of the old frame field content as:

$$\textbf{Field in Jordan frame}: \quad \tilde{\phi} = \int \frac{\sqrt{(2K(\phi)+3)}}{\phi} \, d\phi \quad, \quad (16)$$

where we have throughout used the following conformal transformation in the metric:

$$\textbf{Einstein to Jordan frame}: \quad \tilde{g}_{\mu\nu} = \Omega^2(\phi) \, g_{\mu\nu} \quad \text{where} \quad \Omega^2(\phi) := \phi \ . \quad (17)$$

The above mentioned transformed action is in perfect form to analyze the cosmological perturbation theory, as it is representing the simple Einstein gravity minimally coupled to a canonical scalar field in the Einstein frame. This will further help us to study the cosmological imprints from the above mentioned island action expressed in FLRW spatially flat ($k=0$) background space-time. To avoid any confusion here it is important to note that, for further computational purpose we will drop the ˜ symbol from our analysis which actually distinguish the Einstein and Jordan frame explicitly.

The conditions for such Islands to appear in any gravitational space-time and quantum state are already discussed in greater detail in [29]. Below we summarize the conditions.

- **Condition I:**
  Bekenstein area bound for entropy must be violated in the following way within the island prescription:

$$\tilde{S}_m \geq \frac{Area(\partial I)}{4}, \quad (18)$$

  Here $\tilde{S}_m$ is the finite matter entropy after subtracting the UV divergences appearing at the boundary;

- **Condition II:**
  The region $I$ can be treated as the quantum normal if the following criteria holds good for the generalized entropy:

$$\pm \frac{d}{d\lambda_\pm} S_{gen}(I) \geq 0, \quad (19)$$

  where $\frac{d}{d\lambda_\pm}$ is the null derivative (+ for outward, − for inward) with respect to Island region $I$;

- **Condition III:**
  The region $G$ can be treated as the quantum normal if the following criteria holds good for the generalized entropy:

$$\mp \frac{d}{d\lambda_\pm} S_{gen}(G) \leq 0, \quad (20)$$

  where $\frac{d}{d\lambda_\pm}$ is the null derivative (+ for outward, − for inward) with respect to Island region $I$.

Usually, calculating entanglement entropy is not a very easy task. We want to bypass this cumbersome task in a comparatively easy way by considering the use of circuit complexity within the framework of FLRW cosmology. We do not do any computation from the gravitational theories to calculate the entanglement entropy in the presence of quantum extremal Islands. Instead of doing a complicated computation, we consider the

FLRW metric with radiation along with the negative (AdS) and positive (dS) signature of the cosmological constant as our initial ingredient for the present computational purpose. To understand the effects of quantum mechanics, we must perturb the usual classical FLRW metric in $3 + 1$ dimensional space-time, out of which we will consider only the scalar modes in this paper. In this paper, we study cosmological circuit complexity from squeezed state formalism for both positive (dS) and negative (AdS) cosmological constant with radiation and re-examine the island paradigm in both cases from a cosmology-complexity connecting point of view. We will explicitly show that cosmological circuit complexity for the AdS case will be precisely consistent with the island paradigm, which can produce the Page curve and resolve the well known black hole information loss paradox without exactly computing any gravitational entropy.

### 3. Circuit Complexity and Its Purposes

Complexity as a measure was first suggested by Susskind et al. [84,85] and a series of other papers to explain the increasing size of an ER bridge inside an eternal black hole that connects two copies of the dual CFT. To describe the interior, we need a measure of information that evolves for much longer times after the boundary CFT has reached thermal equilibrium [84,86]. One can think of complexity as the information about the processes in a system. A system with more processes (thermodynamic, mechanical, or energy dissipation) can be considered to be of higher complexity than a system with lesser processes happening. A definition of circuit complexity can then be extended as the minimum number of processes that follow a path along a circuit between a reference state and a target state. In the language of quantum mechanics, we can look at processes as unitary gates and the reference and target quantum mechanical states can be chosen based on our model. In [87], Nielsen had shown the minimization of unitary gates of a circuit between reference and target state is the minimization of geodesic length in circuit space. One can start with a simple evolution of a reference state into the target state with some unitary transformation $U$.

$$\textbf{Target state from initial reference state}: \quad |\Psi_T\rangle = U |\Psi_R\rangle , \tag{21}$$

where the representative unitary operation for the circuit complexity can be represented by $n$ consecutive operations, as given by:

$$\textbf{Unitary operator}: \quad U = \prod_{\alpha=1}^{n} g_{i_\alpha} . \tag{22}$$

There is no unique choice in determining the circuit. We can work directly with the wavefunctions [88] using Nielsen's geometric approach, or we can use an alternative definition through Fubini-study metric [89]. We adapt the approach given in [88] where they work on a lattice of infinite harmonic oscillators and constructing the desired $U$ using path ordered exponential of Hamiltonian in the space of circuits parametrized by $s$.

$$\textbf{Unitary as path ordering}: \quad U(s) = \overleftarrow{\mathcal{P}} \exp\left(-i \int_0^s ds'\, H(s')\right) . \tag{23}$$

$\overleftarrow{\mathcal{P}}$ indicates a path ordering such that the Hamiltonian at the earlier times is applied to the first state. $U(s)$ represents a family of unitaries ranging from $U(s = 0)$—the identity matrix to $U(s = 1)$ which is the final unitary satisfying Equation (21). The Hamiltonian $H(s)$ can be expanded in terms of generalized Pauli matrices as,

$$\textbf{Hamiltonian}: \quad H(s) = \sum_I Y^I(s) M_I , \tag{24}$$

where $M_I$ represents the generalized Pauli matrices, and the coefficients $Y^I(s)$ are control functions that tell which gate to act at a particular value of $s$. These functions specify tangent to the trajectory in the space of unitaries and hence solve the Schrödinger equation

$$\textbf{Path evolution of unitary operator}: \quad \frac{dU(s)}{ds} = -iY(s)^I M_I U(s) . \tag{25}$$

The idea then is to define a cost functional for all possible paths, and minimizing this will give us the optimal circuit.

$$\textbf{Definition of cost functional}: \quad \mathcal{D}(U(s)) = \int_0^1 dt \, F(U(s), Y^I(s)) , \tag{26}$$

where $F$ is a local cost functional depending on the position $U(s)$ and velocity $Y^I(s)$. This problem is now similar to minimizing action on a given Lagrangian—$F(U(s), Y^I(s))$. The cost function needs to satisfy certain properties for it to be physically reasonable. He identified the problem of finding the optimal circuit with the problem of finding extremal curves or geodesics in a *Finsler geometry*, with the cost functional acting as the *Finsler* metric. Complexity is then identified with the length of the geodesic.

Among the many known, the most studied cost functionals are the linearly weighted and the geodesically weighted cost functional and are defined as

$$\textbf{Linear cost functional}: \quad F_1 := \sum_I |Y^I(s)|, \tag{27}$$

$$\textbf{Quadratic cost functional}: \quad F_2 := \sqrt{\sum_I (Y^I(s))^2}, \tag{28}$$

The purpose of computing complexity from two different types of cost functional will enable us to comment on which cost functional is better for probing the underlying quantum chaotic features of a system.

The final step will be to get explicit expressions for the velocity. In our particular case working with squeezed vacuum state and unsqueezed vacuum state Section 5, we can write the wave functions as Gaussian $e^{-x^2}$. By suitably diagonalizing we can represent the wave function in a general form:

$$\textbf{General Wave Fuction}: \quad \psi \approx \exp\left[-\frac{1}{2} x_a A_{ab} x_b\right]. \tag{29}$$

Our reference and target states can then be specified in terms of the positive symmetric matrices $A$. One can then find an expression of gates in their matrix forms that act on $A$ to produce the target state from the reference state. These gate action is defined by a set of generators $M_I$.

$$\textbf{Gate Action}: \quad Q_{ab} = \exp[\epsilon M_{ab}], \tag{30}$$

where $M_I$ are suitable generators decided by the Hamiltonian of the model. and $\epsilon$ is a parameter. We can then find explicit expressions for the velocity $Y^I(s)$ by rewriting Equation (25) in the following form:

$$Y^I(s) M_I = i(\partial_s U(s)) U^{-1}(s). \tag{31}$$

Often the basis generators are simple enough to produce simple inner products seen in many cases [88,90,91] given by

$$\textbf{Inner Product}: \quad \text{Tr}(M_I M_J^T) \approx \delta_{IJ}. \tag{32}$$

With this one can get a straightforward equation of velocity.

$$\textbf{Velocity}: \quad Y^I(s) = \text{Tr}(i(\partial_s U(s))U^{-1}(s)M_I^T). \tag{33}$$

Using this, we can write the velocity in terms of the unitary operation that can be identified from the diagonalized representation of the reference and target state derived from the model's Hamiltonian. We have done this using squeezed state formalism in cosmological perturbations in the following sections for our given model.

To give some motivation about the necessity of complexity in theoretical physics, we review here some of the important aspects of complexity that has been used by many people in this field:

- **Motivation I:**
  The motivation to study circuit complexity in high energy physics arose when it was applied to quantum field theory and gravity sector [88,91–117], particularly from attempts to apply AdS/CFT duality in certain black hole settings. Susskind et al. in Reference [19] proposed ways of probing the interior regions of the black hole horizon. They showed that these probes can be somehow related to a quantum information-theoretic measure, namely, "*Complexity*". Two famous conjectures came into the picture, which opened many new areas of research in the branch of theoretical physics connecting condensed matter and high energy physics with quantum information science being the heart. The two conjectures are famously known as the "*Complexity = Volume*" and "*Complexity = Action*" [1,84,85,118];

- **Motivation II:**
  Apart from its use in the gravity sector, the notion of circuit complexity has found its application in various other areas. Having a close relationship with the *out of time-ordered correlation functions* (OTOC) circuit complexity has recently been used as a diagnostic of quantum chaos and randomness [119,120]. Complexity has been found to provide many important details that are of utmost significance when one speaks about a chaotic system. It can be used to predict the *Lyapunov exponent* [121], scrambling time [122], equilibrium temperature, and many other important properties of a chaotic system. Additionally, in the non-chaotic regime where one cannot connect the circuit complexity function with OTOC through a simple relationship, the present analysis acts a significant theoretical probe to study the underlying various unknown physical properties of the system under consideration. We will show later that instead of getting exponential growth, in the non-chaotic regime which can be studied with very tiny values of the cosmological constant values we get decreasing behavior;

- **Motivation III:**
  Recently people have tried to study and quantify chaos in different cosmological frameworks using the notion of circuit complexity and OTOCs [20–23,123]. By following the same research trend in this article we have studied the same issue for the given model in detail. Though we have not restricted ourself to study only the chaotic features, but also we have explored the other parameter space (tiny value of the cosmological constants) where all the non-chaotic decreasing feature in the circuit complexity function, as well as the Island entropy function can be observed with respect the two possible solutions of the dynamical scale factors obtained for spatially flat FLRW cosmological background in presence of radiation and two possible signatures of cosmological constant.

## 4. Cosmological Models for Islands

In this section, we briefly discuss the cosmological models that we are considering in this paper by following Reference [29]. We consider the solution of FLRW cosmology with radiation along with the negative (AdS) and positive (dS) cosmological constant.

The technical details of these

### 4.1. Model-I

In this case, the corresponding Friedman equation in presence of AdS FLRW space-time, with SO(2, 3) isometry, along with the radiation in the spatially flat ($k = 0$) universe can be written as (see Figure 2):

$$\textbf{AdS FLRW + Radiation}: \quad H^2(t) = \left(\frac{d \ln a(t)}{dt}\right)^2 = \left(\frac{\dot{a}(t)}{a(t)}\right)^2 = \frac{8\pi}{3}\left(\frac{\epsilon_0}{a^4(t)} - \frac{|\Lambda|}{8\pi}\right). \quad (34)$$

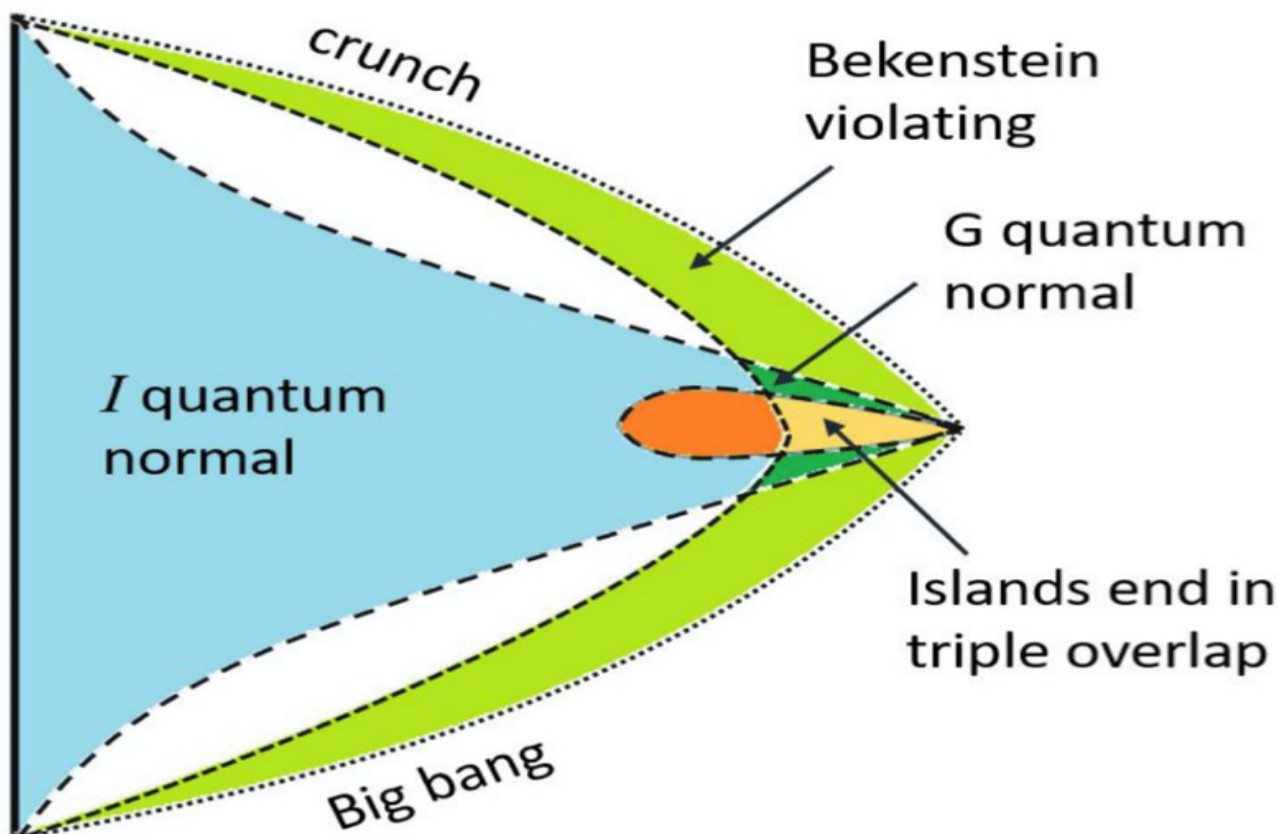

**Figure 2.** Representative Penrose diagram of recollapsing FRW cosmology with radiation and negative cosmological constant showing presence of islands. This diagram has been taken from [29].

The scale factor obtained by solving the above form of Friedman equation for FLRW cosmology with radiation and negative cc is given as follows:

$$\textbf{Scale factor for AdS FLRW + Radiation}: \quad a(t) = a_0 \sqrt{\cos\frac{\pi t}{2t_m}}, \quad (35)$$

where the symbols $a_0$ and $t_m$ are described by the following expressions:

$$a_0 = a(t = 0) = \left(\frac{8\pi\epsilon_0}{|\Lambda|}\right)^{1/4}, \quad t_m = \frac{\pi}{4}\sqrt{\frac{3}{|\Lambda|}} \quad (36)$$

It the context of cosmology literature, people generally use conformal time instead of physical time. Hence, it is useful to convert the scale factors in conformal time which is related to the physical time by the following relation

$$d\tau = \frac{dt}{a(t)}$$

The scale factors of this model considered in terms of the conformal time coordinates is given by the following expression:

$$a(\tau) = a_0 \sqrt{\cos\left[2\text{JacobiAmplitude}\left[\frac{a_0 \pi \tau}{4 t_m}, 2\right]\right]} \quad . \tag{37}$$

*4.2. Model-II*

In this case, the corresponding Friedman equation in presence of dS FLRW space-time with SO(1, 4) isometry along with the radiation in the spatially flat ($k = 0$) universe can be written as (see Figure 3):

$$\textbf{dS FLRW} + \textbf{Radiation}: \quad H^2(t) = \left(\frac{d \ln a(t)}{dt}\right)^2 = \left(\frac{\dot{a}(t)}{a(t)}\right)^2 = \frac{8\pi}{3}\left(\frac{\epsilon_0}{a^4(t)} + \frac{|\Lambda|}{8\pi}\right). \tag{38}$$

The scale factor obtained by solving the above form of Friedman equation for FLRW cosmology with radiation and positive cc is given as follows:

$$\textbf{Scale factor for dS FLRW} + \textbf{Radiation}: \quad a(t) = a_0 \sqrt{\sinh\frac{\pi t}{2 t_m}}, \tag{39}$$

where the symbols $a_0$ and $t_m$ are described by the following expressions:

$$a_0 = a(t = 0) = \left(\frac{8\pi\epsilon_0}{\Lambda}\right)^{1/4}, \quad t_m = \frac{\pi}{4}\sqrt{\frac{3}{\Lambda}} \tag{40}$$

Using further the notion of conformal time coordinate, the scale factors of this model considered in terms of the conformal time coordinates is given by the following expression:

$$a(\tau) = a_0 \sqrt{-i \cos\left[2\text{JacobiAmplitude}\left[\frac{1}{4}\left(-\frac{(1+i)a_0 \pi \tau}{\sqrt{2} t_m} + \frac{(2+2i)\sqrt{\frac{1}{t_m}}\,\text{EllipticK}(1/2)}{\frac{i}{t_m}}\right), 2\right]\right]}. \tag{41}$$

See Appendix A for more details.

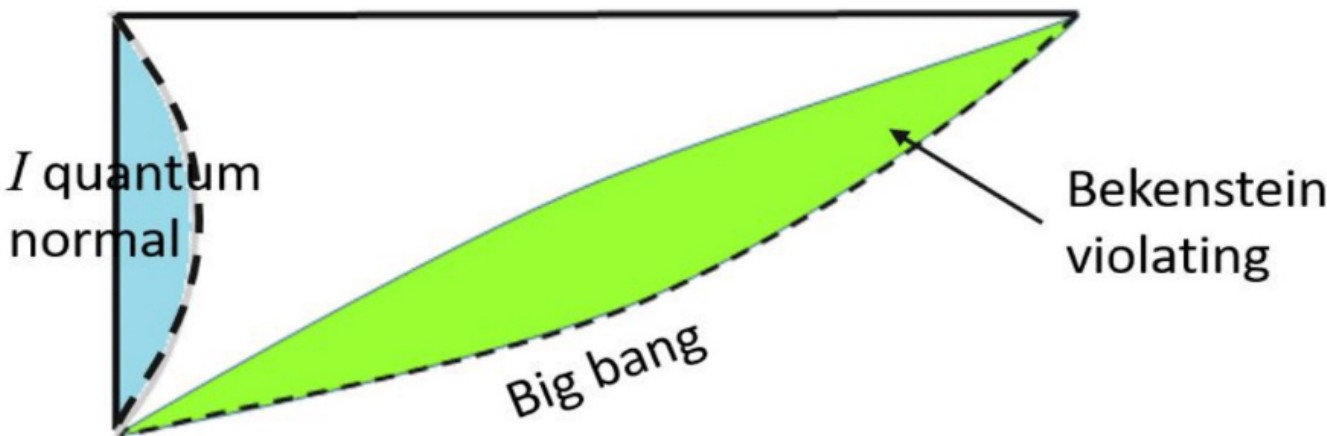

**Figure 3.** Penrose diagram showing regions of FRW cosmology with radiation and positive cosmological constant. It shows that the Bekenstein violating region does not overlap with the quantum normal region. Hence it does not contain any islands. This diagram has been taken from [29].

### 5. Quantum Complexity from Squeezed Quantum States

*5.1. Squeezed States from Perturbation FLRW Cosmology*

In this section, we will study squeezed state formalism within the framework of cosmological perturbation theory for FLRW spatially flat background. As already discussed in the earlier section, we consider the Island action in the 3 + 1 dimension in the Einstein frame given by

$$S_E = \frac{1}{2} \int d\tau \, d^3x \, a^4(\tau) \left\{ R + (\partial\phi)^2 - 2V(\phi) \right\} \tag{42}$$

We again remind the reader that in the above equation, we have deliberately removed the tilde sign from the field variable for the sake of notational simplicity. It is to be noted that the field variable $\phi$ in the above equation represents the redefined field in the Einstein frame. We now consider the following perturbation in the scalar field:

$$\phi(\mathbf{x}, \tau) = \phi(\tau) + \delta\phi(\mathbf{x}, \tau) \tag{43}$$

and the whole dynamics can be expressed in terms of a gauge invariant description through a variable given by:

$$\zeta(\mathbf{x}, \tau) = -\frac{\mathcal{H}(\tau)}{\left(\dfrac{d\phi(\tau)}{d\tau}\right)} \delta\phi(\mathbf{x}, t). \tag{44}$$

We fix some gauge constraints that re-parametrizes space-time for the first order perturbation theory:

$$\delta\phi(\mathbf{x}, \tau) = 0, \;\; g_{ij}(\mathbf{x}, \tau) = a^2(\tau)\left[(1 + 2\zeta(\mathbf{x}, \tau))\delta_{ij} + h_{ij}(\mathbf{x}, \tau)\right], \;\; \partial_i h_{ij}(\mathbf{x}, \tau) = 0 = h_i^i(\mathbf{x}, \tau), \tag{45}$$

This gauge, conserves the curvature perturbation variable is outside the horizon.

We apply ADM formalism to compute the second-order perturbed action for scalar modes. The action, after gauge fixing is:

$$\delta^{(2)}S = \frac{1}{2} \int d\tau \, d^3\mathbf{x} \, \frac{a^2(\tau)}{\mathcal{H}^2} \left(\frac{d\phi(\tau)}{d\tau}\right)^2 \left[(\partial_\tau \zeta(\mathbf{x}, \tau))^2 - (\partial_i \zeta(\mathbf{x}, \tau))^2\right]. \tag{46}$$

To re-parametrize the second-order perturbed action, we introduce the following space-time dependent variable:

$$v(\mathbf{x}, \tau) = z(\tau)\, \zeta(\mathbf{x}, \tau), \quad \text{where } z(\tau) = a(\tau)\sqrt{\epsilon(\tau)}, \tag{47}$$

which transforms the perturbed action to a familiar form of canonical scalar field. This is known as the *Mukhanov variable*. Additionally, note that the newly defined quantity, $\epsilon(\tau)$ is the conformal time dependent slow-varying parameter:

$$\epsilon(\tau) := -\frac{\dot{H}}{H^2} = -\frac{a(\tau)}{\mathcal{H}^2}\frac{d}{d\tau}\left(\frac{\mathcal{H}}{a(\tau)}\right) = 1 - \frac{\mathcal{H}'}{\mathcal{H}^2}. \tag{48}$$

Consequently, second order perturbed action for the scalar perturbation in terms of the *Mukhanov variable* can be written:

$$\delta^{(2)}S = \frac{1}{2}\int d\tau\, d^3\mathbf{x}\left[v'^2(\mathbf{x}, \tau) - (\partial_i v(\mathbf{x}, \tau))^2 + \left(\frac{z'(\tau)}{z(\tau)}\right)^2 v^2(\mathbf{x}, \tau) - 2\left(\frac{z'(\tau)}{z(\tau)}\right)v'(\mathbf{x}, \tau)v(\mathbf{x}, \tau)\right]. \tag{49}$$

The quantity $\frac{z'}{z}$ can be calculated as:

$$\frac{z'(\tau)}{z(\tau)} = \frac{a'(\tau)}{a(\tau)} + \frac{1}{2}\frac{\epsilon'(\tau)}{\epsilon(\tau)} = \mathcal{H}\left[\frac{1}{\epsilon(\tau)} - 1 + \epsilon(\tau) - \frac{1}{2}\frac{1}{\epsilon(\tau)}\frac{\mathcal{H}''}{\mathcal{H}^3}\right] \tag{50}$$

Using the following *ansatz* for the Fourier transformation we now convert the second order perturbed action for the scalar degrees of freedom in terms of the Fourier modes.

$$v(\mathbf{x}, \tau) := \int \frac{d^3\mathbf{k}}{(2\pi)^3} v_{\mathbf{k}}(\tau)\, \exp(-i\mathbf{k}.\mathbf{x}), \tag{51}$$

After substituting the above expression, the second-order perturbation for the scalar modes in Fourier space can recast as:

$$\delta^{(2)}S = \frac{1}{2}\int d\tau\, d^3\mathbf{k}\underbrace{\left[|v'_{\mathbf{k}}(\tau)|^2 + \left(k^2 + \left(\frac{z'(\tau)}{z(\tau)}\right)^2\right)|v_{\mathbf{k}}(\tau)|^2 - 2\left(\frac{z'(\tau)}{z(\tau)}\right)v'_{\mathbf{k}}(\tau)v_{-\mathbf{k}}(\tau)\right]}_{\text{Lagrangian density } \mathcal{L}^{(2)}(\mathbf{v}_{\mathbf{k}}(\mathbf{0}), \mathbf{v}'_{\mathbf{k}}(\mathbf{0}), \mathbf{0})}, \tag{52}$$

where it is important to note that:

$$|v'_{\mathbf{k}}(\tau)|^2 = v'^*_{-\mathbf{k}}(\tau)v'_{\mathbf{k}}(\tau), \quad |v_{\mathbf{k}}(\tau)|^2 = v^*_{-\mathbf{k}}(\tau)v_{\mathbf{k}}(\tau). \tag{53}$$

We vary the second-order perturbed action with respect to the perturbed field variable in the Fourier space, and we get:

$$v''_{\mathbf{k}}(\tau) + \omega^2(k, \tau)v_{\mathbf{k}}(\tau) = 0. \tag{54}$$

This is known as the *Mukhanov-Sasaki equation* and represents the classical equation of motion of a parametric oscillator where the frequency of the oscillator is conformal time dependent and in the present context of discussion, given by:

$$\omega^2(k, \tau) := k^2 + m_{\text{eff}}^2(\tau), \tag{55}$$

where we have introduced a conformal time dependent effective mass, quantified by:

$$m_{\text{eff}}^2(\tau) = -\frac{z''(\tau)}{z(\tau)} = \frac{1}{\tau^2}\left(\nu_{\text{island}}^2(\tau) - \frac{1}{4}\right) \tag{56}$$

The conformal time dependent mass parameter can be calculated for the two models considered in this paper as follows. In the Friedman equations the effective fluid in the presence of radiation and a negative (AdS) / positive (dS) cosmological constant are described by the following effective pressure and energy densities, which are given by:

**AdS FLRW + Radiation:**

$$p_{\text{eff}} = \left( p + \frac{|\Lambda|}{16\pi\epsilon_0} \right), \qquad (57)$$

$$\rho_{\text{eff}} = \left( \rho - \frac{|\Lambda|}{16\pi\epsilon_0} \right). \qquad (58)$$

**dS FLRW + Radiation:**

$$p_{\text{eff}} = \left( p - \frac{|\Lambda|}{16\pi\epsilon_0} \right), \qquad (59)$$

$$\rho_{\text{eff}} = \left( \rho + \frac{|\Lambda|}{16\pi\epsilon_0} \right). \qquad (60)$$

Further, we introduce a quantity called equation of state parameter for the effective fluid, $w_{\text{eff}}$, which is defined as follows:

**AdS FLRW + Radiation:**
$$w_{\text{eff}} = \frac{p_{\text{eff}}}{\rho_{\text{eff}}} = \left[ \frac{\left( p + \frac{|\Lambda|}{16\pi\epsilon_0} \right)}{\left( \rho - \frac{|\Lambda|}{16\pi\epsilon_0} \right)} \right], \qquad (61)$$

**dS FLRW + Radiation:**
$$w_{\text{eff}} = \frac{p_{\text{eff}}}{\rho_{\text{eff}}} = \left[ \frac{\left( p - \frac{|\Lambda|}{16\pi\epsilon_0} \right)}{\left( \rho + \frac{|\Lambda|}{16\pi\epsilon_0} \right)} \right]. \qquad (62)$$

Particularly for radiation dominated epoch the radiation pressure can be expressed in terms of the energy density as, $p = \frac{\rho}{3}$. Thus, the effective equation of state parameter can be further simplified as:

**AdS FLRW + Radiation:**
$$w_{\text{eff}} = \frac{1}{3} \left[ \frac{\left( 1 + \frac{3|\Lambda|}{16\pi\epsilon_0\rho_0} \right)}{\left( 1 - \frac{|\Lambda|}{16\pi\epsilon_0\rho_0} \right)} \right], \qquad (63)$$

**dS FLRW + Radiation:**
$$w_{\text{eff}} = \frac{1}{3} \left[ \frac{\left( 1 - \frac{3|\Lambda|}{16\pi\epsilon_0\rho_0} \right)}{\left( 1 + \frac{|\Lambda|}{16\pi\epsilon_0\rho_0} \right)} \right]. \qquad (64)$$

In the purely radiation dominated epoch the radiation density scales with the scale factor as, $\rho = \rho_0 a^{-4}$, using which the effective equation of state parameter $w_{\text{eff}}$ finally takes the following simplified form:

**AdS FLRW + Radiation:**
$$w_{\text{eff}} = \frac{1}{3}\left[\frac{\left(1 + 3\left(\frac{a}{a_0}\right)^4\right)}{\left(1 - \left(\frac{a}{a_0}\right)^4\right)}\right], \tag{65}$$

**dS FLRW + Radiation:**
$$w_{\text{eff}} = \frac{1}{3}\left[\frac{\left(1 - 3\left(\frac{a}{a_0}\right)^4\right)}{\left(1 + \left(\frac{a}{a_0}\right)^4\right)}\right]. \tag{66}$$

where in both the results we define the following quantity $a_0$ to be

$$a_0 = a(t = 0) = \left(\frac{16\pi\epsilon_0\rho_0}{|\Lambda|}\right)^{1/4} = \left(\frac{8\pi\epsilon_0}{|\Lambda|}\right)^{1/4}, \quad \text{where we fix} \quad \rho_0 = \frac{1}{2}. \tag{67}$$

Consequently, the general mass parameter for cosmological islands can be computed as:

$$\nu_{\text{island}} = \sqrt{\frac{1}{4} + \frac{2(1 - w_{\text{eff}})}{(1 + 3w_{\text{eff}})^2}}. \tag{68}$$

which can be further, explicitly has written for the mentioned two models as:

**AdS FLRW + Radiation:**
$$\nu_{\text{island}}(a) = \frac{1}{2}\sqrt{1 + \Delta_{\text{AdS}}(a)}, \tag{69}$$

**dS FLRW + Radiation:**
$$\nu_{\text{island}}(a) = \frac{1}{2}\sqrt{1 + \Delta_{\text{dS}}(a)}. \tag{70}$$

where the newly introduced scale factor dependent factors, $\Delta_{\text{AdS}}$ and $\Delta_{\text{dS}}$ are defined as follows:

$$\Delta_{\text{AdS}}(a) := \frac{8\left(1 - \frac{1}{3}\left[\frac{\left(1 + 3\left(\frac{a}{a_0}\right)^4\right)}{\left(1 - \left(\frac{a}{a_0}\right)^4\right)}\right]\right)}{\left(1 + \left[\frac{\left(1 + 3\left(\frac{a}{a_0}\right)^4\right)}{\left(1 - \left(\frac{a}{a_0}\right)^4\right)}\right]\right)^2}, \tag{71}$$

$$\Delta_{\text{dS}}(a) := \frac{8\left(1 - \frac{1}{3}\left[\frac{\left(1 - 3\left(\frac{a}{a_0}\right)^4\right)}{\left(1 + \left(\frac{a}{a_0}\right)^4\right)}\right]\right)}{\left(1 + \left[\frac{\left(1 - 3\left(\frac{a}{a_0}\right)^4\right)}{\left(1 + \left(\frac{a}{a_0}\right)^4\right)}\right]\right)^2}. \tag{72}$$

Finally, substituting all the above mentioned expressions for the mass parameters obtained for the two cases, we get the following simplified expressions:

**AdS FLRW + Radiation:**
$$m_{\text{eff}}^2(\tau) = \frac{1}{4\tau^2}\Delta_{\text{AdS}}(a) = \frac{2}{\tau^2}\frac{\left(1 - \frac{1}{3}\left[\frac{\left(1 + 3\left(\frac{a}{a_0}\right)^4\right)}{\left(1 - \left(\frac{a}{a_0}\right)^4\right)}\right]\right)}{\left(1 + \left[\frac{\left(1 + 3\left(\frac{a}{a_0}\right)^4\right)}{\left(1 - \left(\frac{a}{a_0}\right)^4\right)}\right]\right)^2}, \tag{73}$$

**dS FLRW + Radiation:**
$$m_{\text{eff}}^2(\tau) = \frac{1}{4\tau^2}\Delta_{\text{dS}}(a) = \frac{2}{\tau^2}\frac{\left(1 - \frac{1}{3}\left[\frac{\left(1 - 3\left(\frac{a}{a_0}\right)^4\right)}{\left(1 + \left(\frac{a}{a_0}\right)^4\right)}\right]\right)}{\left(1 + \left[\frac{\left(1 - 3\left(\frac{a}{a_0}\right)^4\right)}{\left(1 + \left(\frac{a}{a_0}\right)^4\right)}\right]\right)^2}. \tag{74}$$

In terms of the cosmological constant for both the cases, the above expression can be further recast as:

**AdS FLRW + Radiation:**
$$m_{\text{eff}}^2 = \frac{2}{\tau^2}\left[\frac{1 - \left(\frac{1 + \frac{3|\Lambda|}{8\pi\epsilon_0}}{1 - \frac{|\Lambda|}{8\pi\epsilon_0}}\right)}{1 + \left(\frac{1 + \frac{3|\Lambda|}{8\pi\epsilon_0}}{1 - \frac{|\Lambda|}{8\pi\epsilon_0}}\right)}\right], \tag{75}$$

**dS FLRW + Radiation:**
$$m_{\text{eff}}^2 = \frac{2}{\tau^2}\left[\frac{1 - \left(\frac{1 - \frac{3|\Lambda|}{8\pi\epsilon_0}}{1 + \frac{|\Lambda|}{8\pi\epsilon_0}}\right)}{1 + \left(\frac{1 - \frac{3|\Lambda|}{8\pi\epsilon_0}}{1 + \frac{|\Lambda|}{8\pi\epsilon_0}}\right)}\right]. \tag{76}$$

These obtained results for the mass parameter and the effective mass for the two cases are extremely useful for further analysis, which we will perform in the next section.

*5.2. Scalar Mode Function for Cosmological Islands*

The *Mukhanov-Sasaki equation* can be simplified into:

$$v_{\mathbf{k}}''(\tau) + \left(k^2 - \frac{1}{\tau^2}\left(\nu_{\text{island}}^2(\tau) - \frac{1}{4}\right)\right)v_{\mathbf{k}}(\tau) = 0. \tag{77}$$

The most general analytical solution is:

$$v_{\mathbf{k}}(\tau) := \sqrt{-\tau}\left[\mathcal{C}_1\,\mathcal{H}_{\nu_{\text{island}}}^{(1)}(-k\tau) + \mathcal{C}_2\,\mathcal{H}_{\nu_{\text{island}}}^{(2)}(-k\tau)\right] \tag{78}$$

where $\mathcal{H}^{(1)}_{\nu_{\text{island}}}(-k\tau)$ and $\mathcal{H}^{(2)}_{\nu_{\text{island}}}(-k\tau)$ are Hankel functions of the first and second kind, respectively, with argument $-k\tau$ and order $\nu_{\text{island}}$. $\mathcal{C}_1$ and $\mathcal{C}_2$ can be fixed by the choice of the initial vacuum state and we restrict ourselves, to *Bunch Davies vacuum* or *Hartle Hawking vacuum* or *Chernkov vacuum*, by choosing the integration constants as $\mathcal{C}_1 = 1$ and $\mathcal{C}_2 = 0$.

The solution then becomes:

$$v_{\mathbf{k}}(\tau) = \sqrt{-\tau}\, \mathcal{H}^{(1)}_{\nu_{\text{island}}}(-k\tau). \tag{79}$$

Upon further considering the asymptotic limits, $-k\tau \to 0$ and $-k\tau \to \infty$, the Hankel functions of the first kind are simplified into:

$$\lim_{-k\tau \to \infty} \mathcal{H}^{(1)}_{\nu_{\text{island}}}(-k\tau) = \sqrt{\frac{2}{\pi}}\frac{1}{\sqrt{-k\tau}}\exp\left(-i\left\{k\tau + \frac{\pi}{2}\left(\nu_{\text{island}} + \frac{1}{2}\right)\right\}\right). \tag{80}$$

Using these asymptotic results of the Hankel functions can be expressed as:

$$v_{\mathbf{k}}(\tau) = \frac{2^{\nu_{\text{island}}-\frac{3}{2}}(-k\tau)^{\frac{3}{2}-\nu_{\text{island}}}}{\sqrt{2k}}\left|\frac{\Gamma(\nu_{\text{island}})}{\Gamma(\frac{3}{2})}\right|\left(1-\frac{i}{k\tau}\right)\exp\left(-i\left\{k\tau + \frac{\pi}{2}\left(\nu_{\text{island}}-\frac{3}{2}\right)\right\}\right). \tag{81}$$

*5.3. Quantization of Hamiltonian for Scalar Modes*

Further, we derive the conformal time derivative of the field variable:

$$v'_{\mathbf{k}}(\tau) = i\sqrt{\frac{k}{2}}\,2^{\nu_{\text{island}}-\frac{3}{2}}(-k\tau)^{\frac{3}{2}-\nu_{\text{island}}}\left|\frac{\Gamma(\nu_{\text{island}})}{\Gamma(\frac{3}{2})}\right|\left\{1-\left(\nu_{\text{island}}-\frac{1}{2}\right)\frac{i}{k\tau}\left(1-\frac{i}{k\tau}\right)\right\}$$
$$\exp\left(-i\left\{k\tau + \frac{\pi}{2}\left(\nu_{\text{island}}-\frac{1}{2}\right)\right\}\right). \tag{82}$$

To construct the classical Hamiltonian function, one needs the canonically conjugate momentum associated with the classical cosmologically perturbed scalar field variable and can be calculated as:

$$\pi_{\mathbf{k}}(\tau) := \frac{\partial \mathcal{L}^{(2)}(v_{\mathbf{k}}(\tau), v'_{\mathbf{k}}(\tau), \tau)}{\partial v'_{\mathbf{k}}(\tau)} = v'^{*}_{\mathbf{k}}(\tau) - \left(\frac{z'(\tau)}{z(\tau)}\right)v_{\mathbf{k}}(\tau) \tag{83}$$

The classical Hamiltonian in the present context turns out to be:

$$H(\tau) = \int d^3\mathbf{k}\left[\frac{1}{2}\left|\pi_{\mathbf{k}}(\tau) + \frac{z'(\tau)}{z(\tau)}v_{\mathbf{k}}(\tau)\right|^2 + \frac{1}{2}\mu^2(k,\tau)|v_{\mathbf{k}}(\tau)|^2\right], \tag{84}$$

where the time dependent mass $\mu^2(k,\tau)$ of the oscillator is given by the following expression:

$$\mu^2(k,\tau) := \left[k^2 - \left(\frac{z'(\tau)}{z(\tau)}\right)^2\right]. \tag{85}$$

Using the solutions of the classical mode functions, we can construct the quantum mechanical mechanical operators in the Heisenberg picture as follows:

$$\hat{v}(\mathbf{x},\tau) = \mathcal{U}^\dagger(\tau,\tau_0)\hat{v}(\mathbf{x},\tau_0)\mathcal{U}(\tau,\tau_0)$$
$$= \int \frac{d^3\mathbf{k}}{(2\pi)^3}\left[v^*_{-\mathbf{k}}(\tau)\,\hat{a}_{\mathbf{k}} + v_{\mathbf{k}}(\tau)\,\hat{a}^\dagger_{-\mathbf{k}}\right]\exp(i\mathbf{k}.\mathbf{x}), \tag{86}$$
$$\hat{\pi}(\mathbf{x},\tau) = \mathcal{U}^\dagger(\tau,\tau_0)\hat{\pi}(\mathbf{x},\tau_0)\mathcal{U}(\tau,\tau_0)$$
$$= \int \frac{d^3\mathbf{k}}{(2\pi)^3}\left[\pi^*_{-\mathbf{k}}(\tau)\,\hat{a}_{\mathbf{k}} + \pi_{\mathbf{k}}(\tau)\,\hat{a}^\dagger_{-\mathbf{k}}\right]\exp(i\mathbf{k}.\mathbf{x}). \tag{87}$$

The canonical Hamiltonian for the parametric oscillator can be expressed in terms of the above mentioned quantum operators as follows.

$$
\begin{aligned}
\widehat{H}(\tau) &= \int d^3\mathbf{k}\left[\frac{1}{2}\left|\left[v^{*\prime}_{-\mathbf{k}}(\tau)\,\hat{a}_{\mathbf{k}} + v^{\prime}_{\mathbf{k}}(\tau)\,\hat{a}^{\dagger}_{-\mathbf{k}}\right] + \frac{z^{\prime}(\tau)}{z(\tau)}\left[v^{*}_{-\mathbf{k}}(\tau)\,\hat{a}_{\mathbf{k}} + v_{\mathbf{k}}(\tau)\,\hat{a}^{\dagger}_{-\mathbf{k}}\right]\right|^2 \right.\\
&\qquad\qquad \left. + \frac{1}{2}\mu^2(k,\tau)\left|\left[v^{*}_{-\mathbf{k}}(\tau)\,\hat{a}_{\mathbf{k}} + v_{\mathbf{k}}(\tau)\,\hat{a}^{\dagger}_{-\mathbf{k}}\right]\right|^2\right] \\
&= \frac{1}{2}\int d^3\mathbf{k}\left[\underbrace{\Omega_{\mathbf{k}}(\tau)\left(\hat{a}^{\dagger}_{\mathbf{k}}\hat{a}_{\mathbf{k}} + \hat{a}^{\dagger}_{-\mathbf{k}}\hat{a}_{-\mathbf{k}} + 1\right)}_{\textbf{Contribution from the free term}}\right.\\
&\qquad\qquad \left. + \underbrace{i\,\lambda_{\mathbf{k}}(\tau)\left(\exp(-2i\phi_{\mathbf{k}}(\tau))\hat{a}_{\mathbf{k}}\hat{a}_{-\mathbf{k}} - \exp(2i\phi_{\mathbf{k}}(\tau))\hat{a}^{\dagger}_{\mathbf{k}}\hat{a}^{\dagger}_{-\mathbf{k}}\right)}_{\textbf{Contribution from the Interaction term}}\right],
\end{aligned}
\tag{88}
$$

where we define $\Omega_{\mathbf{k}}(\tau)$ and $\lambda_{\mathbf{k}}(\tau)$ by the following expressions:

$$
\Omega_{\mathbf{k}}(\tau): = \left\{\left|v^{\prime}_{\mathbf{k}}(\tau)\right|^2 + \mu^2(k,\tau)|v_{\mathbf{k}}(\tau)|^2\right\}, \qquad \lambda_{\mathbf{k}}(\tau) := \left(\frac{z^{\prime}(\tau)}{z(\tau)}\right).
\tag{89}
$$

Here $\Omega_{\mathbf{k}}(\tau)$ represents the conformal time dependent dispersion relation for our set-up, and $\lambda_{\mathbf{k}}(\tau)$ is the slowly conformal time varying function $\ln z(\tau)$, where $z(\tau) = a\sqrt{2\epsilon}$, is the *Mukhanov variable*. We request the readers to kindly refer to the appendix of [20] for the details of the computation of the previous subsections.

### 5.4. Fixing the Initial Condition

We fix the initial condition in such a way that, at the time scale $\tau = \tau_0$, we get the following normalization, provided we have imposed a constraint that, $k\tau_0 = -1$:

$$
v_{\mathbf{k}}(\tau_0) = \frac{1}{\sqrt{2k}}\,2^{\nu_{\text{island}}-1}\left|\frac{\Gamma(\nu_{\text{island}})}{\Gamma\left(\frac{3}{2}\right)}\right|\,\exp\left(-i\left\{\frac{\pi}{2}(\nu_{\text{island}}-2)-1\right\}\right),
\tag{90}
$$

$$
\begin{aligned}
\pi_{\mathbf{k}}(\tau_0) = {}& i\sqrt{\frac{k}{2}}\,2^{\nu_{\text{island}}-\frac{3}{2}}\left|\frac{\Gamma(\nu_{\text{island}})}{\Gamma\left(\frac{3}{2}\right)}\right|\,\exp\left(-i\left\{\frac{\pi}{2}(\nu_{\text{island}}-2)-1\right\}\right)\\
&\left[1 - \sqrt{2}\frac{\left(\nu_{\text{island}}-\frac{1}{2}\right)\left(\nu_{\text{B}}+\frac{1}{2}+i\right)}{\left(\nu_{\text{island}}+\frac{1}{2}\right)}\exp\left(-\frac{i\pi}{4}\right)\right],
\end{aligned}
\tag{91}
$$

It is expected that at any arbitrary time scale $\tau$, the associated quantum operators can be written in the Heisenberg picture as:

$$
\hat{v}_{\mathbf{k}}(\tau) = v_{\mathbf{k}}(\tau_0)\left(a_{\mathbf{k}}(\tau) + a^{\dagger}_{-\mathbf{k}}(\tau)\right),
\tag{92}
$$

$$
\hat{\pi}_{\mathbf{k}}(\tau) = -\pi_{\mathbf{k}}(\tau_0)\left(a_{\mathbf{k}}(\tau) - a^{\dagger}_{-\mathbf{k}}(\tau)\right),
\tag{93}
$$

The ladder operators at any later time scale $\tau$ can also be expressed in terms of the initial time scale $\tau_0$ using the similarity transformation in the Heisenberg picture.

$$
a_{\mathbf{k}}(\tau) := \mathcal{U}^{\dagger}(\tau,\tau_0)a_{\mathbf{k}}\mathcal{U}(\tau,\tau_0),
\tag{94}
$$

$$
a^{\dagger}_{-\mathbf{k}}(\tau) := \mathcal{U}^{\dagger}(\tau,\tau_0)a^{\dagger}_{-\mathbf{k}}\mathcal{U}(\tau,\tau_0).
\tag{95}
$$

The role of the squeezed state formalism in QM can be realised while determining the expression of the Unitary operator in the context of cosmological perturbations of the scalar modes.

### 5.5. Squeezed State Formalism in Island Cosmology

Following [45,46], we factorize the unitary evolution operator produced by the above Hamiltonian $\mathcal{U}$, as follows

$$\mathcal{U}(\tau, \tau_0) = \hat{\mathcal{S}}(r_{\mathbf{k}}(\tau, \tau_0), \phi_{\mathbf{k}}(\tau)) \hat{\mathcal{R}}(\theta_{\mathbf{k}}(\tau)), \tag{96}$$

where $\mathcal{R}$ is the two mode rotation operator, defined as:

$$\hat{\mathcal{R}}(\theta_{\mathbf{k}}(\tau)) = \exp\left(-i\theta_k(\tau)\left(\hat{a}_{\mathbf{k}}\hat{a}_{\mathbf{k}}^\dagger + \hat{a}_{-\mathbf{k}}^\dagger \hat{a}_{-\mathbf{k}}\right)\right), \tag{97}$$

and $\hat{\mathcal{S}}$ is the two-mode squeezing operator, defined as:

$$\hat{\mathcal{S}}(r_{\mathbf{k}}(\tau), \phi_{\mathbf{k}}(\tau)) = \exp\left(\frac{r_{\mathbf{k}}(\tau)}{2}\left[\exp(-2i\phi_{\mathbf{k}}(\tau))\hat{a}_{\mathbf{k}}\hat{a}_{-\mathbf{k}} - \exp(2i\phi_{\mathbf{k}}(\tau))\hat{a}_{-\mathbf{k}}^\dagger\hat{a}_{\mathbf{k}}^\dagger\right]\right). \tag{98}$$

The time-dependent parameters, $r_{\mathbf{k}}(\tau)$ and $\phi_{\mathbf{k}}(\tau)$ describes the squeezing amplitude and the squeezing angle, respectively. The two-mode rotation operator, $\hat{\mathcal{R}}$, also produces an irrelevant phase factor $\exp(i\theta_{\mathbf{k}}(\tau))$ while acted upon the initial quantum vacuum state and can be safely ignored. The appearance of the squeezed quantum state can be realized through the interaction of the cosmological perturbation with the conformal time-dependent scale factor. This leads to a conformal time-dependent frequency for the parametric oscillator, whose quantization is described in terms of the two-mode squeezed state formalism, as described in [45]. We choose the ground state of the free Hamiltonian as the initial quantum mechanical state:

$$\hat{a}_{\mathbf{k}}\left|0\right\rangle_{\mathbf{k},-\mathbf{k}} = 0 \quad \forall\ \mathbf{k}, \tag{99}$$

which is basically a Poincare invariant vacuum state in the present context of discussion.

The action of the squeezed quantum operator $\hat{\mathcal{S}}$ on the above initial vacuum state produces a two-mode squeezed quantum vacuum state, as:

$$\begin{aligned}\left|\Psi_{\mathbf{sq}}\right\rangle_{\mathbf{k},-\mathbf{k}} &= \hat{\mathcal{S}}(r_{\mathbf{k}}(\tau), \phi_{\mathbf{k}}(\tau))\left|0\right\rangle_{\mathbf{k},-\mathbf{k}} \\ &= \frac{1}{\cosh r_{\mathbf{k}}(\tau)} \sum_{n=0}^{\infty} (-1)^n \exp(-2in\,\phi_{\mathbf{k}}(\tau)\tanh^n r_{\mathbf{k}}(\tau)\left|n_{\mathbf{k}}, n_{-\mathbf{k}}\right\rangle,\end{aligned} \tag{100}$$

with the following two-mode excited or usually known as the occupation number state given by the following expression:

$$\left|n_{\mathbf{k}}, n_{-\mathbf{k}}\right\rangle = \frac{1}{n!}\left(\hat{a}_{\mathbf{k}}^\dagger\right)^n\left(\hat{a}_{-\mathbf{k}}^\dagger\right)^n\left|0\right\rangle_{\mathbf{k},-\mathbf{k}}. \tag{101}$$

Consequently, in the present context of discussion the full quantum wave function can be expressed in terms of the product of the wave function for each two-mode pair as $\mathbf{k}, -\mathbf{k}$ given by the following expression:

$$\begin{aligned}\left|\Psi_{\mathbf{sq}}\right\rangle &= \bigotimes_{\mathbf{k}}\left|\Psi_{\mathbf{sq}}\right\rangle_{\mathbf{k},-\mathbf{k}} \\ &= \bigotimes_{\mathbf{k}} \frac{1}{\cosh r_{\mathbf{k}}(\tau)}\left(\sum_{n=0}^{\infty} \frac{(-1)^n}{n!} \exp(-2in\,\phi_{\mathbf{k}}(\tau)\tanh^n r_{\mathbf{k}}(\tau)\left(\hat{a}_{\mathbf{k}}^\dagger\right)^n\left(\hat{a}_{-\mathbf{k}}^\dagger\right)^n\right)\left|0\right\rangle_{\mathbf{k},-\mathbf{k}},\end{aligned}$$
$$\tag{102}$$

### 5.6. Time Evolution in Squeezed State Formalism

We begin by expressing the creation and annihilation operators of the parametric oscillator in terms of the squeezed states and using the factorized form of the unitary operator introduced in the previous subsection, the expression for the creation and annihilation operator can be written at any arbitrary time scale as:

$$
\begin{aligned}
\hat{a}_{\mathbf{k}}(\tau) &= \hat{\mathcal{U}}^{\dagger}(\tau, \tau_0)\, \hat{a}_{\mathbf{k}}\, \hat{\mathcal{U}}(\tau, \tau_0) \\
&= \hat{\mathcal{R}}^{\dagger}(\theta_{\mathbf{k}}(\tau))\hat{\mathcal{S}}^{\dagger}(r_{\mathbf{k}}(\tau), \phi_{\mathbf{k}}(\tau))\, \hat{a}_{\mathbf{k}}\, \hat{\mathcal{R}}(\theta_{\mathbf{k}}(\tau))\hat{\mathcal{S}}(r_{\mathbf{k}}(\tau), \phi_{\mathbf{k}}(\tau)) \\
&= \cosh r_{\mathbf{k}}(\tau)\, \exp(-i\theta_{\mathbf{k}}(\tau))\, \hat{a}_{\mathbf{k}} - \sinh r_{\mathbf{k}}(\tau)\, \exp(i(\theta_{\mathbf{k}}(\tau) + 2\phi_{\mathbf{k}}(\tau)))\, \hat{a}^{\dagger}_{-\mathbf{k}}, \quad (103)
\end{aligned}
$$

$$
\begin{aligned}
\hat{a}^{\dagger}_{-\mathbf{k}}(\tau) &= \hat{\mathcal{U}}^{\dagger}(\tau, \tau_0)\, \hat{a}^{\dagger}_{-\mathbf{k}}\, \hat{\mathcal{U}}(\tau, \tau_0) \\
&= \hat{\mathcal{R}}^{\dagger}(\theta_{\mathbf{k}}(\tau))\hat{\mathcal{S}}^{\dagger}(r_{\mathbf{k}}(\tau), \phi_{\mathbf{k}}(\tau))\, \hat{a}^{\dagger}_{-\mathbf{k}}\, \hat{\mathcal{R}}(\theta_{\mathbf{k}}(\tau))\hat{\mathcal{S}}(r_{\mathbf{k}}(\tau), \phi_{\mathbf{k}}(\tau)) \\
&= \cosh r_{\mathbf{k}}(\tau)\, \exp(i\theta_{\mathbf{k}}(\tau))\, \hat{a}^{\dagger}_{-\mathbf{k}} - \sinh r_{\mathbf{k}}(\tau)\, \exp(-i(\theta_{\mathbf{k}}(\tau) + 2\phi_{\mathbf{k}}(\tau)))\, \hat{a}_{\mathbf{k}}. \quad (104)
\end{aligned}
$$

Consequently, the quantum operator associated with the cosmological perturbation field variable for the scalar fluctuation and the its canonically conjugate momenta can be expressed as:

$$
\begin{aligned}
\hat{v}_{\mathbf{k}}(\tau) &= v_{\mathbf{k}}(\tau_0)\left(\hat{a}_{\mathbf{k}}(\tau) + \hat{a}^{\dagger}_{-\mathbf{k}}(\tau)\right) \\
&= v_{\mathbf{k}}(\tau_0)\Bigg[\hat{a}_{\mathbf{k}}\bigg(\cosh r_{\mathbf{k}}(\tau)\, \exp(-i\theta_{\mathbf{k}}(\tau)) - \sinh r_{\mathbf{k}}(\tau)\, \exp(-i(\theta_{\mathbf{k}}(\tau) + 2\phi_{\mathbf{k}}(\tau)))\bigg) \quad (105) \\
&\qquad\qquad + \hat{a}^{\dagger}_{-\mathbf{k}}\bigg(\cosh r_{\mathbf{k}}(\tau)\, \exp(i\theta_{\mathbf{k}}(\tau)) - \sinh r_{\mathbf{k}}(\tau)\, \exp(i(\theta_{\mathbf{k}}(\tau) + 2\phi_{\mathbf{k}}(\tau)))\bigg)\Bigg], \\
&= \left[v^{*}_{-\mathbf{k}}(\tau)\, \hat{a}_{\mathbf{k}} + v_{\mathbf{k}}(\tau)\, \hat{a}^{\dagger}_{-\mathbf{k}}\right],
\end{aligned}
$$

$$
\begin{aligned}
\hat{\pi}_{\mathbf{k}}(\tau) &= -\pi_{\mathbf{k}}(\tau_0)\left(a_{\mathbf{k}}(\tau) - a^{\dagger}_{-\mathbf{k}}(\tau)\right) \\
&= -\pi_{\mathbf{k}}(\tau_0)\Bigg[\hat{a}_{\mathbf{k}}\bigg(\cosh r_{\mathbf{k}}(\tau)\, \exp(-i\theta_{\mathbf{k}}(\tau)) + \sinh r_{\mathbf{k}}(\tau)\, \exp(-i(\theta_{\mathbf{k}}(\tau) + 2\phi_{\mathbf{k}}(\tau)))\bigg) \quad (106) \\
&\qquad\qquad - \hat{a}^{\dagger}_{-\mathbf{k}}\bigg(\cosh r_{\mathbf{k}}(\tau)\, \exp(i\theta_{\mathbf{k}}(\tau)) + \sinh r_{\mathbf{k}}(\tau)\, \exp(i(\theta_{\mathbf{k}}(\tau) + 2\phi_{\mathbf{k}}(\tau)))\bigg)\Bigg], \\
&= \left[\pi^{*}_{-\mathbf{k}}(\tau)\, \hat{a}_{\mathbf{k}} + \pi_{\mathbf{k}}(\tau)\, \hat{a}^{\dagger}_{-\mathbf{k}}\right].
\end{aligned}
$$

The classical mode function and its associated canonically conjugate momentum in terms of the squeezed parameters can be identified as:

$$
v_{\mathbf{k}}(\tau) = v_{\mathbf{k}}(\tau_0)\left(\cosh r_{\mathbf{k}}(\tau)\, \exp(i\theta_{\mathbf{k}}(\tau)) - \sinh r_{\mathbf{k}}(\tau)\, \exp(i(\theta_{\mathbf{k}}(\tau) + 2\phi_{\mathbf{k}}(\tau)))\right), \quad (107)
$$

$$
\pi_{\mathbf{k}}(\tau) = \pi_{\mathbf{k}}(\tau_0)\left(\cosh r_{\mathbf{k}}(\tau)\, \exp(i\theta_{\mathbf{k}}(\tau)) + \sinh r_{\mathbf{k}}(\tau)\, \exp(i(\theta_{\mathbf{k}}(\tau) + 2\phi_{\mathbf{k}}(\tau)))\right). \quad (108)
$$

The time evolution of the quantum operators $\hat{\mathcal{R}}$ and $\hat{\mathcal{S}}$ leads to the following sets of differential equations for the squeezing parameters:

$$
\frac{dr_{\mathbf{k}}(\tau)}{d\tau} = -\lambda_{\mathbf{k}}(\tau)\, \cos(2\phi_{\mathbf{k}}(\tau)), \quad (109)
$$

$$
\frac{d\phi_{\mathbf{k}}(\tau)}{d\tau} = \Omega_{\mathbf{k}}(\tau) + \lambda_{\mathbf{k}}(\tau)\, \coth(2r_{\mathbf{k}}(\tau))\sin(2\phi_{\mathbf{k}}(\tau)), \quad (110)
$$

where the time dependent factors, $\lambda_{\mathbf{k}}(\tau)$ and $\Omega_{\mathbf{k}}(\tau)$ in the squeezed state picture in the $-k\tau \gg 1$ can be recast as:

$$
\lambda_{\mathbf{k}}(\tau) : \quad = \quad \left( \frac{z'(\tau)}{z(\tau)} \right) = \mathcal{H}\left[ \frac{1}{\epsilon(\tau)} - 1 + \epsilon(\tau) - \frac{1}{2}\frac{1}{\epsilon(\tau)}\frac{\mathcal{H}''}{\mathcal{H}^3} \right],
\tag{111}
$$

$$
\Omega_{\mathbf{k}}(\tau) : \quad = \quad \left\{ |\pi_{\mathbf{k}}(\tau) + \lambda_{\mathbf{k}}(\tau)v_{\mathbf{k}}(\tau)|^2 + \left( k^2 - \lambda_{\mathbf{k}}^2(\tau) \right)|v_{\mathbf{k}}(\tau)|^2 \right\}
$$

$$
\approx \quad 3k\, 2^{2(\nu_{\text{island}}-2)} \left| \frac{\Gamma(\nu_{\text{island}})}{\Gamma(\frac{3}{2})} \right|^2 .
\tag{112}
$$

See Appendix B for more details.

*5.7. Quantum Complexity from Squeezed Quantum States in Island Cosmology*

To compute the complexity from squeezed formalism we use the wave function formalism of computing circuit complexity developed by [88] and used extensively in [21,22,92]. We fix a reference state $|0\rangle_{\mathbf{k},-\mathbf{k}}$, commonly used in cosmological perturbations. The squeezed two-mode vacuum state $|\Psi_{\mathbf{sq}}\rangle_{\mathbf{k},-\mathbf{k}}$ becomes the target state.

The reference two-mode vacuum state wave function is given by:

$$
\hat{a}_{\mathbf{k}} |0\rangle_{\mathbf{k},-\mathbf{k}} = 0 \quad \forall\, \mathbf{k}
\tag{113}
$$

which has the following usual Gaussian structure:

$$
\Psi_{\text{Ref}}(v_{\mathbf{k}}, v_{-\mathbf{k}}) := \left( \frac{\Omega_{\mathbf{k}}}{\pi} \right)^{1/4} \exp\left( -\frac{\Omega_{\mathbf{k}}}{2}(v_{\mathbf{k}}^2 + v_{-\mathbf{k}}^2) \right)
\tag{114}
$$

where we have used the expression for $\Omega_{\mathbf{k}}$ in the sub-Hubble region, that we analytically approximated.

By noting that a specific squeezing parameters with the annihilation and creation operators fixes the wave function we can write it as:

$$
\left( \cosh r_{\mathbf{k}}(\tau)\, \hat{a}_{\mathbf{k}} + \exp(-2i\phi_{\mathbf{k}}(\tau)) \sinh r_{\mathbf{k}}(\tau)\, \hat{a}_{-\mathbf{k}}^{\dagger} \right) |\Psi_{\mathbf{sq}}\rangle_{\mathbf{k},-\mathbf{k}} = 0.
\tag{115}
$$

The perturbed field space representation is given by:

$$
\begin{aligned}
\Psi_{\mathbf{sq}}(v_{\mathbf{k}}, v_{-\mathbf{k}}) \quad &= \quad \langle v_{\mathbf{k}}, v_{-\mathbf{k}} | \Psi_{\mathbf{sq}} \rangle_{\mathbf{k},-\mathbf{k}} \\
&= \quad \frac{\exp\left( \mathcal{A}(\tau)\,(v_{\mathbf{k}}^2 + v_{-\mathbf{k}}^2) - \mathcal{B}(\tau)\, v_{\mathbf{k}}\, v_{-\mathbf{k}} \right)}{\cosh r_{\mathbf{k}}(\tau)\sqrt{\pi(1 - \exp(-4i\phi_{\mathbf{k}}(\tau))\, \tanh^2 r_{\mathbf{k}}(\tau) - 1)}},
\end{aligned}
\tag{116}
$$

where the coefficients $\mathcal{A}(\tau)$ and $\mathcal{B}(\tau)$ are the functions of $r_{\mathbf{k}}(\tau)$ and $\phi_{\mathbf{k}}(\tau)$, given by:

$$
\mathcal{A}(\tau) := \frac{\Omega_{\mathbf{k}}}{2}\left( \frac{\exp(-4i\phi_{\mathbf{k}}(\tau))\, \tanh^2 r_{\mathbf{k}}(\tau) + 1}{\exp(-4i\phi_{\mathbf{k}}(\tau))\, \tanh^2 r_{\mathbf{k}}(\tau) - 1} \right),
\tag{117}
$$

$$
\mathcal{B}(\tau) := 2\Omega_{\mathbf{k}}\left( \frac{\exp(-2i\phi_{\mathbf{k}}(\tau))\, \tanh^2 r_{\mathbf{k}}(\tau)}{\exp(-4i\phi_{\mathbf{k}}(\tau))\, \tanh^2 r_{\mathbf{k}}(\tau) - 1} \right).
\tag{118}
$$

Generally in literature people use conformal time as the dynamical variable for computational purposes. However, to make our computation physically justifiable we use the scale factor as the dynamical variable. Performing the change in the dynamical variable is a trivial task:

$$
\tau \to a(\tau) : \frac{d}{d\tau} = a'(\tau)\frac{d}{da(\tau)}
\tag{119}
$$

The evolution equations of the squeezed state state parameter and the squeezed state angle can written in terms of the new dynamical variable a($\tau$) as:

$$\frac{dr_k(a)}{da} = -\frac{\lambda_k(a)}{a'} \cos 2\phi_k(a), \tag{120}$$

$$\frac{d\phi_k(a)}{da} = \frac{\Omega_k}{a'} - \frac{\lambda_k(a)}{a'} \coth 2r_k(a) \sin 2\phi_k(a) \tag{121}$$

The vacuum reference and the target squeezed state written in Equations (114) and (116) is eventually used to calculate the complexity from two types of cost functions namely the "linear weighting" ($\mathcal{C}_1$) and the "geodesic weighting" ($\mathcal{C}_2$), respectively, within the framework of cosmology and represented by the following expressions:

$$\mathcal{C}_1(k) = \frac{1}{2}\left( \ln\left|\frac{\Sigma_\mathbf{k}}{\omega_\mathbf{k}}\right| + \ln\left|\frac{\Sigma_\mathbf{-k}}{\omega_\mathbf{-k}}\right| + \tan^{-1}\frac{\text{Im}\,\Sigma_\mathbf{k}}{\text{Re}\,\omega_\mathbf{k}} + \tan^{-1}\frac{\text{Im}\,\Sigma_\mathbf{-k}}{\text{Re}\,\omega_\mathbf{-k}} \right)$$

$$\mathcal{C}_2(k) = \frac{1}{2}\sqrt{\left( \ln\left|\frac{\Sigma_\mathbf{k}(\tau)}{\omega_\mathbf{k}(\tau)}\right|\right)^2 + \left( \ln\left|\frac{\Sigma_\mathbf{-k}(\tau)}{\omega_\mathbf{-k}(\tau)}\right|\right)^2 + \left( \tan^{-1}\frac{\text{Im}\,\Sigma_\mathbf{k}(\tau)}{\text{Re}\,\omega_\mathbf{k}(\tau)} + \right)^2 + \left( \tan^{-1}\frac{\text{Im}\,\Sigma_\mathbf{-k}(\tau)}{\text{Re}\,\omega_\mathbf{-k}(\tau)} \right)^2}. \tag{122}$$

where we define the following functions:

$$\Sigma_\mathbf{k}(\tau) = \mathcal{B}(\tau) - 2\mathcal{A}(\tau), \tag{123}$$

$$\Sigma_\mathbf{-k}(\tau) = -\mathcal{B}(\tau) - 2\mathcal{A}(\tau), \tag{124}$$

$$\omega_\mathbf{k}(\tau) = \frac{1}{2}\Omega_\mathbf{k}(\tau) = \omega_\mathbf{-k}(\tau). \tag{125}$$

Below, we provide a formal derivation of the expressions of the circuit complexities. As already discussed, the above expressions are derived using the Nielsen's wave function approach. However another approach which people uses is the covariance matrix method. A formal derivation of the circuit complexities in the covariance matrix approach can be found in [124]. The Nielsen's wave function approach uses wavefunctions to give circuit complexities of two mode squeezed states that is sensitive to both squeezing parameters: $r_k$ and $\phi_k$. The complexity is calculated using the reference and target two-mode squeezed states. This enables to write the circuit complexity in terms of squeezing parameters $r_k$ and $\phi_k$.

The exponent of the target state, i.e., two-mode squeezed states Equation (116) can be diagonalized as:

$$\Psi_\text{sq} = \mathcal{N}\exp\left( -\frac{1}{2}\tilde{\mathcal{M}}^{ab}v_k v_{-k} \right) \tag{126}$$

where, $\mathcal{N}$ is the normalization constant, i.e., denominator in Equation (116) and,

$$\tilde{\mathcal{M}} = \begin{bmatrix} -2A + B & 0 \\ 0 & -2A - B \end{bmatrix} = \begin{bmatrix} \Sigma_\mathbf{k} & 0 \\ 0 & \Sigma_\mathbf{-k} \end{bmatrix} \tag{127}$$

The unsqueezed state, reference state can also be written in a similar form as above, as it is also a Gaussian wave function represented by:

$$\Psi_\text{Ref} = \mathcal{N}\exp\left( -\frac{\Omega_k}{2}\left( v_\mathbf{k}^2 + v_\mathbf{-k}^2 \right)\right) = \mathcal{N}\exp\left( \frac{1}{2}\sum_{k,-k} \Omega_k|\mathbf{k}|^2 \right) \tag{128}$$

Thus, our two required state has a Gaussian wave function and is of the form:

$$\Psi^\eta = \mathcal{N}\exp\left(-\frac{1}{2}\left(q_a.\mathcal{A}^\eta_{ab}.q_b\right)\right) \tag{129}$$

where, $q = (v_{\mathbf{k}}, v_{-\mathbf{k}})$ and $\mathcal{A}^\eta$ is a $2 \times 2$ diagonal matrix. For the target state Equation (116),

$$\mathcal{A}^{\eta=1} = \mathcal{M} = \begin{bmatrix} \Sigma_{\mathbf{k}} & 0 \\ 0 & \Sigma_{-\mathbf{k}} \end{bmatrix} \tag{130}$$

while for our reference state Equation (114), matrix $\mathcal{A}$ is $\mathcal{A}^{\eta=0}$. So,

$$\mathcal{A}^{\eta=0} = \begin{bmatrix} \Omega_k & 0 \\ 0 & \Omega_{-k} \end{bmatrix} \tag{131}$$

The unitary transformation acts like,

$$\mathcal{A}^\eta = \mathcal{U}(\eta).\mathcal{A}^{\eta=0}.\mathcal{U}^T(\eta) \tag{132}$$

The boundary conditions is given by:

$$\begin{aligned} \mathcal{A}^{\eta=1} &= \mathcal{U}(\eta=1).\mathcal{A}^{\eta=0}.\mathcal{U}^T(\eta=1) \\ \mathcal{A}^{\eta=0} &= \mathcal{U}(\eta=0).\mathcal{A}^{\eta=0}.\mathcal{U}^T(\eta=0) \end{aligned} \tag{133}$$

$\mathcal{U}$ can be parametrized in terms of the tangent vectors, as discussed earlier, such that at $\eta = 1$, the required target state is achieved. Since, $\mathcal{A}^{\eta=1}$ and $\mathcal{A}^{\eta=0}$ can have complex elements, elementary gates are restricted to $GL(2, C)$ unitaries. The tangent vector components are given by:

$$Y^I = \text{Tr}(\partial_\eta U(\eta)U^{-1}(\eta)(\mathcal{O}_I)^T) \tag{134}$$

where, it is to be noted that:

$$\text{Tr}(\mathcal{O}_I.\mathcal{O}_J^T) = \delta^{IJ}, \tag{135}$$

and $I, J = 0, 1, 2, 3$. The metric is then given by:

$$ds^2 = G_{IJ}dY^I dY^{*J}. \tag{136}$$

For simplicity, we will choose penalty factors $G_{IJ} = \delta^{IJ}$ where we fix it to unity. The off-diagonal elements in $GL(2, C)$ can be set to zero as they increase the distance between states. The $U(\eta)$ will become:

$$U(\eta) = \exp\left(\sum_{i\in(k,-k)} \alpha^i(\eta)\mathcal{O}_i^{diagonal}\right) \tag{137}$$

where, $\alpha^i(\eta)$ are complex parameters and $\mathcal{O}_i^{diagonal}$ are generators with identity at $i$ diagonal elements. The metric takes a simple form:

$$ds^2 = \sum_{i\in(k,-k)} (d\alpha^{i,\text{Re}})^2 + (d\alpha^{i,\text{Im}})^2 \tag{138}$$

where, Re and Im indicates real and imaginary part of $\alpha_k$, respectively. The geodesic is again a straight line in the manifold given by:

$$\alpha^{i,p}(\eta) = \alpha^{i,p}(\eta = 1) + \alpha^{i,p}(\eta = 0) \tag{139}$$

for each $(i \in k, -k)$ and $(p = \text{Re and Im})$. For the given boundary conditions written earlier, one gets

$$
\begin{aligned}
\alpha^{i,\text{Re}}(\eta = 0) &= \alpha^{i,\text{Im}}(\eta = 0) = 0, \\
\alpha^{i,\text{Re}}(\eta = 1) &= \frac{1}{2}\ln\left|\frac{\Sigma_{\mathbf{i}}}{\omega_{\mathbf{i}}}\right| \\
\alpha^{i,\text{Im}}(\eta = 1) &= \frac{1}{2}\tan^{-1}\frac{\text{Im}(\Sigma_{\mathbf{i}})}{\text{Re}(\Sigma_{\mathbf{i}})}
\end{aligned}
\tag{140}
$$

for each $(i \in k, -k)$. Now, the circuit complexity for linear $C_1(\Omega_k)$ and quadratic cost $C_2(\Omega_k)$ functions can be derived as follows:

$$
\begin{aligned}
C_1(\Omega_k) &= \alpha^{k,\text{Re}}(\eta = 1) + \alpha^{-k,\text{Re}}(\eta = 1) + \alpha^{k,\text{Im}}(\eta = 1) + \alpha^{-k,\text{Im}}(\eta = 1) \\
&= \frac{1}{2}\left(\ln\left|\frac{\Sigma_{\mathbf{k}}}{\omega_{\mathbf{k}}}\right| + \ln\left|\frac{\Sigma_{-\mathbf{k}}}{\omega_{-\mathbf{k}}}\right| + \tan^{-1}\frac{\text{Im}(\Sigma_{\mathbf{k}})}{\text{Re}(\Sigma_{\mathbf{k}})} + \tan^{-1}\frac{\text{Im}(\Sigma_{-\mathbf{k}})}{\text{Re}(\Sigma_{-\mathbf{k}})}\right)
\end{aligned}
\tag{141}
$$

$$
\begin{aligned}
C_2(\Omega_k) &= \sqrt{(\alpha^{k,\text{Re}}(\eta = 1))^2 + (\alpha^{-k,\text{Re}}(\eta = 1))^2 + (\alpha^{k,\text{Im}}(\eta = 1))^2 + (\alpha^{-k,\text{Im}}(\eta = 1))^2} \\
&= \frac{1}{2}\sqrt{\left(\ln\left|\frac{\Sigma_{\mathbf{k}}}{\omega_{\mathbf{k})}}\right|\right)^2 + \left(\ln\left|\frac{\Sigma_{-\mathbf{k}}}{\omega_{-\mathbf{k})}}\right|\right)^2 + \left(\tan^{-1}\frac{\text{Im}(\Sigma_{\mathbf{k}})}{\text{Re}(\Sigma_{\mathbf{k}})}\right)^2 + \left(\tan^{-1}\frac{\text{Im}(\Sigma_{-\mathbf{k}})}{\text{Re}(\Sigma_{-\mathbf{k}})}\right)^2}
\end{aligned}
\tag{142}
$$

Using the expressions of $\Sigma_{\mathbf{k}}$, $\Sigma_{-\mathbf{k}}$, $\omega_{\mathbf{k}}$, and $\omega_{-\mathbf{k}}$ the general circuit complexity takes the following form:

$$
C_1(\Omega_k, \eta) = \left|\ln\left|\frac{1 + \exp(-2i\phi_k(\eta))\tanh r_k(\eta)}{1 - \exp(-2i\phi_k(\eta))\tanh r_k(\tau)}\right|\right| + \left|\tanh^{-1}(\sin(2\phi_k(\eta))\sinh(2r_k(\eta)))\right| \tag{143}
$$

$$
C_2(\Omega_k, \eta) = \frac{1}{\sqrt{2}}\sqrt{\left(\ln\left|\frac{1 + \exp(-2i\phi_k(\eta))\tanh r_k(\eta)}{1 - \exp(-2i\phi_k(\eta))\tanh r_k(\eta)}\right|\right)^2 + \left(\tanh^{-1}(\sin(2\phi_k(\eta))\sinh(2r_k(\eta)))\right)^2} \tag{144}
$$

## 6. Entanglement Entropy of Two Mode Squeezed States

In this section, the prime motivation is to compute the entanglement entropy for the two-mode squeezed states and compare it to the circuit complexity. Apart from being entangled, there exists a strong correlation between the two modes of the state. $|\Psi_{\text{sq}}\rangle_{\mathbf{k},-\mathbf{k}}$ is also an eigenstate of the operator $\hat{n}_{\mathbf{k}} - \hat{n}_{-\mathbf{k}}$ with eigenvalue 0, where $\hat{n}_k = \hat{a}^\dagger_{\mathbf{k}}\hat{a}_{-\mathbf{k}}$ and $\hat{n}_{-k} = \hat{a}^\dagger_{-\mathbf{k}}\hat{a}_{\mathbf{k}}$. Due to this strong correlation and symmetry between the two modes, average photon number is identical in each mode:

$$\langle\hat{n}_k\rangle = \langle\hat{n}_{-k}\rangle = \sinh^2 r_k \tag{145}$$

The reduced density operators for the individual modes can be written as:

$$\hat{\rho}_k = \sum_{n=0}^{\infty}\frac{1}{(\cosh r_k)^2}(\tanh r_k)^{2n}\langle n_k|n_k\rangle, \tag{146}$$

$$\hat{\rho}_{-k} = \sum_{n=0}^{\infty}\frac{1}{(\cosh r_{-k})^2}(\tanh r_{-k})^{2n}\langle n_{-k}|n_{-k}\rangle. \tag{147}$$

The probability of having $n$ photons in a single mode $k$ or $-k$ is given by:

$$P_n^{(i)} = \frac{(\tanh r_k)^{2n}}{(\cosh r_k)^2}, i = k, -k \tag{148}$$

The most commonly used entanglement entropies are the von-Neumann entanglement entropy and the Renyi entropy, on which we have focussed on this paper. For a density operator $\hat{\rho}$, von-Neumann entropy is given by:

$$S(\hat{\rho}) = -\text{Tr}[\hat{\rho} \ln\hat{\rho}] \tag{149}$$

For a pure state the von-Neumann entropy is zero while for mixed states it is greater than zero. However, it is usually not a trivial task to calculate the entropy, but for the basis in which density operator is diagonal, such as in Schmidt basis, entropy can be calculated simply from the diagonal elements as:

$$S(\hat{\rho}) = -\text{Tr}[\hat{\rho}\ln\hat{\rho}] = -\sum_k \rho_{kk}\ln\rho_{kk} \tag{150}$$

Since the two mode squeezed state Equation (116) is already in the form of Schmidt decomposition, and the form of reduced density operators of individual modes $k$ and $-k$ is also known, the von-Neumann entanglement entropy can be calculated by realizing that the diagonal elements $\rho_{kk}$ is $P_n^{(i)}$. Then, the von-Neumann entropy is given by:

$$S(\hat{\rho}_k) = -\text{Tr}[\hat{\rho}_k\ln\hat{\rho}_k] = S(\hat{\rho}_{-k}) = -\sum_{n=0}^{\infty} P_n \ln P_n$$
$$= \ln(\cosh^2 r_k)\cosh^2 r_k - \ln(\sinh^2 r_k)\sinh^2 r_k \tag{151}$$

It is to be noted that the entropy corresponding to the squeezed state Equation (116) is not calculated because naturally this entropy is going to be zero as it is a pure state. Instead, we have calculated entropy for the reduced density matrix.

The von-Neumann entropy can now be generalized to get the Rényi entropy for the reduced density operator:

$$S_\mu = \frac{1}{1-\mu}\ln\sum_{n=1}^{d} P_n = \frac{2\mu \ln \cosh r_k + \ln(1 - \tanh^{2\mu} r_k)}{\mu - 1} \tag{152}$$

where $\mu \geq 0$ is the Rényi parameter and $d$ is the Schmidt rank of the squeezed state Equation (116) which is infinity.

For very large squeezing parameter, we get:

$$S_\mu(r_k \to \infty) \approx \frac{2\mu r_k}{(\mu - 1)} \tag{153}$$

On taking the limit, $\mu \to 1$, we get the von-Neumann entropy Equation (151). Meanwhile, Rényi-2 entropy is given by $S_2(r_k) = \ln \cosh 2r_k$.

One can also calculate the effective temperature of the source by computing the thermal distribution with an average photon number $\langle \hat{n}_i \rangle = \sinh^2 r_k$. The average photon number of the thermal field is given by:

$$\langle \hat{n}_i \rangle = \bar{n} = \frac{1}{\exp(\beta\omega) - 1} \tag{154}$$

Then, one can compute the effective temperature as:

$$T = \omega_i \ln\left(\frac{\langle \hat{n}_i \rangle}{\langle \hat{n}_i \rangle + 1}\right)$$
$$= \frac{\omega_i}{2 \ln(\coth r_k)}$$

where, $\omega_i = i/c$ is the frequency of the mode and $i \in (k, -k)$.

## 7. Numerical Study with Cosmological Islands

Our objective is to numerically solve the time evolution differential equations satisfied by the squeezed state parameters in this section. We have used the scale factor as the dynamical variable instead of the conformal time, making our computation physically justifiable. This change in the variable is commonly known as field redefinition. Once we solve the differential equations, it will enable us to compute the circuit complexity between two reference states within the framework of cosmological perturbation theory. The effects of quantum fluctuations are treated in terms of squeezed states. We numerically plot the complexities calculated from two different cost functionals for both the models of the cosmological scale factors. Using the logic given in [20], we write the expression for the complexity in the exponentially increasing region as

$$\mathcal{C}_i(a) \approx c_i \, \exp(\lambda_i a)_{a=a_{\exp}} \quad \forall \ \ i = 1, 2 \tag{155}$$

It is to be noted that the above equation is valid only for the exponentially rising region; hence the subscript $a_{\exp}$ has been used, which we indeed observe for both the measures of complexity in both the models. The index "i" in the above equation indicates which measure of complexity is being used. The slopes and the amplitudes are written with index $i$ to indicate that they are different for different models. Mathematically, this can be represented as

$$\lambda_i = \left(\frac{d \ln \mathcal{C}_i(a)}{da}\right) \quad \forall \quad i = 1, 2, \tag{156}$$

One can also conjecture a similar relation between OTOC and complexity for the exponentially rising region keeping in mind that complexity and OTOC are related by $C = -\ln(OTOC)$. Hence for the exponentially rising region, the out-of-time ordered correlation function can be written as

$$OTOC \approx \exp(-c \, \exp(\lambda a)) \tag{157}$$

In Reference [20], the authors identified the slope $\lambda$ as the quantum Lyapunov exponent which captures the effect of chaos in the quantum regime and showed the existence of a universal relation between the different measures of complexity. It is represented as

$$\mathcal{C} = -\ln(OTOC) \approx \mathcal{C}_i \ \forall \ i = 1, 2 \tag{158}$$

The above universal relation between the complexities can be translated to the Lyapunov exponent through the MSS bound. Thus,

$$\lambda_i \precsim \lambda \leq \frac{2\pi}{\beta} \ \forall \ i = 1, 2 \tag{159}$$

This relation can further be used to estimate the lower bound on the equilibrium temperature, which can be done using the following relation

$$T \succsim \frac{\lambda_i}{2\pi} \ \forall \ i = 1, 2 \quad \implies \quad T \succsim \frac{1}{2\pi}\left(\frac{d \ln \mathcal{C}_i(a)}{da}\right)_{a=a_{\exp}} \ \forall \ i = 1, 2 \tag{160}$$

For our purpose, we have numerically estimated the values of the Lyapunov exponents for the both the models of scale factors using the following relation

$$\lambda_i = \frac{\ln\; C_i(a_{peak}) - \ln\; C_i(a_{rise})}{a_{peak} - a_{rise}} \tag{161}$$

The use of the above relation simplifies our task and prevents the complications of implementing numerical differentiation. Furthermore, using the relation between the circuit complexity and entanglement entropy, we numerically plotted the entropy with respect to the scale factor.

### 7.1. Islands in Recollapsing FLRW (Cosine Scale Factor)

In Figures 4 and 5 the squeezed state parameter $r_k$ and the squeezing angle $\phi_k$ are plotted with respect to the scale factor. The behavior of the squeezed state parameters determines the nature of complexity.

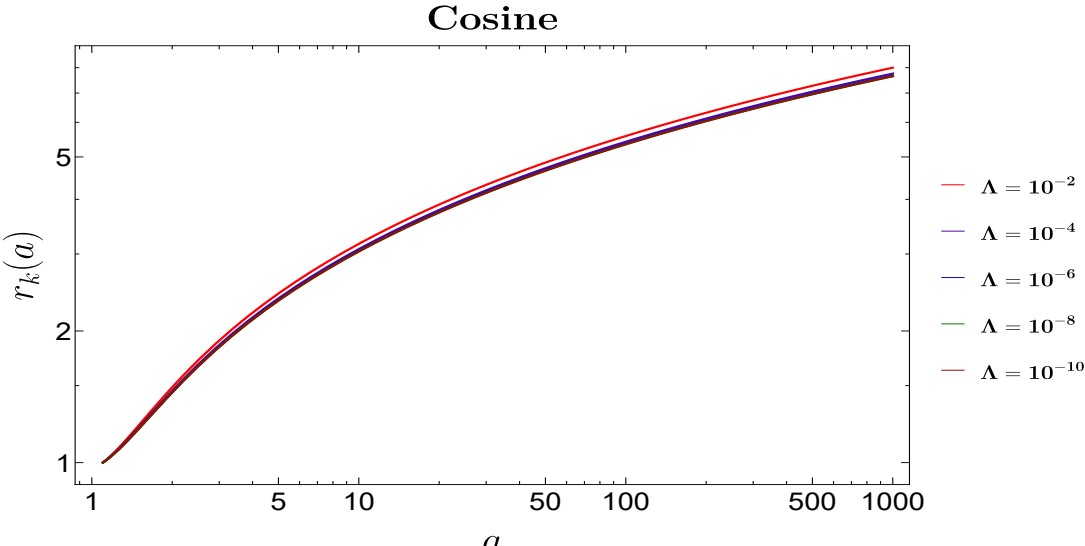

**Figure 4.** Squeezed state parameter $r_k$ plotted against scale factor.

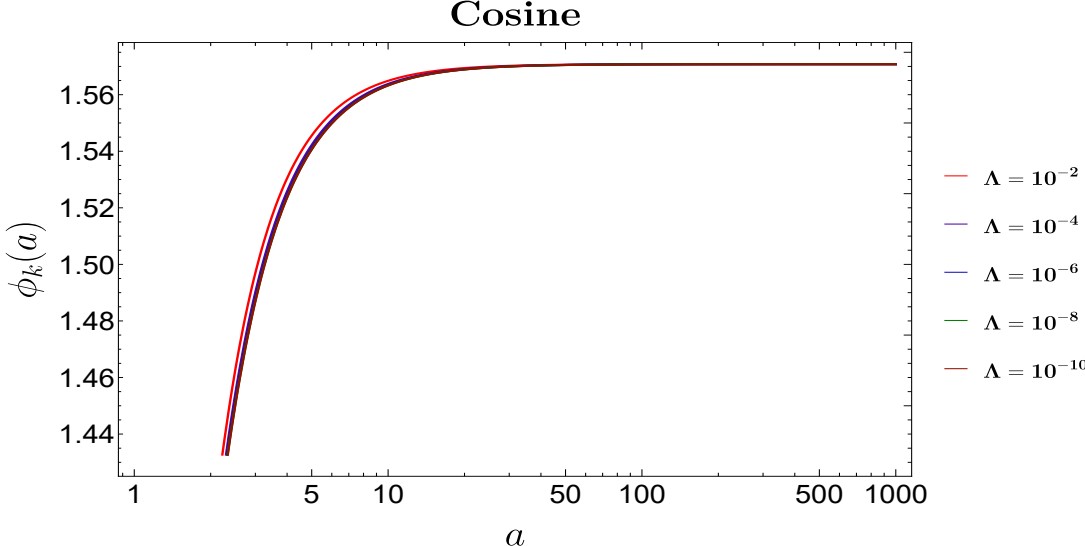

**Figure 5.** Squeezing angle $\phi_k$ plotted against scale factor.

- In Figures 6 and 7 the behavior of the circuit complexity computed from the linearly weighted and geodesically weighted cost functional are shown with respect to the scale factor. Although the overall behavior of the complexity measures are identical, some noticeable differences do occur, which are appended below:
  - The complexity measure $\mathcal{C}_1$ (linearly weighted measure) is larger than $\mathcal{C}_2$ for the entire range of scale factor;
  - At the transition point, a slight dip in $\mathcal{C}_1$ is observed, whereas, for the same point, there is a peak for $\mathcal{C}_2$.
- Figures 8 and 9 shows the plots of the *Out-of-Time-Ordered* correlation functions. Up to a certain value of scale factor the OTOC decreases exponentially as expected from [23]. However after a certain transition scale factor it starts increasing exponentially.

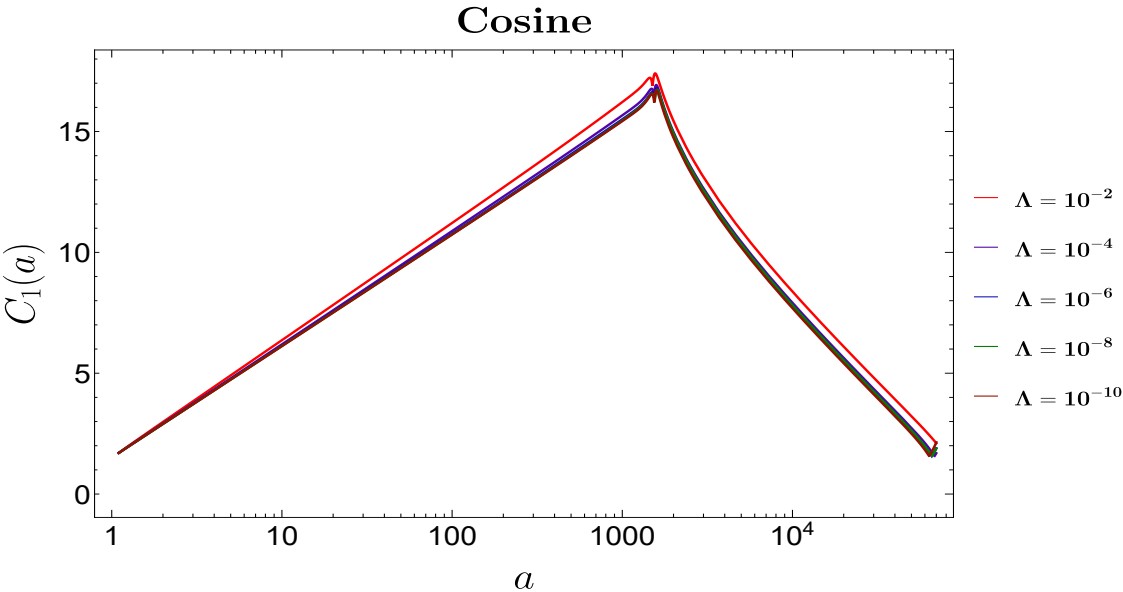

**Figure 6.** Linearly weighted complexity value plotted against scale factor.

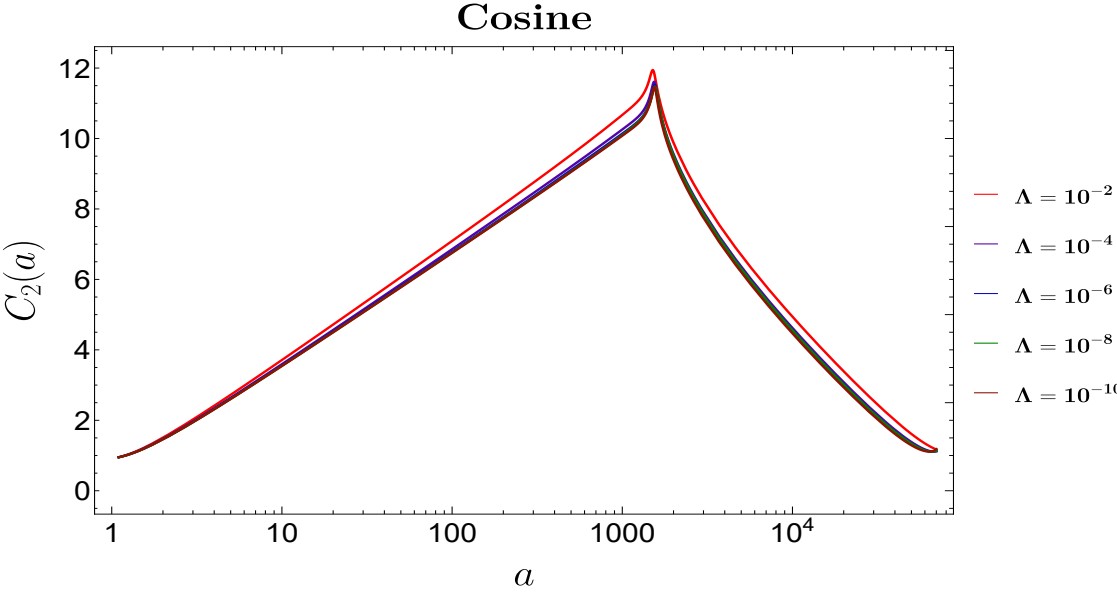

**Figure 7.** Geodesically weighted complexity value plotted against scale factor.

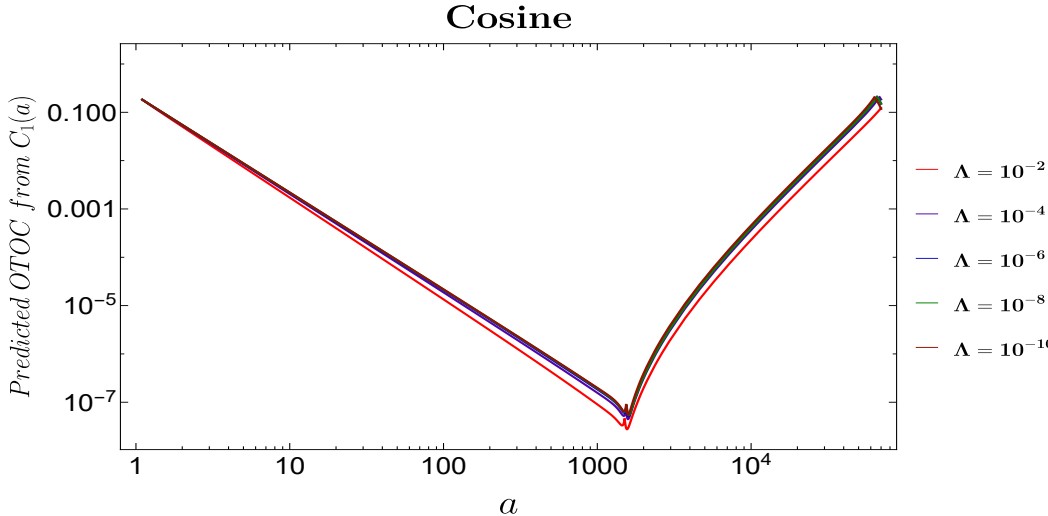

**Figure 8.** Predicted OTOC from linearly weighted cost functional plotted against scale factor.

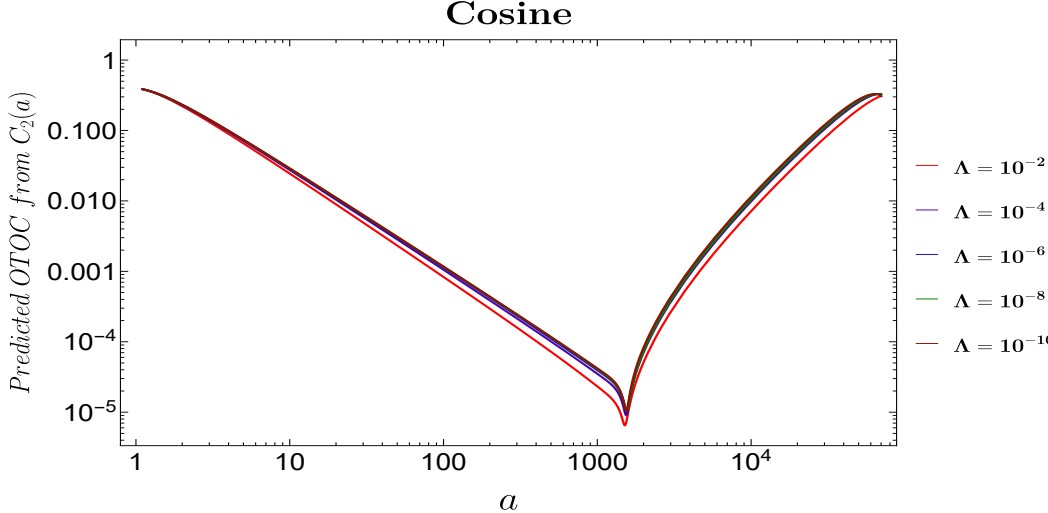

**Figure 9.** Predicted OTOC from geodesically weighted cost functional plotted against scale factor.

Table 1 shows the value of the Lyapunov exponents calculated for the initial exponentially rising region before the transition scale factor. The interesting feature to notice from the values of the Lyapunov exponents is that they obey the universal relation established in [20] which states that the Lyapunov exponents of complexity measures computed from different cost functionals are of the same order,

**Table 1.** Lyapunov exponents calculated from the $\ln(\mathcal{C})$ vs. a plots for all the different chosen values of the cosmological constants. The symbols $\Lambda_i$ in the table denotes the five chosen values of the cosmological constant, as visible in the plots.

| Complexity Measures | $\lambda_{\Lambda_1}$ | $\lambda_{\Lambda_2}$ | $\lambda_{\Lambda_3}$ | $\lambda_{\Lambda_4}$ | $\lambda_{\Lambda_5}$ |
|---|---|---|---|---|---|
| $\mathcal{C}_1$ | $9.44 \times 10^{-4}$ | $9.37 \times 10^{-4}$ | $9.35 \times 10^{-4}$ | $9.35 \times 10^{-4}$ | $9.34 \times 10^{-4}$ |
| $\mathcal{C}_2$ | $10.66 \times 10^{-4}$ | $10.59 \times 10^{-4}$ | $10.58 \times 10^{-4}$ | $10.57 \times 10^{-4}$ | $10.56 \times 10^{-4}$ |

- In Figures 10 and 11, we have plotted the entanglement entropy, viz. von-Neumann entanglement entropy and Renyi entropy of the two modes with respect to the scale factor. The entanglement entropies increases linearly with the scale factor, suggesting that the entanglement between the two modes increases linearly with the evolutionary

scale. From the similarity of the nature of entropies with the circuit complexities, at least up to a certain evolutionary scale, suggests that there might be a connecting relation between circuit complexity and entanglement entropies. An important point worth raising at this point is whether the entanglement entropy between the two modes of the squeezed state computed from the squeezed state formalism can be related to the generalized entropy of the quantum extremal island in FRW space-time. Though, not directly but some information about the quantum extremal islands is indeed encoded in the entanglement entropy of the two modes of the squeezed state. This is because, the information of the model is provided by the solution of the scale factor which has been used as the dynamical variable in solving the evolution equations of the squeezed parameters.

- In Figure 12, we have plotted the behavior of the equilibrium temperature of the two modes squeezed state with respect to the scale factor. We observe that for initial evolutionary scales, the equilibrium temperature rises sharply. This rise slows down for the intermediate scales and moves towards saturation at the large evolutionary scales. Thus, we can see that the equilibrium temperature is not a constant but has different values at different phases of the evolutionary scales.

- In Figures 13 and 14 we have plotted the complexity measures in a different parameter space, precisely for extremely small values of the cosmological constant. We observe that unlike the parameter space where the values of the cosmological constants were taken to be large, the complexity in this parameter space just shows an exponentially rising behavior throughout the entire evolutionary scale. The decreasing behavior that was observed for large values of cosmological constants is not observed in this case suggesting that the behavior of the complexity is not independent of the parameters of the chosen model.

- Figures 15 and 16, shows the behavior of the OTOC predicted from the circuit complexities in the parameter space where the cosmological constant values are very small. We again observe a feature that is different from the OTOCs computed in the other parameter space. Unlike the previous case, in this parameter space, the OTOC saturates at large evolutionary scales. The initial decreasing behavior at the early evolutionary scales is, however, identical in both the parameter space. This suggests that the behavior of the circuit complexity and the OTOCs are not universal for a given model and depends on the choice of the parameter space.

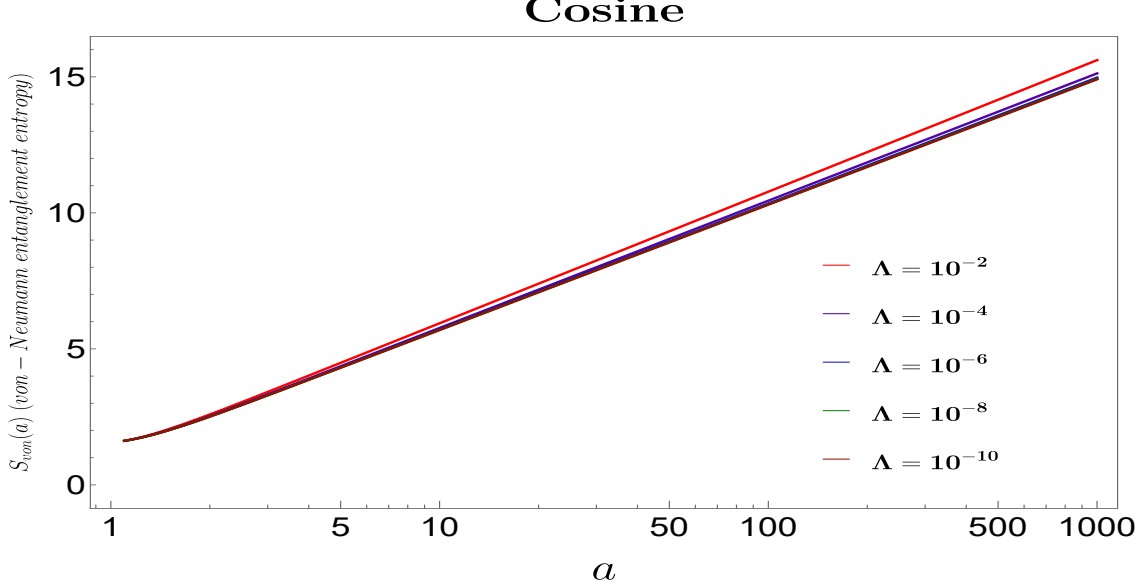

**Figure 10.** Von-Neumann entanglement entropy plotted as a function of the scale factor.

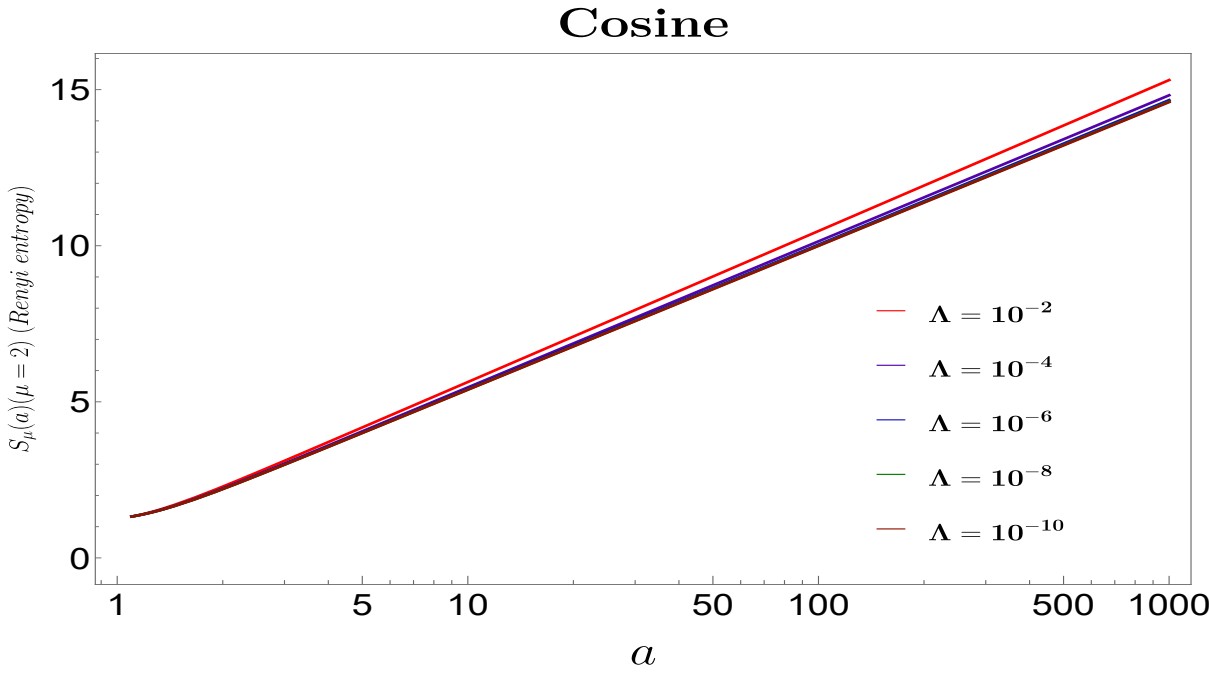

**Figure 11.** Renyi entanglement entropy plotted as a function of the scale factor.

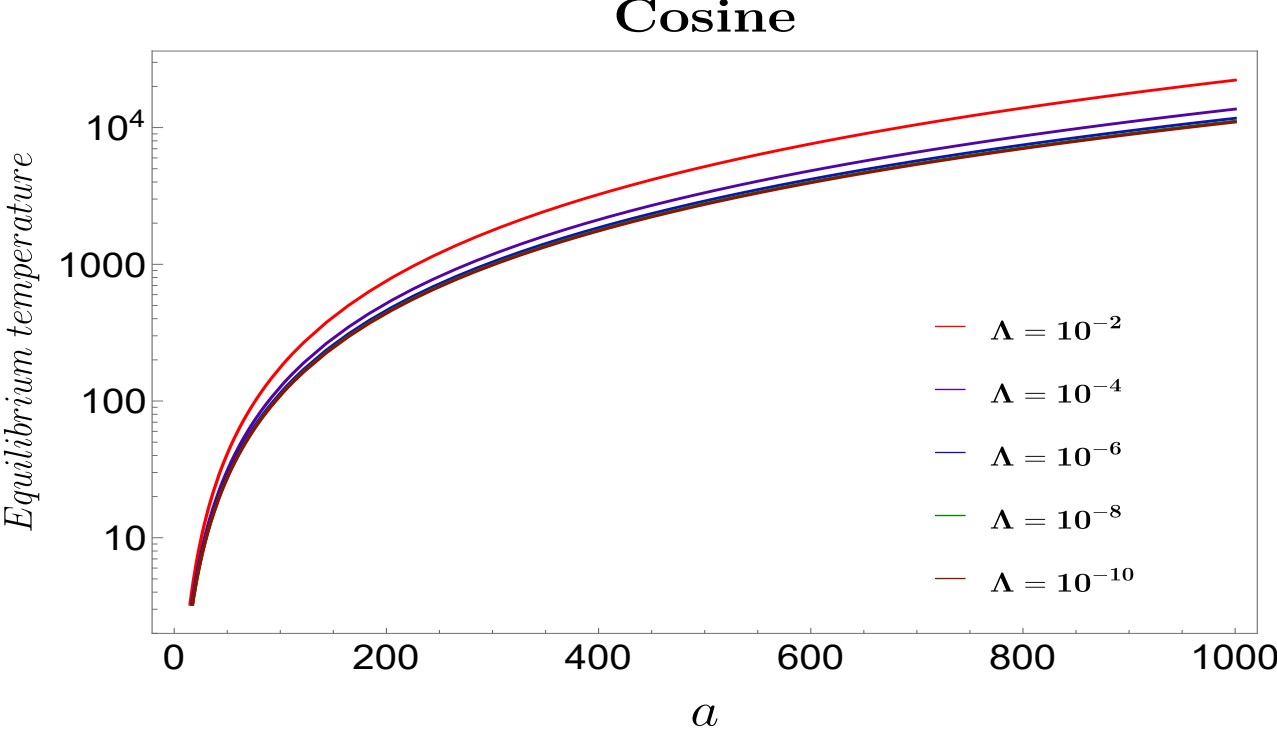

**Figure 12.** Entanglement entropy computed from $\mathcal{C}_2$ plotted against scale factor in the chaotic region.

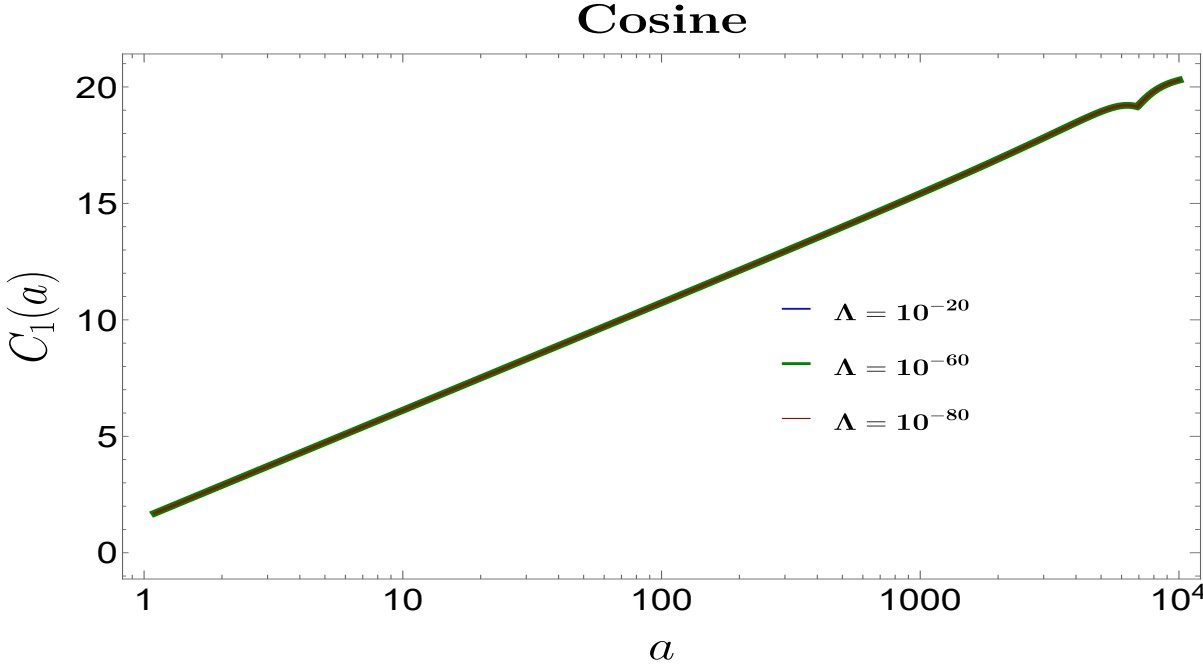

**Figure 13.** Behavior of $\mathcal{C}_1$ against scale factor in a different parameter space.

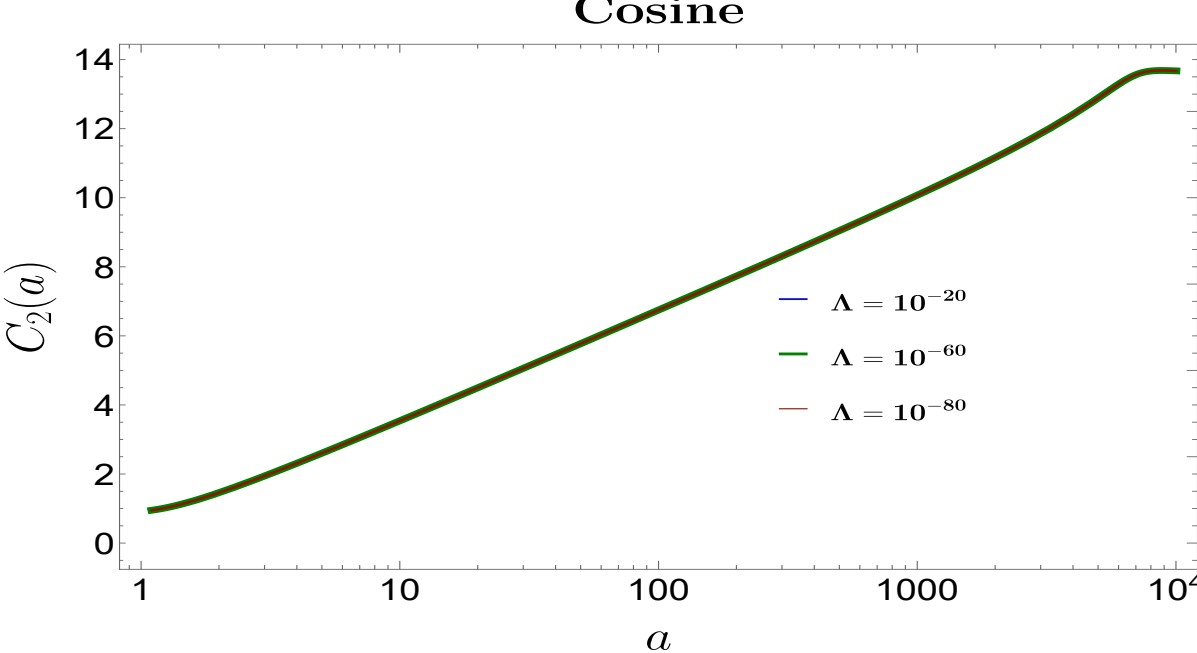

**Figure 14.** Behavior of $\mathcal{C}_2$ against scale factor in a different space.

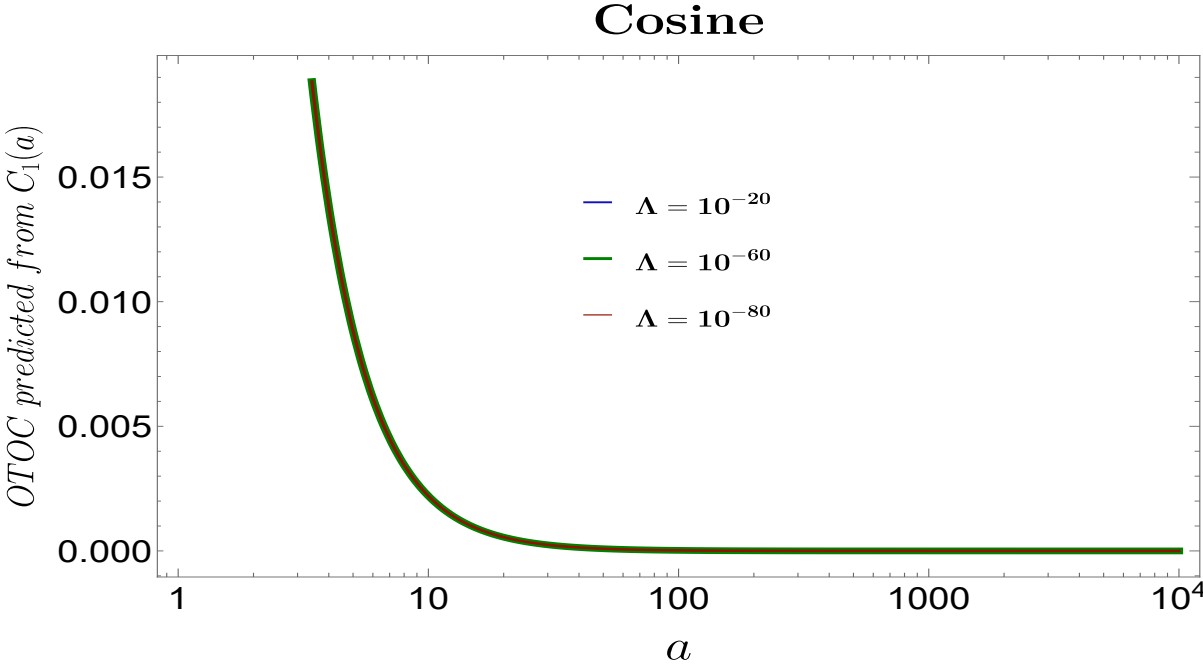

**Figure 15.** Behavior of OTOC predicted from $\mathcal{C}_1$ against scale factor in a different space.

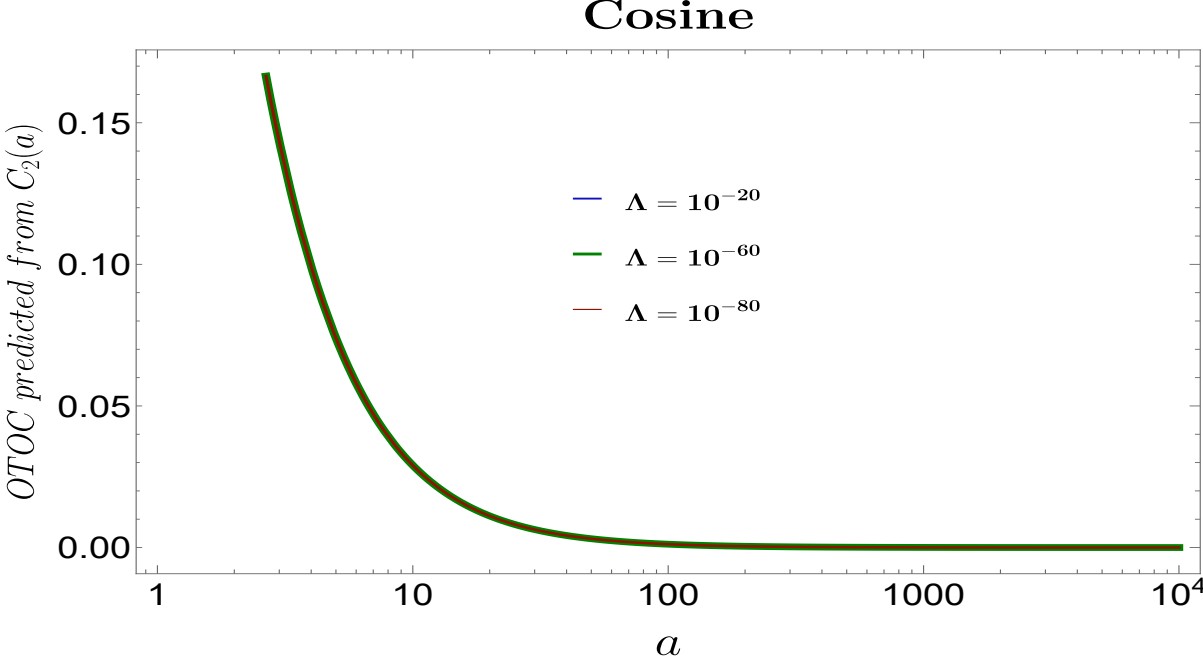

**Figure 16.** Behavior of OTOC predicted from $\mathcal{C}_2$ against scale factor in a different space.

### 7.2. No Islands in Recollapsing FLRW (Sine Hyperbolic Scale Factor)

In Figures 17 and 18 the squeezed state parameter $r_k$ and the squeezing angle $\phi_k$ are plotted with respect to the scale factor. The behavior of the squeezed state parameters determines the nature of complexity. In Figure 17 we see there are cut-off values of the scale factor, so we start observing deviations from the linear graph. Moreover, in Figure 18, we see an initial rise, followed by saturation and then a sharp upward deviation at particular values of the scale factor. We expect complexity measures also to experience similar deviation near particular values of the scale factor.

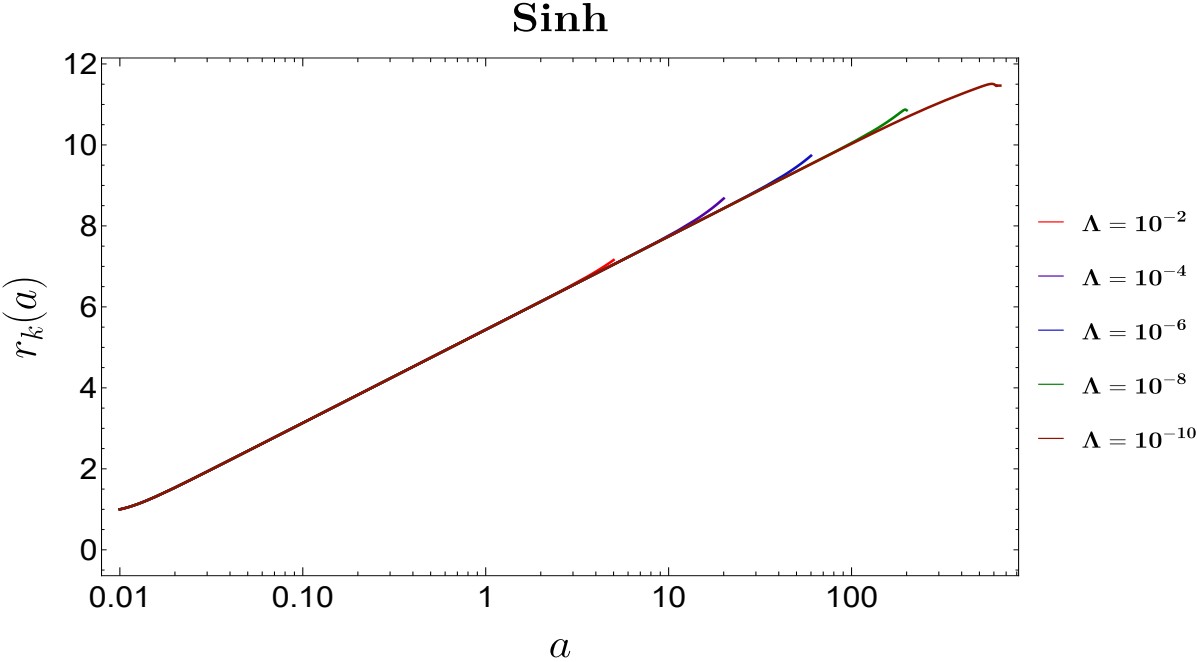

**Figure 17.** Squeezed state parameter $r_k$ plotted against scale factor.

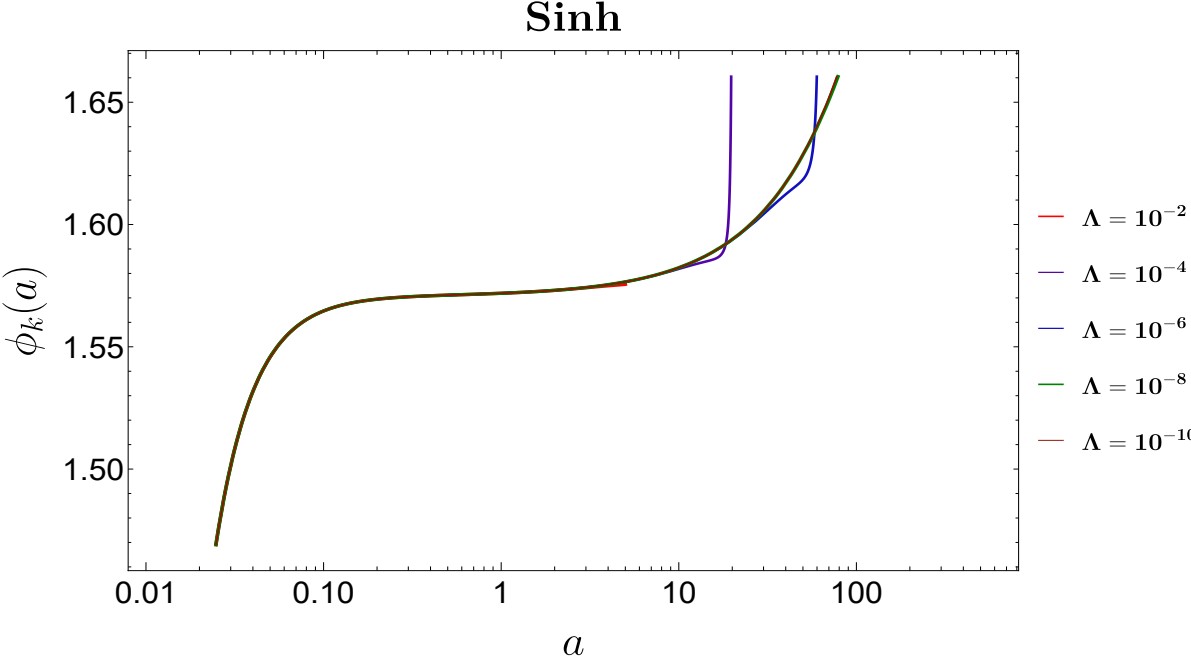

**Figure 18.** Squeezing angle $\phi_k$ plotted against scale factor.

- In Figures 19 and 20 the behavior of the circuit complexity computed from the linearly weighted and geodesically weighted cost functional is shown with respect to the scale factor. Although the overall behavior of the complexity measures is identical, some noticeable differences are mentioned below:

  - The complexity measure $\mathcal{C}_1$ (linearly weighted measure) is larger than $\mathcal{C}_2$ for the entire range of scale factor;
  - The peak in $\mathcal{C}_1$ is a non-uniform double peak, whereas for $\mathcal{C}_2$ this becomes a more uniform and smooth peak at the top;

-   The initial rise in $\mathcal{C}_1$ is more linear when compared to the initial rising part of $\mathcal{C}_2$. We also observe the rise begins a little later in the case for $\mathcal{C}_2$.

-   The general trend that we observe for the family of complexity values is that it initially rises, reaches a peak and then falls. The most peculiar difference is the deviation at particular values of scale factor for each cosmological constant. We observe some cut off values of the scale factor in this model. The values become unsolvable, signifying a blow-up or erratic behavior after a point.

-   Figures 21 and 22 shows the plots of *Out-of-Time-Ordered* correlation functions. Up to a certain value of scale factor, the OTOC decreases exponentially. However, after a certain transition scale factor, it starts increasing exponentially. Here too, we observe the deviations of the curve after a given value of scale factor for a chosen value of the cosmological constant.

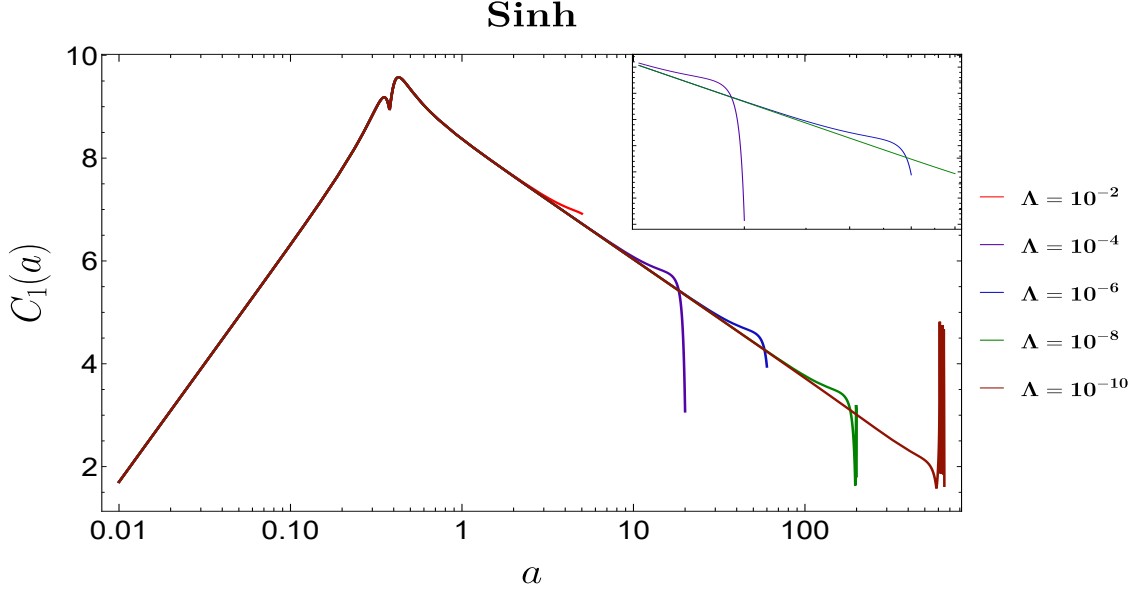

**Figure 19.** Linearly weighted complexity value plotted against scale factor.

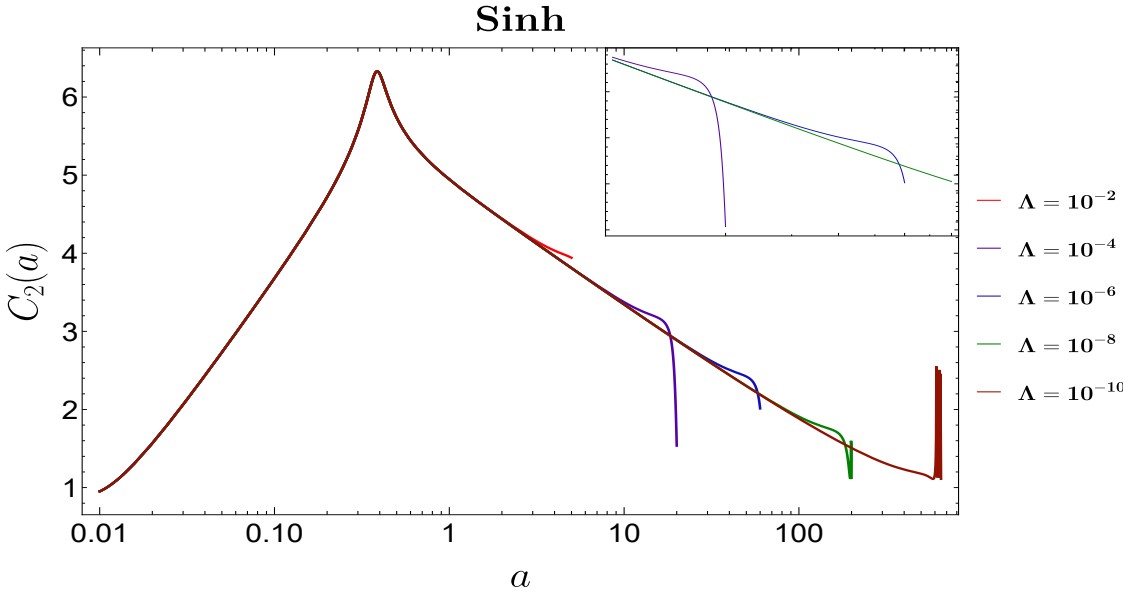

**Figure 20.** Geodesically weighted complexity value plotted against scale factor.

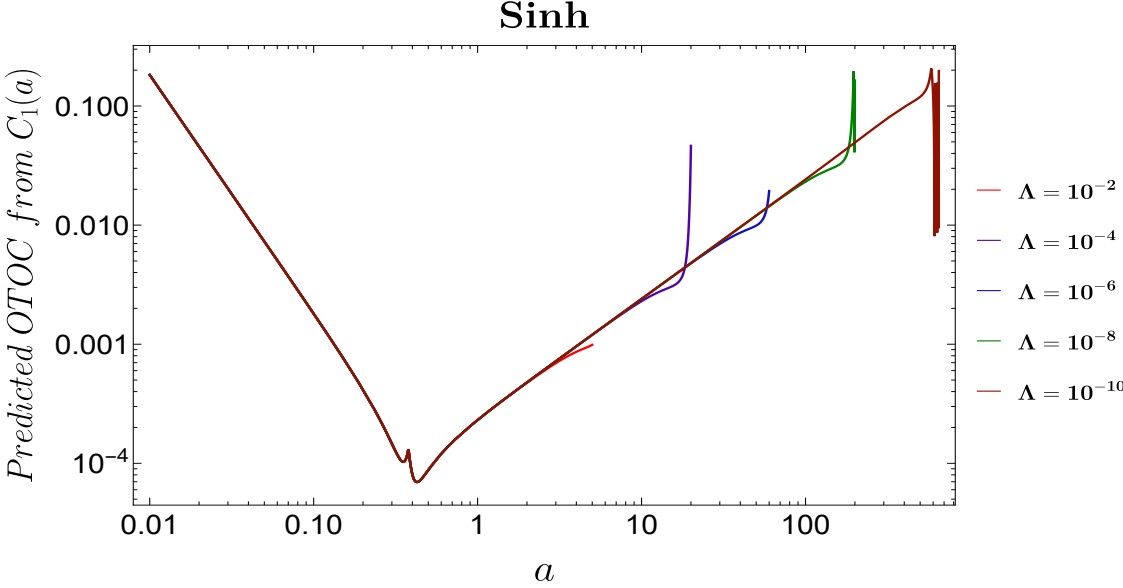

**Figure 21.** Predicted OTOC from linearly weighted cost functional plotted against scale factor.

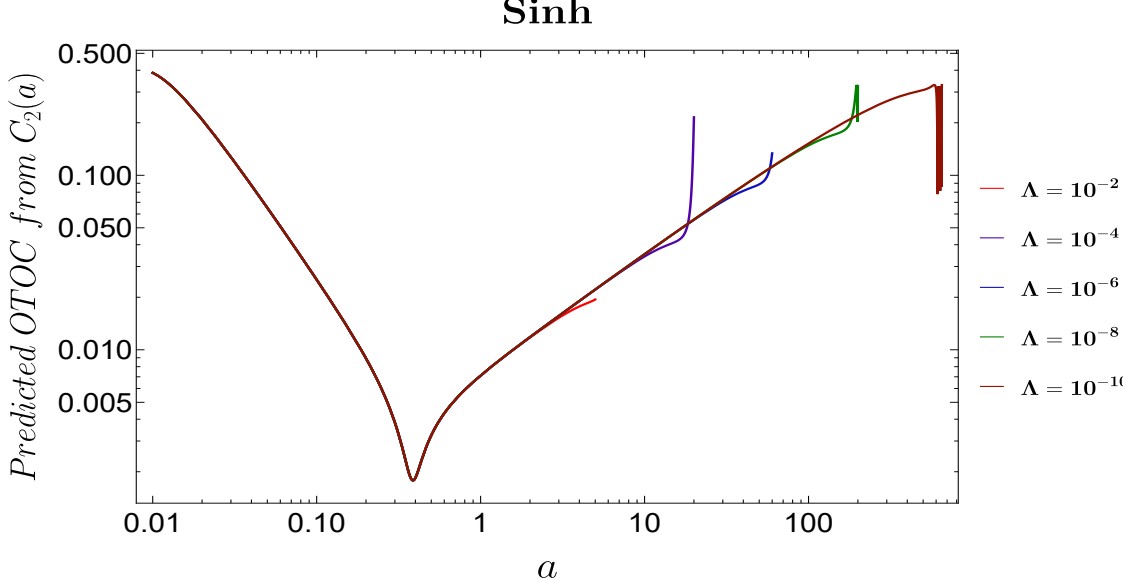

**Figure 22.** Predicted OTOC from geodesically weighted cost functional plotted against scale factor.

Table 2 shows the value of the Lyapunov exponents calculated for the initial exponentially rising region before the transition scale factor. The family of complexity curves follow a very similar trend during the rising portion. The estimation of the Lyapunov exponent from different complexity measures gives very similar values. There are slight differences in the values in the 3rd or 4th decimal places as we go to smaller values of the cosmological constant.

**Table 2.** Lyapunov exponenets calculated from the $\ln(\mathcal{C})$ vs. a plots for all the different chosen values of the cosmological constants.

| Complexity Measures | $\lambda_{\Lambda_1}$ | $\lambda_{\Lambda_2}$ | $\lambda_{\Lambda_3}$ | $\lambda_{\Lambda_4}$ | $\lambda_{\Lambda_5}$ |
|---|---|---|---|---|---|
| $\mathcal{C}_1$ | 6.017 | 6.01699 | 6.01699 | 6.01696 | 6.015 |
| $\mathcal{C}_2$ | 6.0611 | 6.0611 | 6.0611 | 6.0610 | 6.058 |

We observe deviations only after a few decimal places, and they are only significant in smaller values of the cosmological constant and higher cost functional family. This shows a higher sensitivity in smaller cosmological constant and higher-order cost functional.

- In Figures 23–25, we have plotted the behavior of the entanglement entropy, i.e., von-Neumann entanglement entropy and Renyi entropy between the two modes of the squeezed states. We again observe, the increasing behavior with the evolutionary time scales. However, for this model, it can be seen that for different values of the parameter (cosmological constant) the entanglement entropy can be probed up to different evolutionary scales. This is due to the existence of some cut-off values of the scale factor beyond which the squeezed parameters cannot be solved. This values of the evolutionary scales (cut off values) up to which the entanglement entropy can be probed is larger for smaller values of the cosmological constant;
- A similar increasing behavior followed by saturation is observed for the equilibrium temperature, as was seen for the cosine model case. However, the difference again lies in the existence of the cut off values of the scale factor for the sinh model beyond which for that particular parameter, one cannot probe the equilibrium temperature;
- In Figures 26 and 27 we have plotted the complexity measures for a different parameter space, i.e., choosing extremely small values of the cosmological constant. We see the behavior of the circuit complexity in this region is drastically different than that was observed in the earlier parameter space. For the earlier and intermediate part of the evolutionary scales, the circuit complexity measures shows a decreasing behavior while at large scales it just shows a random fluctuating behavior. This feature was not observed in the earlier parameter space;
- The behavior of the OTOCs in this parameter space is also remarkably different from the ones that we observed in the earlier parameter space. See Figures 28 and 29 for details. In this regime, the OTOC shows a slowly increasing behavior in the early evolutionary scales, followed by a sharp increase in the intermediate regions. However, at late time scales, the OTOC shows a similar fluctuating behavior as the circuit complexity. Another important feature is the absence of the cut off values of the evolutionary scales in this parameter space which was observed in the earlier case.

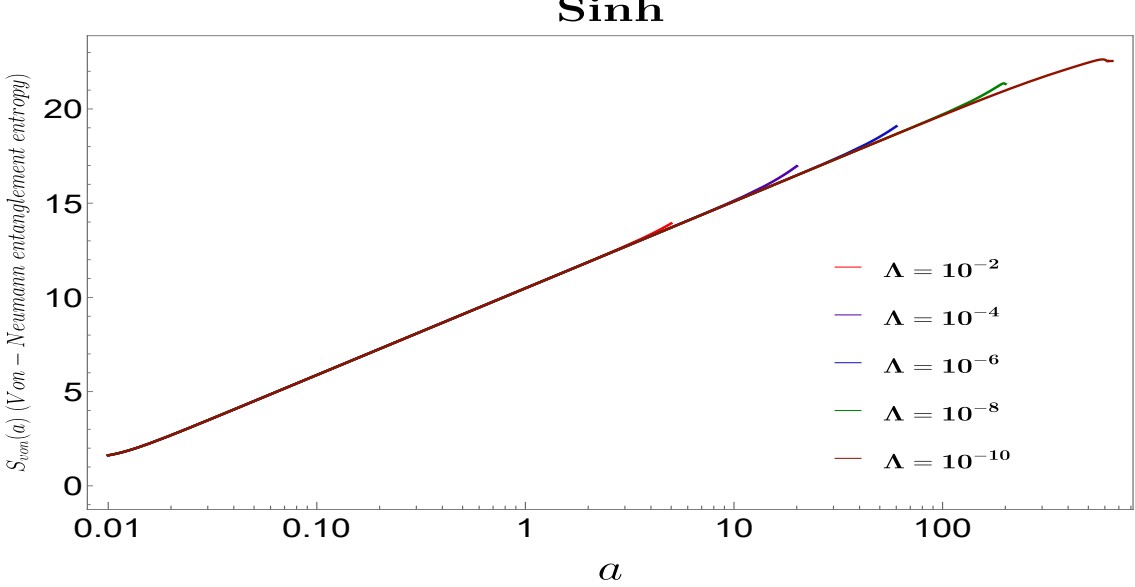

**Figure 23.** Von-Neumann entanglement entropy plotted as a function of the scale factor.

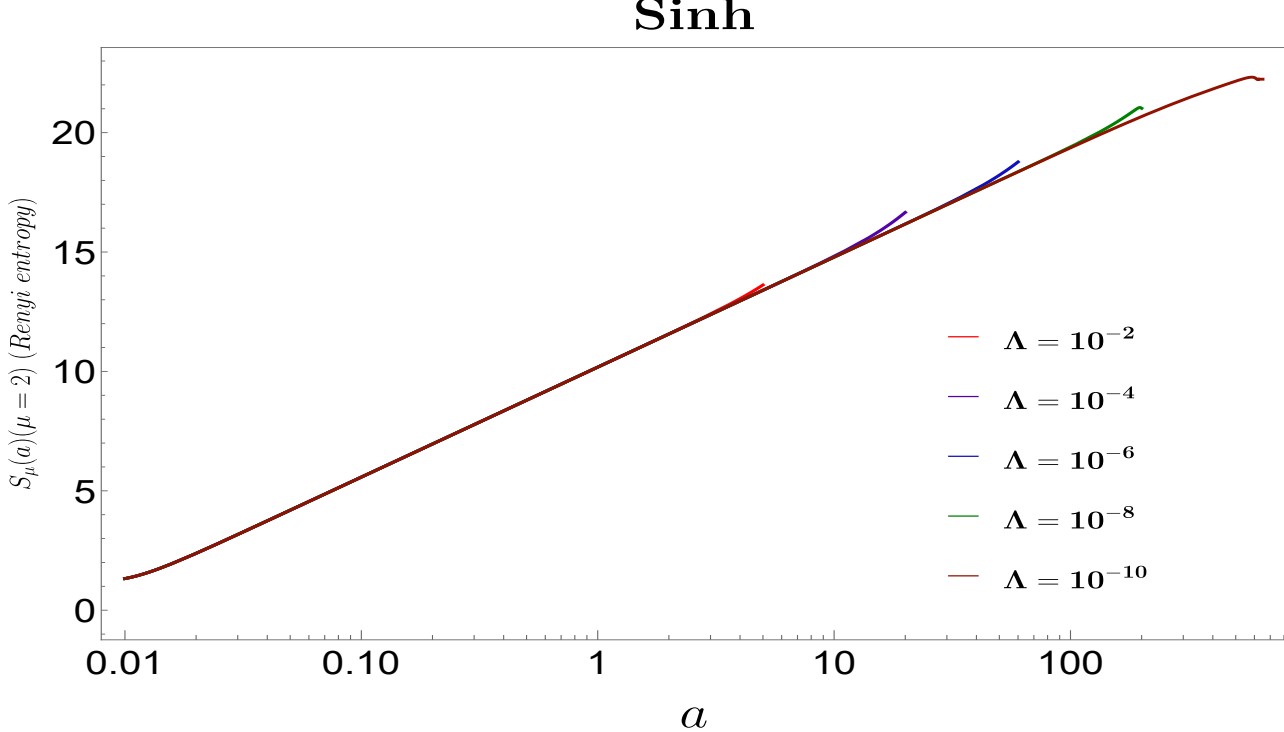

**Figure 24.** Renyi entanglement entropy plotted as a function of the scale factor.

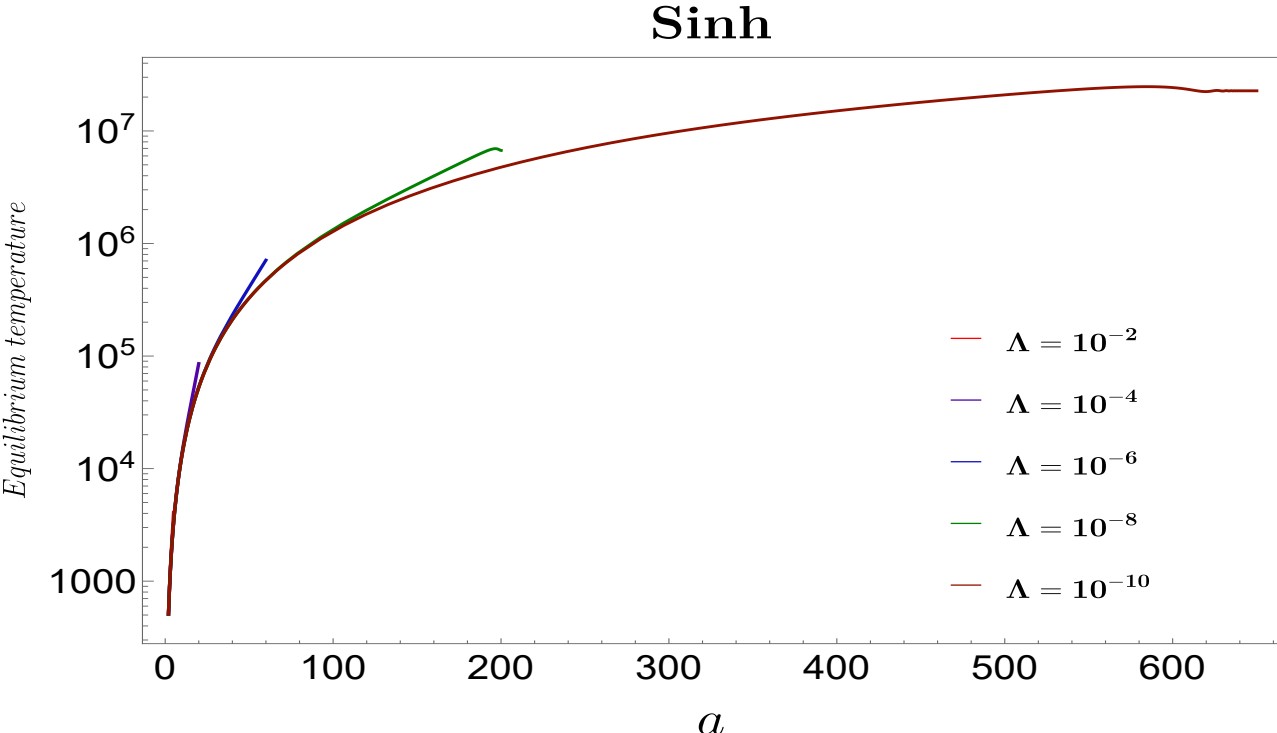

**Figure 25.** Entanglement entropy computed from $\mathcal{C}_2$ plotted against scale factor in the chaotic region.

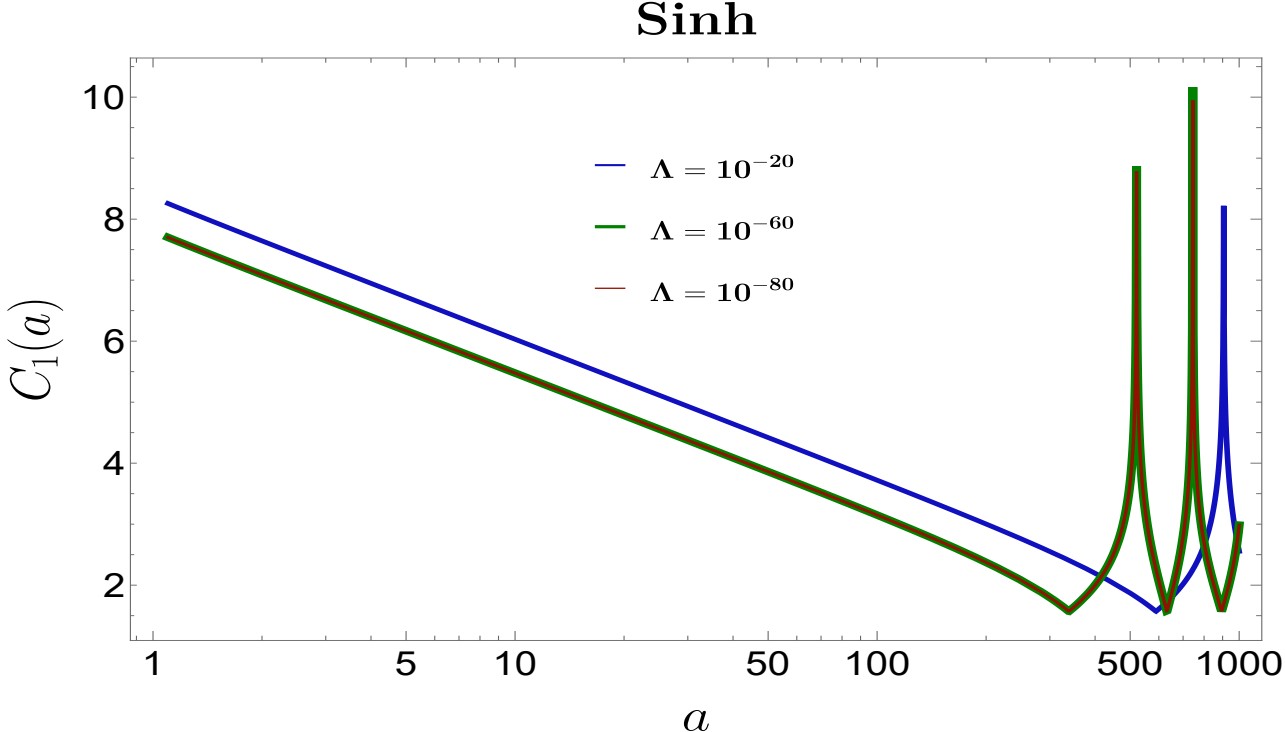

**Figure 26.** Behavior of $\mathcal{C}_1$ against scale factor in a different space.

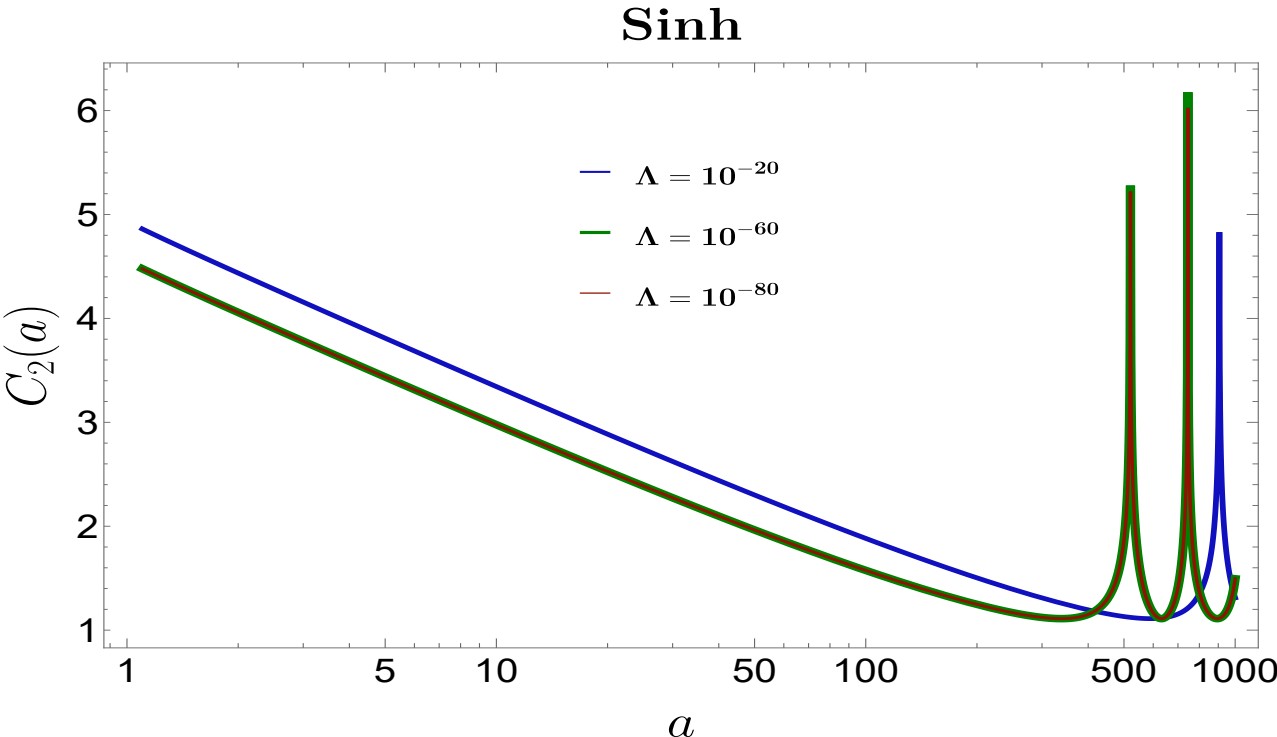

**Figure 27.** Behavior of $\mathcal{C}_2$ against scale factor in a different space.

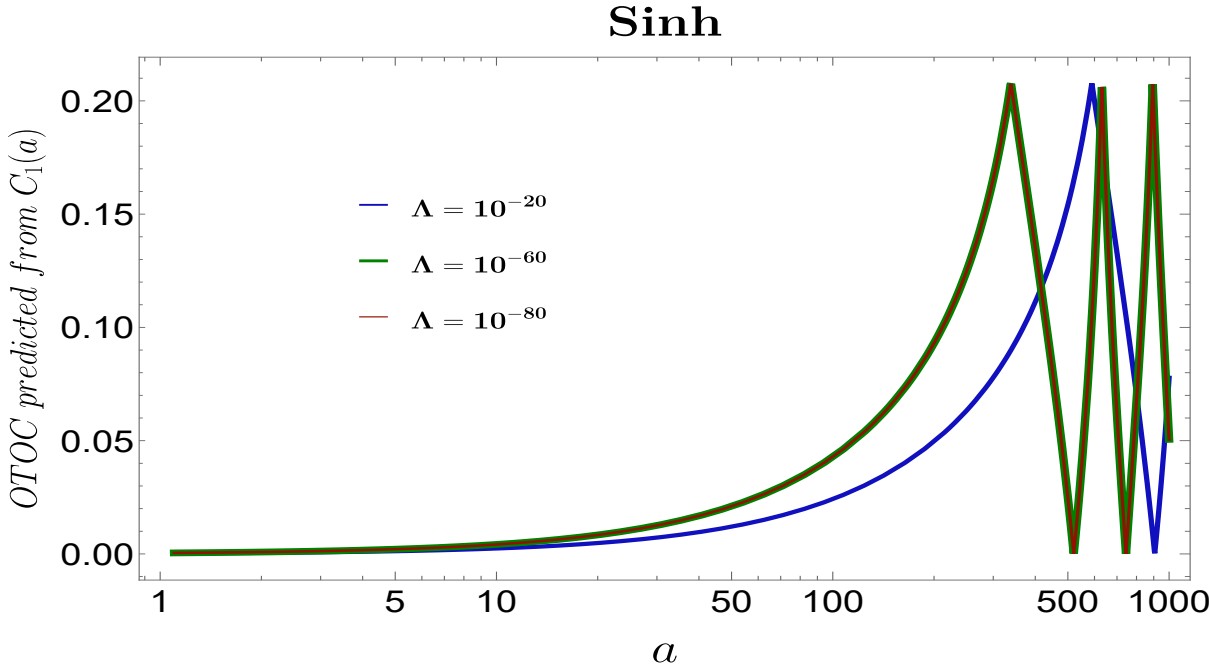

**Figure 28.** Behavior of OTOC predicted from $\mathcal{C}_1$ against scale factor in a different space.

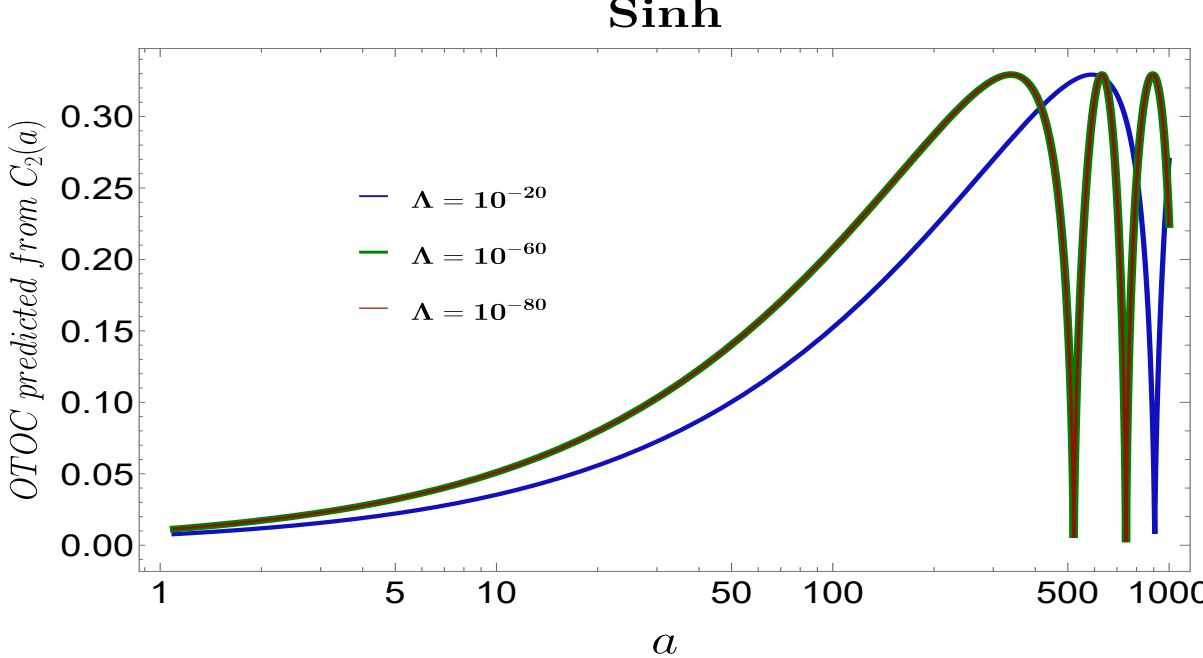

**Figure 29.** Behavior of OTOC predicted from $\mathcal{C}_2$ against scale factor in a different space.

In Table 3, a comparative analysis of the two models has been done in the two different parameter space chosen in the paper. The remarkable difference in the properties of the circuit complexity and OTOC in the two different parameter spaces shows the non-universality of the complexities and the OTOCs and suggests that they are dependent on the parameters of the chosen model and also on the regime of the evolutionary scale in which it is studied.

**Table 3.** Comparative study of the two models of scale factors in the two different parameter space chosen in the paper.

| Measures | Parameter Space I | Parameter Space II |
|---|---|---|
| **Values of Parameters** | High value of Cosmological constant, i.e., early time scale ($\leq 10^{-10}$) | extremely low value of cosmological constant, i.e., late time scale. ($10^{-20}$–$10^{-80}$) |
| **Complexity behavior of Cosine model** | increases and then decreases | always increases |
| **Complexity behavior of Sinh model** | increases and then decreases but with cut offs | decreases and then oscillates |
| **OTOC behavior of Cosine model** | decreases and then increases | decreases and then saturates |
| **OTOC behavior of Sinh model** | decreases and then increases with cut offs | increases and oscillates |

Table 4 summarizes all our important conclusions from the study of the two cosmological scale factors, one with island and the other without island.

**Table 4.** Comparative study of the two models of scale factors considered in this paper.

| Measures | AdS+Radiation FLRW | dS+Radiation FLRW |
|---|---|---|
| **Complexity plots in parameter space I** | exponential rise before a characteristic scale factor followed by a smooth decay | exponential rise before a characteristic scale factor followed by decay with cut offs. |
| **Complexity plots in parameter space II** | increasing behavior throughout | decreasing feature initially followed by random fluctuations. |
| **OTOC plots in parameter space I** | exponential decay observed before a characteristic scale factor followed by increasing behavior | exponential decay observed before a characteristic scale factor followed by increasing behavior with cut offs. |
| **OTOC plots in parameter space II** | decreasing feature initially followed by saturation | increasing feature initially followed by random fluctuations. |
| **Lyapunov exponent** | obeys the universal relation | obeys the universal relation |
| **Von-Neumann entanglement entropy of the modes of squeezed state** | increases throughout the evolutionary scales | increases throughout the evolutionary scales |
| **Renyi entropy** | increases throughout the evolutionary scales | increases throughout the evolutionary scales |
| **Equilibrium temperature of the squeezed state** | increases initially and saturates at late evolutionary scale | increases initially and saturates at late evolutionary scale |

## 8. Conclusions and Prospects

From our study of cosmological islands, we have the following final remarks:

- **Remark I:**
  The single field two mode squeezed state formalism enables us to express the various measures computed in this paper in terms of only two variables, the squeezed state parameter and the squeezed angle instead of adopting the general semi-classical approach. The squeezed state formalism approach provides an elegant way of comparing various measures calculated in this paper.
- **Remark II:**
  The notion of circuit complexity and OTOC can be used as a useful tool for elucidating many unknown aspects of gravitational and cosmological models. One can comment on the difference between the two cosmological models considered in this paper by computing the circuit complexity within the framework of spatially flat FLRW cosmology in the presence of quantum extremal islands, having AdS with radiation and dS with radiation.
- **Remark III:**
  In any chosen parameter space, the complexity behavior in spatially flat FLRW cosmology in the presence and absence of islands shows remarkably different features.
- **Remark IV:**
  The behavior of the out-of-time-ordered correlation functions are also drastically different for the two different cases considered in this paper.
- **Remark V:**
  Circuit complexity and OTOCs are universal in the entire region of the parameter space of the chosen model. This can be seen from the different behavior of the complexity measures for different ranges of the cosmological models.
- **Remark VI:**
  The quantum Lyapunov exponent and equilibrium temperature calculated from different complexity measures satisfy the universality relation established in Reference [20].
- **Remark VII:**
  The entropy of the modes of the squeezed states shows an increasing behavior for both the models with some minute differences, showing that the presence or absence of islands in FRW cosmology does not effect the entropy of the modes of the squeezed state. The equilibrium temperature of the two mode squeezed state also shows identical overall behavior irrespective of the presence or absence of islands.
- **Remark VIII:**
  In the Sine hyperbolic model (without islands), one can see the initial portion of the complexity curve resembling the cosine model (with islands). However, due to the deviation at different cut-off values of the scale factor, the behavior in the decreasing part at late evolutionary scales is not identical.

Some of the prospects can be in the following direction:

- **Prospect I:**
  As discussed earlier, apart from the quantum extremal surface or island prescription, other proposals have also been suggested to solve the black hole information paradox. However, none of them has been studied using the notion of circuit complexity and OTOC. A very intuitive study will be to try and predict the entanglement entropy from the computation of circuit complexity for the other proposals. One can then comment on the best proposal for reproducing the page curve from the perspective of circuit complexity and OTOC.
- **Prospect II:**
  It is a well-known fact that black holes are highly chaotic systems. One can then ask the question from the perspective of black hole chaos about which one is a better proposal in reproducing the page curve and revealing the chaotic features of black holes. This question can be addressed from the study of circuit complexity and OTOC, which are the most relevant probes of quantum chaos.
- **Prospect III:**
  An extension of the present work can be done for the primordial gravitational waves,

requiring the inclusion of the tensor mode fluctuations generated from cosmological perturbations in the spatially flat FLRW background rather than the scalar modes considered in this case. It would be an interesting study as to how the two-mode squeezed state formalism brings about the phenomenon of chaos and complexity in primordial gravitational waves.

- **Prospect IV:**
  A model-independent notion of circuit complexity can be given from the perspective of effective field theory, where one starts from a single EFT action and derives all models under various constraints satisfied by the action's parameters. Squeezed state formalism for such a universal action can be developed to generalize an give a model-independent prescription of complexity.

- **Prospect V:**
  Recently there has been a study of the Islands contribution in the entanglement negativity [125]. This motivates us to rethink various entanglement-related phenomena studied in [35,36,126,127] and whether those aspects can be studied from the perspective of islands.

- **Prospect VI:**
  Recently, there have been many studies in the field of open quantum systems (OQS) [128–130]. It is natural to expect that an OQS will exhibit chaotic behavior due to its constant interaction with its immediate surroundings. One can utilize the concept of circuit complexity and OTOC to probe the chaos shown by an OQS.

**Author Contributions:** Conceptualization, S.C. (Sayantan Choudhury); methodology, S.C. (Sayantan Choudhury); software, S.C. (Satyaki Chowdhury), N.G., A.M., S.P.S., G.D.P., C.S., A.S.; validation, S.C. (Sayantan Choudhury); formal analysis, S.C. (Sayantan Choudhury), and all other authors of the paper; investigation, S.C. (Sayantan Choudhury); resources, S.C. (Sayantan Choudhury); writing—original draft preparation, S.C. (Sayantan Choudhury), S.C. (Satyaki Chowdhury), N.G., A.S.; writing—review and editing, S.C. (Sayantan Choudhury); visualization, S.C. (Sayantan Choudhury); supervision, S.C. (Sayantan Choudhury); project administration, S.C. (Sayantan Choudhury) and S.P.; funding acquisition, S.C. (Sayantan Choudhury) and S.P. All authors have read and agreed to the published version of the manuscript.

**Funding:** This research received no external funding.

**Institutional Review Board Statement:** Not applicable.

**Informed Consent Statement:** Not applicable.

**Data Availability Statement:** No data is involved or explicitly used for this work.

**Acknowledgments:** The research fellowship of S.C. is supported by the J. C. Bose National Fellowship of Sudhakar Panda. Additionally, S.C. takes this opportunity to sincerely thank Sudhakar Panda for his constant support and huge inspiration. S.C. also would like to thank the School of Physical Sciences, National Institute for Science Education and Research (NISER), Bhubaneswar, for providing a work-friendly environment. S.C. also thank all the members of our newly formed virtual international non-profit consortium "Quantum Structures of the Space-Time and Matter" (QASTM), for elaborative discussions. S.C. also would like to thank all the speakers of the QASTM zoominar series from different parts of the world (For the uploaded YouTube link, look at https://www.youtube.com/playlist?list=PLzW8AJcryManrTsG-4U4z9ip1J1dWoNgd, accessed on 31 March 2021) for supporting my research forum by giving outstanding lectures and their valuable time during this COVID pandemic time. Satyaki Choudhury, Nitin Gupta, Anurag Mishra, Sachin Panner Selvam, Gabriel D. Pasquino Ciranjeeb Singha, and Abinash Swain would like to thank NISER Bhubaneswar, NIT Rourkela, BITS Hyderabad, University of Waterloo, IISER Kolkata, CMI Chennai, respectively, for providing fellowships. S.P. acknowledges the J. C. Bose National Fellowship for support of his research. Finally, we would like to acknowledge our debt to the people belonging to the various part of the world for their generous and steady support for research in natural sciences.

**Conflicts of Interest:** The authors declare no conflict of interest.

**Appendix A. Horizon Constraints on the FLRW Cosmological Islands**

In Section 5.4 we have imposed the constraint $k\tau_0 = -1$. It is important to note what constraint this places on the scale factor, when we consider the cosmological horizon for islands.

The cosmological horizon is given by

$$\mathcal{D}_H := \frac{k}{a(t)H(t)} = \frac{k}{\mathcal{H}(\tau)} = 1, \quad \text{where} \quad \mathcal{H}(\tau) = \frac{d \ln a(\tau)}{d\tau} = \frac{a'(\tau)}{a(\tau)}. \tag{A1}$$

This along with the constraint gives us,

$$\frac{1}{\tau \mathcal{H}} = -1 \tag{A2}$$

**Model I (AdS+Radiation):**

$$-\text{EllipticF}\left[\frac{1}{2}\cos^{-1}\left(\frac{a^2}{a_0^2}\right), 2\right] \text{JacobiDN}\left[\text{EllipticF}\left[\frac{1}{2}\cos^{-1}\left(\frac{a^2}{a_0^2}\right), 2\right], 2\right]$$
$$\tan\left(\text{JacobiDN}\left[\text{EllipticF}\left[\frac{1}{2}\cos^{-1}\left(\frac{a^2}{a_0^2}\right), 2\right], 2\right]\right) = -1 \tag{A3}$$

The expression can be calculated for both models by using the expression for $\tau$ and $\mathcal{H}$ that we already calculated.

**Model II (dS+Radiation):**

$$-\sqrt{t_m}\left(\frac{1-\iota}{2}\right)\left((1+i)\sqrt{\frac{1}{t_m}}\text{EllipticF}\left[\frac{1}{2}\cos^{-1}\left(\frac{ia^2}{a_0^4}\right), 2\right] - \sqrt{\frac{1}{t_m}}\text{EllipticK}\left[\frac{1}{2}\right]\right)$$
$$\text{JacobiDN}\left[\text{EllipticF}\left[\frac{1}{2}\cos^{-1}\left(\frac{ia^2}{a_0^4}\right), 2\right], 2\right] \tag{A4}$$
$$\tan\left(2\text{JacobiAmplitude}\left[\text{EllipticF}\left[\frac{1}{2}\cos^{-1}\left(\frac{ia^2}{a_0^4}\right), 2\right], 2\right]\right) = -1$$

For a given range of scale factor, one can then determine whether we are probing inside or outside the event horizon of the space-time based on the cosmological horizon condition.

**Appendix B. Dispersion Relation in Cosmological Islands**

In this appendix, our prime objective is to derive the expression for the dispersion relation in terms of the squeezed parameter $r_{\mathbf{k}}(\tau)$ and the squeezed angle $\phi_{\mathbf{k}}(\tau)$, where the dispersion relation appears in the Hamiltonian after quantization that we studied in the paper explicitly.

Let us first write down the expression for the conformal time dependent dispersion relation $\Omega_{\mathbf{k}}$ in terms of the canonical field variable and its associated canonically conjugate momentum that appears after performing the cosmological perturbation theory for a single scalar field:

$$\Omega_{\mathbf{k}}(\tau) := \left\{ \left|v'_{\mathbf{k}}(\tau)\right|^2 + \mu^2(k,\tau)|v_{\mathbf{k}}(\tau)|^2 \right\}$$
$$= \left\{ |\pi_{\mathbf{k}}(\tau)|^2 + k^2|v_{\mathbf{k}}(\tau)|^2 + \lambda_{\mathbf{k}}(\tau)\left(\pi_{\mathbf{k}}^*(\tau)v_{\mathbf{k}}(\tau) + v_{\mathbf{k}}^*(\tau)\pi_{\mathbf{k}}(\tau)\right) \right\},$$

Now, we plug in the expressions for $\pi_{\mathbf{k}}(\tau)$ and $v_{\mathbf{k}}(\tau)$, which are reproduced here for convenience:

$$v_{\mathbf{k}}(\tau) = v_{\mathbf{k}}(\tau_0)\bigg(\cosh r_{\mathbf{k}}(\tau)\,\exp(i\theta_{\mathbf{k}}(\tau)) - \sinh r_{\mathbf{k}}(\tau)\,\exp(i(\theta_{\mathbf{k}}(\tau) + 2\phi_{\mathbf{k}}(\tau)))\bigg), \tag{A5}$$

$$\pi_{\mathbf{k}}(\tau) = \pi_{\mathbf{k}}(\tau_0)\bigg(\cosh r_{\mathbf{k}}(\tau)\,\exp(i\theta_{\mathbf{k}}(\tau)) + \sinh r_{\mathbf{k}}(\tau)\,\exp(i(\theta_{\mathbf{k}}(\tau) + 2\phi_{\mathbf{k}}(\tau)))\bigg), \tag{A6}$$

and after doing algebraic manipulation we get the following result:

$$
\begin{aligned}
\Omega_{\mathbf{k}}(\tau) = {}& \bigg(|\pi_{\mathbf{k}}(\tau_0)|^2 + k^2|v_{\mathbf{k}}(\tau_0)|^2\bigg)\bigg(\cosh^2 r_{\mathbf{k}}(\tau) + \sinh^2 r_{\mathbf{k}}(\tau)\bigg) \\
&+ \sinh r_{\mathbf{k}}(\tau)\cdot\cos 2\phi_{\mathbf{k}}(\tau)\bigg(|\pi_{\mathbf{k}}(\tau_0)|^2 - k^2|v_{\mathbf{k}}(\tau_0)|^2\bigg) \\
&+ \lambda_{\mathbf{k}}(\tau)\bigg\{\bigg(\pi_{\mathbf{k}}^*(\tau_0)v_{\mathbf{k}}(\tau_0) + v_{\mathbf{k}}^*(\tau_0)\pi_{\mathbf{k}}(\tau_0)\bigg) \\
&+ i\sinh 2r_{\mathbf{k}}(\tau)\sin 2\phi_{\mathbf{k}}(\tau)\bigg(\pi_{\mathbf{k}}^*(\tau_0)v_{\mathbf{k}}(\tau_0) - v_{\mathbf{k}}^*(\tau_0)\pi_{\mathbf{k}}(\tau_0)\bigg)\bigg\}.
\end{aligned}
\tag{A7}
$$

Here we have chosen the initial condition at the time scale $\tau = \tau_0$ by considering, $-k\tau_0 = 1$. We impose this condition on the perturbation field variable and on the canonically conjugate momentum obtained for scalar fluctuation. We finally get:

$$v_{\mathbf{k}}(\tau_0) = \frac{1}{\sqrt{2k}}\,2^{\nu_{\text{island}}-1}\left|\frac{\Gamma(\nu_{\text{island}})}{\Gamma(\frac{3}{2})}\right|\exp\left(-i\left\{\frac{\pi}{2}(\nu_{\text{island}} - 2) - 1\right\}\right), \tag{A8}$$

$$
\begin{aligned}
\pi_{\mathbf{k}}(\tau_0) = {}& i\sqrt{\frac{k}{2}}\,2^{\nu_{\text{island}}-\frac{3}{2}}\left|\frac{\Gamma(\nu_{\text{island}})}{\Gamma(\frac{3}{2})}\right|\exp\left(-i\left\{\frac{\pi}{2}(\nu_{\text{island}} - 2) - 1\right\}\right) \\
&\left[1 - \sqrt{2}\frac{\left(\nu_{\text{island}} - \frac{1}{2}\right)\left(\nu_{\text{island}} + \frac{1}{2} + i\right)}{\left(\nu_{\text{island}} + \frac{1}{2}\right)}\exp\left(-\frac{i\pi}{4}\right)\right].
\end{aligned}
\tag{A9}
$$

the general mass parameter for cosmological Islands can be computed as:

$$\nu_{\text{island}} = \sqrt{\frac{1}{4} + \frac{2(1 - w_{\text{eff}})}{(1 + 3w_{\text{eff}})^2}}. \tag{A10}$$

where the effective equation of state parameter $w_{\text{eff}}$ is defined for the two prescribed models as:

**AdS FLRW + Radiation:** $\quad w_{\text{eff}} = \frac{1}{3}\left[\frac{\left(1 + \frac{3|\Lambda|}{16\pi\epsilon_0\rho_0}\right)}{\left(1 - \frac{|\Lambda|}{16\pi\epsilon_0\rho_0}\right)}\right] = \frac{1}{3}\left[\frac{\left(1 + 3\left(\frac{a}{a_0}\right)^4\right)}{\left(1 - \left(\frac{a}{a_0}\right)^4\right)}\right], \tag{A11}$

**dS FLRW + Radiation:** $\quad w_{\text{eff}} = \frac{1}{3}\left[\frac{\left(1 - \frac{3|\Lambda|}{16\pi\epsilon_0\rho_0}\right)}{\left(1 + \frac{|\Lambda|}{16\pi\epsilon_0\rho_0}\right)}\right] = \frac{1}{3}\left[\frac{\left(1 - 3\left(\frac{a}{a_0}\right)^4\right)}{\left(1 + \left(\frac{a}{a_0}\right)^4\right)}\right]. \tag{A12}$

where we use the fact that the radiation dominated epoch the radiation density scales with the scale factor as, $\rho = \rho_0 a^{-4}$ where $a_0$ is given by:

$$a_0 = a(t=0) = \left(\frac{16\pi\epsilon_0\rho_0}{|\Lambda|}\right)^{1/4} = \left(\frac{8\pi\epsilon_0}{|\Lambda|}\right)^{1/4}, \quad \text{where we fix} \quad \rho_0 = \frac{1}{2}. \tag{A13}$$

Further, one can recast the expression for the generalized mass parameter for the mentioned two models in the following simplified and compact form:

$$\underline{\textbf{AdS FLRW + Radiation:}} \qquad \nu_{\text{island}}(a) = \frac{1}{2}\sqrt{1 + \Delta_{\text{AdS}}(a)}, \tag{A14}$$

$$\underline{\textbf{dS FLRW + Radiation:}} \qquad \nu_{\text{island}}(a) = \frac{1}{2}\sqrt{1 + \Delta_{\text{dS}}(a)}. \tag{A15}$$

where the newly introduced scale factor dependent factors, $\Delta_{\text{AdS}}$ and $\Delta_{\text{dS}}$ are defined as follows:

$$\Delta_{\text{AdS}}(a) := \frac{8\left(1 - \frac{1}{3}\left[\frac{\left(1 + 3\left(\frac{a}{a_0}\right)^4\right)}{\left(1 - \left(\frac{a}{a_0}\right)^4\right)}\right]\right)}{\left(1 + \left[\frac{\left(1 + 3\left(\frac{a}{a_0}\right)^4\right)}{\left(1 - \left(\frac{a}{a_0}\right)^4\right)}\right]\right)^2}, \tag{A16}$$

$$\Delta_{\text{dS}}(a) := \frac{8\left(1 - \frac{1}{3}\left[\frac{\left(1 - 3\left(\frac{a}{a_0}\right)^4\right)}{\left(1 + \left(\frac{a}{a_0}\right)^4\right)}\right]\right)}{\left(1 + \left[\frac{\left(1 - 3\left(\frac{a}{a_0}\right)^4\right)}{\left(1 + \left(\frac{a}{a_0}\right)^4\right)}\right]\right)^2}. \tag{A17}$$

Neglecting the phase contributions, we get a very simplified expression for $\Omega_{\mathbf{k}}(\tau)$, which is given by:

$$\Omega_{\mathbf{k}}(\tau) = 2^{2\nu_{\text{island}}-2}\left|\frac{\Gamma(\nu_{\text{island}})}{\Gamma\left(\frac{3}{2}\right)}\right|^2\left[\frac{3k}{4}\left(\cosh^2 r_{\mathbf{k}}(\tau) + \sinh^2 r_{\mathbf{k}}(\tau)\right) - \frac{k}{4}\sinh r_{\mathbf{k}}(\tau)\cos 2\phi_{\mathbf{k}}(\tau)\right.$$
$$\left. - \frac{1}{\sqrt{2}}\lambda_{\mathbf{k}}(\tau)\ \sinh 2r_{\mathbf{k}}(\tau)\sin 2\phi_{\mathbf{k}}(\tau)\right]. \tag{A18}$$

Now we consider a specific situation in the time line of our FLRW universe, where it is expected to have very small contribution from the squeezed parameter, $r_{\mathbf{k}}(\tau)$ for which one can use the following approximations:

$$\cosh r_{\mathbf{k}}(\tau) \approx 1, \quad \sinh r_{\mathbf{k}}(\tau) \approx r_{\mathbf{k}}(\tau). \tag{A19}$$

Consequently, we get the following result for the island dispersion relation:

$$\Omega_{\mathbf{k}}(\tau) \approx \underbrace{3k\, 2^{2(\nu_{\text{island}}-2)} \left| \frac{\Gamma(\nu_{\text{island}})}{\Gamma(\frac{3}{2})} \right|^2}_{\textbf{Leading contribution}} \left(1 + r_{\mathbf{k}}^2(\tau) + \cdots \right), \tag{A20}$$

which is basically dependent on the co-moving wave number and the time dependent quantity $\nu_{\text{island}}$. Further, if we assume that the contributions appearing through the factors $\Delta_{\text{AdS}}(a)$ and $\Delta_{\text{dS}}(a)$ are appearing as a correction terms due to its smallness the by applying the binomial approximation the conformal time dependent generalized mass parameter $\nu_{\text{island}}$ can be approximately written by considering the contribution up to the next-to-leading order term as:

$$\underline{\textbf{AdS FLRW + Radiation:}} \quad \nu_{\text{island}}(a) \approx \left( \frac{1}{2} + \frac{1}{4}\Delta_{\text{AdS}}(a) + \cdots \right), \tag{A21}$$

$$\underline{\textbf{dS FLRW + Radiation:}} \quad \nu_{\text{island}}(a) \approx \left( \frac{1}{2} + \frac{1}{4}\Delta_{\text{dS}}(a) + \cdots \right). \tag{A22}$$

The similar approximation can also be realised in terms of the effective equation of state parameter as well, which can written as:

$$\nu_{\text{island}} \approx \left( \frac{1}{2} + \frac{4(1-w_{\text{eff}})}{(1+3w_{\text{eff}})^2} + \cdots \right). \tag{A23}$$

where we have neglected the contributions of all higher order small correction terms appearing as $\cdots$ from AdS+radiation and dS+radiation sectors, respectively. Now after substituting the above mentioned expression for the mass parameter $\nu_{\text{island}}$ one can further write the following simplified form of the dispersion relation, $\Omega_{\mathbf{k}}(\tau)$, which is given by:

$$
\begin{aligned}
\Omega_{\mathbf{k}}(\tau) &\approx \frac{3}{2}\, k\, 2^{\left( \frac{8(1-w_{\text{eff}})}{(1+3w_{\text{eff}})^2} + \cdots \right)} \left| \frac{\Gamma\left( \frac{1}{2} + \frac{4(1-w_{\text{eff}})}{(1+3w_{\text{eff}})^2} + \cdots \right)}{\Gamma\left( \frac{1}{2} \right)} \right|^2 \left( 1 + r_{\mathbf{k}}^2(\tau) + \cdots \right) \\
&= \frac{3}{2\pi}\, k\, 2^{\left( \frac{8(1-w_{\text{eff}})}{(1+3w_{\text{eff}})^2} + \cdots \right)} \left| \Gamma\left( \frac{1}{2} + \frac{4(1-w_{\text{eff}})}{(1+3w_{\text{eff}})^2} + \cdots \right) \right|^2 \left( 1 + r_{\mathbf{k}}^2(\tau) + \cdots \right) \\
&\approx \frac{3}{2}\, k \left( 1 + 8\ln 2\, \frac{(1-w_{\text{eff}})}{(1+3w_{\text{eff}})^2} + \cdots \right) \left[ 1 + \frac{8(1-w_{\text{eff}})}{(1+3w_{\text{eff}})^2} \psi^{(0)}\left( \frac{1}{2} \right) + \cdots \right] \left( 1 + r_{\mathbf{k}}^2(\tau) + \cdots \right) \\
&\approx \frac{3}{2}\, k \left[ 1 + \frac{8(1-w_{\text{eff}})}{(1+3w_{\text{eff}})^2} \left( \ln 2 + \psi^{(0)}\left( \frac{1}{2} \right) \right) + \cdots \right] \left( 1 + r_{\mathbf{k}}^2(\tau) + \cdots \right).
\end{aligned}
\tag{A24}
$$

Here for the above computation we have used the following important results for the series expansion:

$$2^{\left( \frac{8(1-w_{\text{eff}})}{(1+3w_{\text{eff}})^2} + \cdots \right)} = \left( 1 + 8\ln 2\, \frac{(1-w_{\text{eff}})}{(1+3w_{\text{eff}})^2} + \cdots \right), \tag{A25}$$

$$\left| \Gamma\left( \frac{1}{2} + \frac{4(1-w_{\text{eff}})}{(1+3w_{\text{eff}})^2} + \cdots \right) \right|^2 = \pi \left[ 1 + \frac{8(1-w_{\text{eff}})}{(1+3w_{\text{eff}})^2} \psi^{(0)}\left( \frac{1}{2} \right) + \cdots \right]. \tag{A26}$$

Now after substituting the above mentioned expression for the mass parameter $\nu_{\text{island}}$ one can further write the following simplified form of the dispersion relation, $\Omega_{\mathbf{k}}(a)$, in terms of the FLRW scale factor for AdS+radiation and dS+radiation are given by the following expressions:

$$
\begin{aligned}
\Omega_{\mathbf{k}}(a) \;\;\approx\;\; & \frac{3}{2}\, k\, 2^{\left(\frac{1}{4}\Delta_{\mathrm{AdS/dS}}(a)+\cdots\right)} \left|\frac{\Gamma\left(\frac{1}{2}+\frac{1}{4}\Delta_{\mathrm{AdS/dS}}(a)+\cdots\right)}{\Gamma\left(\frac{1}{2}\right)}\right|^{2}\left(1+r_{\mathbf{k}}^{2}(\tau)+\cdots\right) \\[2mm]
=\;\; & \frac{3}{2\pi}\, k\, 2^{\left(\frac{1}{4}\Delta_{\mathrm{AdS/dS}}(a)+\cdots\right)}\left|\Gamma\left(\frac{1}{2}+\frac{1}{4}\Delta_{\mathrm{AdS/dS}}(a)+\cdots\right)\right|^{2}\left(1+r_{\mathbf{k}}^{2}(\tau)+\cdots\right) \qquad\text{(A27)} \\[2mm]
\approx\;\; & \frac{3}{2}\, k\, \left(1+\frac{1}{4}\Delta_{\mathrm{AdS/dS}}(a)+\cdots\right)\left[1+\frac{1}{4}\Delta_{\mathrm{AdS/dS}}(a)\psi^{(0)}\left(\frac{1}{2}\right)+\cdots\right]\left(1+r_{\mathbf{k}}^{2}(\tau)+\cdots\right) \\[2mm]
\approx\;\; & \frac{3}{2}\, k\, \left[1+\frac{1}{4}\Delta_{\mathrm{AdS/dS}}(a)\left(\ln 2+\psi^{(0)}\left(\frac{1}{2}\right)\right)+\cdots\right]\left(1+r_{\mathbf{k}}^{2}(\tau)+\cdots\right).
\end{aligned}
$$

Here for the above computation we have used the following important results for the series expansion:

$$
2^{\left(\frac{1}{4}\Delta_{\mathrm{AdS/dS}}(a)+\cdots\right)} \;=\; \left(1+\frac{1}{4}\ln 2\,\Delta_{\mathrm{AdS/dS}}(a)+\cdots\right), \qquad\text{(A28)}
$$

$$
\left|\Gamma\left(\frac{1}{2}+\frac{1}{4}\Delta_{\mathrm{AdS/dS}}(a)+\cdots\right)\right|^{2} \;=\; \pi\left[1+\frac{1}{4}\Delta_{\mathrm{AdS/dS}}(a)\psi^{(0)}\left(\frac{1}{2}\right)+\cdots\right]. \qquad\text{(A29)}
$$

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
