# Peer review of "Circuit Complexity from Cosmological Islands"

_symmetry, doi:10.3390/sym13071301_

Round 1

Reviewer 1 Report

This is an interesting paper that considers "CIRCUIT COMPLEXITY FROM COSMOLOGICAL ISLANDS" and discuss within the Quantum Extremal Islands approach, using the methods of Out-of-Time Ordered Correlators, and entanglement entropy of the modes of squeezed states. The results are interesting and deserve publication. I would just make a small comment, that the authors may take into consideration, the authors discuss the Jackiw − Teitelboim (JT) gravity : setting  K(Ď•) = 0, V (Ď•) = −λ^2,  in their equation (2.3), this is just an example of the modified measure theory, where one uses a lagrangian density L= R, and S =  integral Phi L,

where Phi is a metric ndependent measure build out "measure fields" independent of metric, this density is a total derivative and the equation of motion of the measure field yields R= constant,  see for example The Principle of nongravitating vacuum energy and some of its consequences, E.I. Guendelman, A.B. Kaganovich, Phys.Rev.D 53 (1996) 7020-7025 • e-Print: gr-qc/9605026 [gr-qc] , Scale invariance, new inflation and decaying lambda terms ,E.I. Guendelman, Mod.Phys.Lett.A 14 (1999) 1043-1052 • e-Print: gr-qc/9901017 [gr-qc] and others. The paper can be published and the authors may consider citing these papers .

Author Response

Response to referee 1 is attached as a .pdf file.

Reviewer 2 Report

This ms is the continuation of serie of works written by (sub)groups of current authors. At least two of them were published in Symmetry. Hence, what is the novelty of this specific paper if compare with previous publications of same authors?

It is totally unclear the motivation of this ms. It is not described well in the introduction. OK about islands and entropy using this formulation for BHs.

However, in this paper the authors discuss cosmology (even cosmological islands) using same quantum mechanics they try to develop!

It looks they are not aware about QFT in curved spacetime (for general review, see monograph Buchbinder, Odintsov,Shapiro, effective Action in Quantum Gravity, IOP 1992). The QFT calculation of effective action of any theory (scalar, electrodynamics or gravity) in dS universe and subsequent entropy calculation  maybe done in full QFT without any problems, for boundary conditions one wishes to select. It is rather understandable that method developed in this ms is just over-simplified approach to the problem. So it should be clearly explained why the authors do not wish or cannot use QFT in FRW universe for their purposes. The comparison with such effective action calculation in dS or AdS universe and results of this ms should be done in detail.

This is major point.

Author Response

Response to referee 2 is attached as .pdf file.

Round 2

Reviewer 2 Report

Unfortunately, the authors did not really address the referee report. There is no clear explanation why this approach (oversimplified!) is preferrable if compare with standard QFT in FRW universe. All results the authors discuss can be obtained in QFT, using Effective Action or some other approach.

So i invite the authors to rewrite their introduction and conclusion and clearly answer to the points suggested in my first report. Otherwise, it looks that authors only aware of their approach which is mainly developed by this group. If it is so, maybe then better submit their ms to local journal of lesser importance?

Author Response

The answer is attached as.pdf file

Round 3

Reviewer 2 Report

maybe accepted